# TimeDiT: General-purpose Diffusion Transformers for Time Series Foundation Model

## Abstract

With recent advances in building foundation models for text and video data, such as Large Language Models (LLMs), there is a surge of interest in foundation modeling for time series. However, real-world time series exhibit unique challenges, such as variable channel sizes across domains, missing values, and varying signal sampling intervals due to the multi-resolution nature of real-world data, which pose fundamental challenges for de-facto tailored transformer models to adapt complex and flexible data scenarios uniformly. Additionally, the unidirectional nature of temporally autoregressive decoding typically learns a deterministic mapping relationship and limits the incorporation of domain knowledge, such as physical laws. To address these challenges, we introduce the Time Diffusion Transformer (TimeDiT), a general foundation model for time series that jointly leverages the transformer inductive bias to capture temporal dependencies and the diffusion processes to generate high-quality candidate samples. The proposed mask unit for task-agnostic pretraining and task-specific sampling enables direct processing of multivariate inputs even with missing values or multi-resolution. Furthermore, we introduce a theoretically justified finetuning-free model editing strategy that allows the flexible integration of external knowledge during the sampling process. Extensive experiments conducted on a variety of tasks, such as forecasting, imputation, and anomaly detection highlight TimeDiT's adaptability as a foundation model, addressing diverse time series challenges and advancing analysis in various fields.

## 1 Introduction

Time series analysis is pivotal in a diverse set of applications, such as natural science, sustainability, health care, etc (Kamra et al., 2021; Cuomo et al., 2022). These applications are rooted in diverse domains, leading to time series with various distributions and a diverse set of analysis tasks including forecasting, imputation, anomaly detection, etc. Although significant progress has been made in developing specialized models like TCNs (Franceschi et al., 2019), LSTMs (Siami-Namini et al., 2019), GNNs (Wu et al., 2020), and Transformers (Zhang & Yan, 2022), the dataset- and task-specific design limits their generalizability. Inspired by the success of pre-trained models such as GPT (Radford et al., 2018) and ViT (Dosovitskiy et al., 2021) in achieving multiple downstream tasks in natural language processing and computer vision, recent studies have explored universal time series models. These models, trained on diverse datasets, can perform zero-shot forecasting on unseen time series (Ansari et al., 2024; Liu et al., 2024b; Gruver et al., 2024). However, time series data (TSD) emphasizes temporal continuity and progression—unlike text data's discrete, hierarchical tokens and image data's continuous pixel grids with spatial patterns—leaving an open question remaining: *'Can single time series foundation model excel across diverse, realistic applications?'*

Moreover, real-world time series exhibit unique characteristics such as *missing values* (Kollovieh et al., 2023), *multi-resolution* (Niu et al., 2023), *irregular sampling* (Cao et al., 2023a), etc. These challenges are particularly prevalent in domains such as healthcare, where patient data may be inconsistently recorded, financial markets with varying trading frequencies, environmental monitoring systems where sensor failures can lead to data gaps or outliers, and large-scale systems that aggregate data from multiple sources at different time scales. However, current benchmark datasets (Li et al., 2018; Zhou et al., 2021; Alexandrov et al., 2020) often fail to reflect such real-world TSD's complexities, potentially leading to models that underperform in practical applications. In addition, time series processes are often governed by underlying *physical principles* (Meng et al., 2022). Incorporating

physics knowledge can further enhance model performance and interpretability, especially in data-scarce domains. Addressing the aforementioned challenges requires innovative approaches in data preprocessing, model architecture, and training strategies to create models that can seamlessly handle the diverse and complex nature of TSD with varying historical lengths and features.

Recently, the emergence of LLMs like GPT-4 (OpenAI, 2023) and LLaMA (Touvron et al., 2023) suggests the potential for building time series foundation models handling multiple time series tasks under scaling-laws (Edwards et al., 2024). Previous works typically adopt transformer architecture with autoregressive processes as the de-facto choice of backbone. However, these approaches have the following limitations, which restrict the model's practical value in real-world: First, their tokenization methods, such as patching (Woo et al., 2024a), discretization tokens (Talukder et al., 2024), and feature-based tokens (Ansari et al., 2024), has inherent parameter sensitivity, creating a critical bottleneck in foundation model development, as tokens optimized for specific datasets often fail to generalize across real-world scenarios where data characteristics exhibit dynamic shifts. Second, most existing approaches employ a channel independence strategy (Nie et al., 2023), which, while facilitating model scaling, fails to capture the complex interplay between temporal patterns and cross-feature dependencies inherent in real-world time series data. Third, regression models typically learn a deterministic, unique mapping relationship from historical data, limiting their ability to capture the inherent uncertainties and stochastic nature of TSD. In contrast, diffusion models (Ho et al., 2020; Blattmann et al., 2023), offer a promising alternative to autoregressive methods for time series takes. These models reframe data generation as a series of conditional transformations, effectively recasting density estimation as sequential reconstruction. As diffusion models are well-poised to benefit from transformer inductive bias (Peebles & Xie, 2022), Diffusion Transformers present an opportunity to develop a versatile and robust time series foundation model.

In this work, we introduce TimeDiT, a diffusion transformer-based foundation model designed to process practical TSD across domains, frequencies, and sampling patterns. TimeDiT combines the transformer architecture's generalizability and expertise in capturing temporal dependencies with diffusion models' capacity to explore diverse solutions within a broad prior space, enabling the direct generation of high-quality samples. TimeDiT provides a novel paradigm that offers flexibility in handling varying input shapes and enables self-supervised learning (SSL) without external labels. Specifically, TimeDiT incorporates a comprehensive time series mask unit, featuring position, stride, and block masks for both task-agnostic pre-training and task-specific inference. This standardized pipeline handles multiple tasks without additional modules or parameters. By mirroring real-world scenarios of missing values, varying sampling rates, and partial observations, TimeDiT creates a unified framework that adapts to the diverse challenges inherent in time series analysis, from multi-horizon forecasting to irregular sampling, positioning them as ideal candidates for robust foundation models in temporal data processing. Furthermore, during the sampling stage, TimeDiT can incorporate physics knowledge as a theoretically grounded energy-based prior, generating samples that adhere to known physical laws, thereby enhancing sample quality and model applicability across various scientific and engineering contexts.

We evaluate TimeDiT on diverse datasets from real-world practical datasets, including traffic, climate, finance, and healthcare, as well as diverse, challenging time series tasks, including forecasting, imputation, anomaly detection, and synthetic data generation. The model's performance is compared against a spectrum of baselines, including linear-based, diffusion-based, transformer-based models, and other forecasting foundation models. Notably, TimeDiT achieved new state-of-the-art (SOTA) performance in uncertainty quantification (UQ) across real-world datasets for probabilistic forecasting with missing values or multi-resolution. In addition, the results of zero-shot experiments show that our model can be used as a foundation model even without fine-tuning, although fine-tuning may be necessary in some cases. Furthermore, TimeDiT's scalability and adaptability are evident in its ability to incorporate external knowledge, such as physical constraints, during the sampling stage. This combination of SOTA performance, adaptability across tasks, and the ability to incorporate domain knowledge naturally positions TimeDiT as a powerful and versatile foundation model.

## 2 RELATED WORK

**General Purpose Time Series Model** In the past decades, researchers have excelled in designing sophisticated models for specific time series analysis tasks (Zhang et al., 2024b; Fan et al., 2024a; Cao

et al., 2020; Bi et al., 2023; Zhang et al., 2021; Ye & Gao, 2022; Jia et al., 2024). However, the recent emergence of LLMs has inspired the development of general-purpose time series models and the field of time series has seen tremendous exploration efforts towards foundation models (Zerveas et al., 2021; Zhang et al., 2024a) . Specifically, (Gruver et al., 2024) simply encodes time series as strings while TimeLLM (Jin et al., 2023) convertes time series into language representations by alignment. TEMPO (Cao et al., 2023b) and S$^2$IP-LLM (Pan et al., 2024) further incorporate decomposition technique and prompt design and generalize to unseen data and multimodal scenarios. Additionally, many studies start to follow a two-stage training paradigm of pretraining and finetuning (Chang et al., 2023; Dong et al., 2024). However, previous works including Chronos (Ansari et al., 2024), TimeGPT (Garza & Mergenthaler-Canseco, 2023), UniTime (Liu et al., 2024a), TTM (Ekambaram et al., 2024) and Moirai (Woo et al., 2024b) mainly focus on the forecasting task only. (Zhou et al., 2023a) first adapted GPT2 as a general-purpose time series analysis model and extended it to various time series tasks. (Talukder et al., 2024) leveraged VQVAE as a tokenizer for transformer to handle time series tasks and (Ansari et al., 2024) employed a scaling and quantization technique to embed time series. For more detailed literatures of the general-purpose and foundation time series model, please refer to recent surveys (Liang et al., 2024; Jin et al., 2024b; Jiang et al., 2024)

**Diffusion models for Time Series**  Despite growing interest in diffusion models across various scenarios (Li et al., 2022a; Lu et al., 2024; Sui et al., 2024a;b), their application in time series analysis remains less explored compared to pre-trained language models. Most existing studies also focus solely on forecasting and the choice of backbone model also varies among VAE(Li et al., 2022b), RNN(Rasul et al., 2021), and transformers. Recently, CSDI (Tashiro et al., 2021) first utilizes a diffusion model for time series imputation with a self-supervised approach. SSSD (Alcaraz & Strodthoff, 2023) combines the structured state space model with the diffusion model for imputation. ImDiffusion (Chen et al., 2023) leverages diffusion models as time series imputers to achieve accurate anomaly detection. D$^3$VAE (Li et al., 2022b) proposes a generative time series forecasting method on top of VAE equipped with the diffusion model. Meanwhile, DiffusionTS (Yuan & Qiao, 2024) incorporates decomposition into the diffusion model to improve interoperability. Although TSDiff (Kollovieh et al., 2023) build a diffusion pipeline for multiple tasks with refinement, they still train different models for each task. Based on our knowledge, no unified diffusion transformer model has yet been explored for a comprehensive set of time series tasks. For a thorough literature review on diffusion models in time series analysis, please refer to  (Yang et al., 2024).

## 3    PRELIMINARIES OF DIFFUSION MODELS

In recent years, diffusion models have emerged as a promising approach to generative modeling. A diffusion process is a Markov chain that incrementally adds Gaussian noise to data over a sequence of steps, effectively destroying the data structure in the forward process and reconstructing the data structure during the reverse process.

**The forward process** adds noise to the data $\mathbf{x}_0$ over a series of timesteps $t$ according to a variance schedule $\beta_t$, resulting in a set of noisy intermediate variables $\mathbf{x}_1, \mathbf{x}_2, \ldots, \mathbf{x}_T$. Each subsequent $\mathbf{x}_t$ is derived from the previous step by applying Gaussian noise:

$$q(\mathbf{x}_t \mid \mathbf{x}_{t-1}) = \mathcal{N}(\mathbf{x}_t; \sqrt{1 - \beta_t}\mathbf{x}_{t-1}, \beta_t\mathbf{I}) \tag{1}$$

**The reverse process** aims to denoise the noisy variables step by step, sampling each $\mathbf{x}_{t-1}$ from the learned distribution $p_\theta(\mathbf{x}_{t-1} \mid \mathbf{x}_t)$. This distribution, modeled by a neural network parameterized by $\theta$, approximates the Gaussian distribution:

$$p_\theta(\mathbf{x}_{t-1} \mid \mathbf{x}_t) = \mathcal{N}(\mathbf{x}_{t-1}; \mu_\theta(\mathbf{x}_t, t), \Sigma_\theta(\mathbf{x}_t, t)) \tag{2}$$

By iterating this reverse process from $t = T$ down to $t = 0$, the model gradually reconstructs the original data from noise. Learning to clean $\mathbf{x}_T$ through the reversed diffusion process reduces to building a surrogate approximator to parameterize $\mu_\theta(\mathbf{x}_t, t)$ for all $t$. The reverse process learns to predict the mean and covariance of each intermediate distribution, effectively approximating the original data distribution.

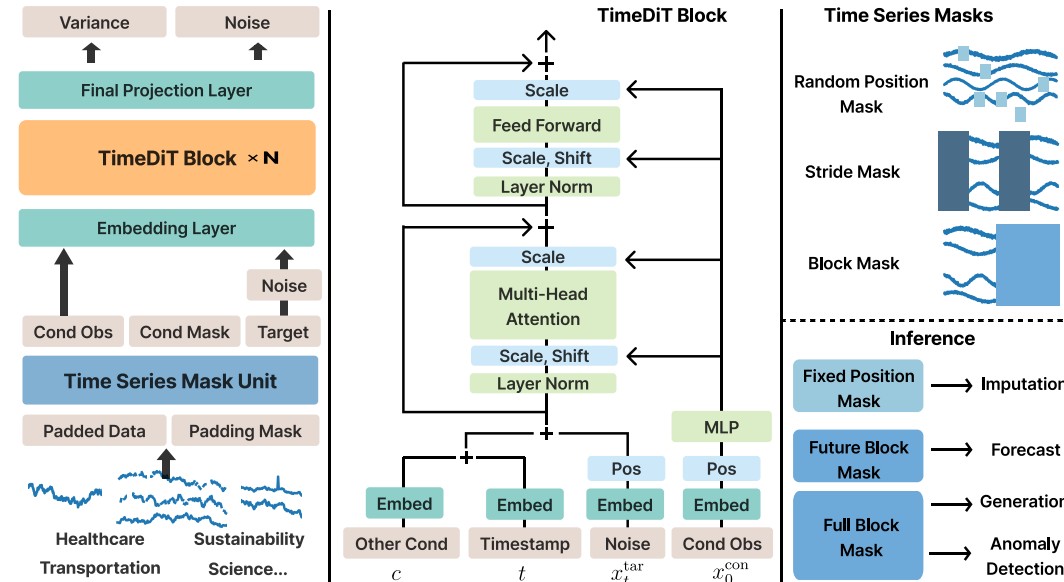

Figure 1: TimeDiT Architecture. Left: TimeDiT framework with diverse multivariate time series from different domains with multi-resolution or missing values; Middle: Structure of TimeDiT block; Right top: Illustration of masks generated by Time Series Mask Unit; Right bottom: Masks for downstream tasks that TimeDiT handles during inference.

## 4 METHODOLOGY

In this section, we present our main contributions: the proposed foundation model, TimeDiT , a diffusion model with the transformer backbone designed for multiple time series tasks. We first outline the uniform problem setting for multiple downstream tasks and offer an in-depth examination of the model architecture. Subsequently, we delve into the training pipeline with mask strategies, which help to build the training scheme in self-supervised learning for time series. Next, we present how to incorporate external information to improve the model's performance during inference stages by generating samples that better conform to real-world requirements. These extensions showcase the flexibility and adaptability of our proposed model, making it a powerful foundation model for a wide range of time series applications.

### 4.1 PROBLEM DEFINITION

We denote a multivariate time series as $\mathbf{X} = \{x_{i,j}\} \in \mathbb{R}^{K \times L}$, where $K$ is the number of features and $L$ is the length of the time series. Each individual entry $x_{i,j}$ represents the $j$-th feature at time step $i$, for $i \in \{1, \ldots, L\}$ and $j \in \{1, \ldots, K\}$. We define an observation mask $\mathbf{M_{obs}} = \{m_{i,j}\} \in \{0,1\}^{K \times L}$, where $m_{i,j} = 0$ if $x_{i,j}$ is missing, otherwise, $m_{i,j} = 1$. Let $\mathbf{x}_0^{\mathrm{obs}} \in X^{\mathrm{obs}}$ denote the observed subsequence; $\mathbf{x}_0^{\mathrm{tar}}$ denote the target subsequence of $\mathbf{x}_0^{\mathrm{obs}}$ which could be forecast target or imputation target or the whole sequence depending on the task. Let $\mathbf{x}_0^{\mathrm{con}}$ denote the unmasked partial observations in $\mathbf{x}_0^{\mathrm{obs}}$ which acts like self-conditions for the masked area $\mathbf{x}_0^{\mathrm{tar}}$. Let us use all subscripts of $\mathbf{x}$ to denote diffusion timestamp, and a subscript of 0 means no noise has been applied to the original data. Formally, the goal of our task is to approximate the true conditional time series distribution given the conditional information $q_{\mathbf{X}} \left( \mathbf{x}_0^{\mathrm{tar}} \mid \mathbf{x}_0^{\mathrm{con}} \right)$ with a model distribution $p_\theta(\mathbf{x}_0^{\mathrm{tar}} \mid \mathbf{x}_0^{\mathrm{con}})$, which can be calculated by a diffusion model with conditional information:

$$p_\theta \left( \mathbf{x}_{0:T}^{\mathrm{tar}} \mid \mathbf{x}_0^{\mathrm{con}} \right) := p \left( \mathbf{x}_T^{\mathrm{tar}} \right) \prod_{t=1}^{T} p_\theta \left( \mathbf{x}_{t-1}^{\mathrm{tar}} \mid \mathbf{x}_t^{\mathrm{tar}}, \mathbf{x}_0^{\mathrm{con}} \right), \mathbf{x}_T^{\mathrm{tar}} \sim \mathcal{N}(\mathbf{0}, \mathbf{I}), \text{where}$$

$$p_\theta \left( \mathbf{x}_{t-1}^{\mathrm{tar}} \mid \mathbf{x}_t^{\mathrm{tar}}, \mathbf{x}_0^{\mathrm{con}} \right) := \mathcal{N} \left( \mathbf{x}_{t-1}^{\mathrm{tar}}; \boldsymbol{\mu}_\theta \left( \mathbf{x}_t^{\mathrm{tar}}, t \mid \mathbf{x}_0^{\mathrm{con}} \right), \sigma_\theta \left( \mathbf{x}_t^{\mathrm{tar}}, t \mid \mathbf{x}_0^{\mathrm{con}} \right) \mathbf{I} \right). \quad (3)$$

The mask mechanism $\mathbf{M}$ plays a critical role in identifying the positions of $\mathbf{x}_0^{\mathrm{con}}$ and $\mathbf{x}_0^{\mathrm{tar}}$. By leveraging these positional differences, our model can adeptly adapt to tasks like forecasting, imputation, and anomaly detection in a unified framework.

## 4.2 TIME SERIES DIFFUSION TRANSFORMER

Figure 1 shows the overall framework of TimeDiT. Firstly, we establish $\mathbf{M_{obs}}$ and $\mathbf{x}_0^{obs}$ based on inputs with varying shapes, missing values, and multi-resolution data. By injecting placeholders, we identify corresponding positions and standardize input shapes across different time series, enabling more efficient and consistent processing. Then, the unified time series mask unit constructs $\mathbf{M}$ and adapts to diverse scenarios, generating $\mathbf{x}_0^{con}$ and $\mathbf{x}_0^{tar}$ with shape $\mathbb{R}^{B \times L \times K}$, where $B$ is the batch size. This enables TimeDiT to learn robust representations in a self-supervised manner by reconstructing the original sequence through denoising $\mathbf{x}_T^{tar}$. Adopting a "What You See Is What You Get" (WYSIWYG) design philosophy, our model represents tokens as direct, contiguous arrays of inputs. After that, the embedding layer with linear projection maps $\mathbf{x}_0^{con}$ and the noised $\mathbf{x}_0^{tar}$ into a continuous token space without vector quantization (Li et al., 2024), thereby preserving input integrity. To model the per-token probability distribution, the TimeDiT block is designed to autonomously learn cross-channel and temporal correlations through end-to-end training.

**Diffusion process.** TimeDiT unconditional diffusion process comprises a forward process that gradually adds noise to a data sample $x_0 \sim q(x)$, transforming it into Gaussian noise $x_T \sim \mathcal{N}(0, I)$ as defined by Eq. 1 and a reverse denoising process learned by a neural network (Eq. 2). To guide samples toward regions of high classifier likelihood, a self-conditional component $\mathbf{x}_0^{con}$ is integrated. We can train the denoising model $\boldsymbol{\mu}_\theta (\mathbf{x}_t^{tar}, \mathbf{x}_0^{con})$ in Eq. 3 using a weighted mean squared error (MSE) loss, which can be justified as optimizing a weighted variational lower bound on the data log-likelihood:

$$L(\mathbf{x}_0^{con}) = \sum_{t=1}^{T} \mathbb{E}_{q(\mathbf{x}_t^{tar}|\mathbf{x}_0^{con})} \|\mu(\mathbf{x}_t^{tar}, \mathbf{x}_0^{con}) - \mu_\theta(\mathbf{x}_t^{tar}, t|\mathbf{x}_0^{con})\|^2, \tag{4}$$

where $\mu(\mathbf{x}_t^{tar}, \mathbf{x}_0^{con})$ is the mean of the posterior $q(x_{t-1}^{tar}|\mathbf{x}_0^{con}, \mathbf{x}_t^{tar})$.

**Transformer-based Condition Injection.** TimeDiT employs a transformer-based architecture to process multivariate time series data. We feed the embedding of noised target series $\mathbf{x}_t^{tar}$ (with noise schedule $\beta_t \in (0, 1)$), and conditional observation $\mathbf{x}_0^{con}$ into the TimeDiT block, where the multi-head attention aims to then learns complex relationships within the data. During the diffusion process, unlike previous approaches (Peebles & Xie, 2022; Lu et al., 2024), we innovatively inject diffusion time information directly into the target noise as these represent universal information across the noised series. For self-conditional information, while a straightforward approach would be to include conditional information directly in the input sequence through concatenation (Rombach et al., 2022), we employ adaptive layer normalization (AdaLN) to control the scale and shift of $x_0^{tar}$ using partial observations $x_0^{con}$:

$$\text{AdaLN}(h, c) = c_{scale}\text{LayerNorm}(h) + c_{shift}, \tag{5}$$

where $h$ is the hidden state and $c_{scale}$ and $c_{shift}$ are the scale and shift parameters derived from the $x_0^{con}$. This method proved empirically more effective than simple input concatenation, as it leverages the scale and shift of $x_0^{con}$, which are crucial for capturing temporal continuity and progression.

**Time Series Mask Unit.** The Time Series Mask Unit is a key component of our model, designed to enhance its versatility and performance across various time series tasks. This unified mechanism incorporates multiple mask types that seamlessly integrate with the model throughout its lifecycle - from self-supervised task-agnostic pre-training to task-specific fine-tuning and inference. The time series mask unit generates four distinct mask types: random mask $\mathbf{M}^R$, block mask $\mathbf{M}^B$, stride mask $\mathbf{M}^S$, and reconstruction mask $\mathbf{M}^{Rec}$. During task-agnostic pre-training, these masks help the model develop robust and generalizable features from the input data, improving overall time series representation. In task-specific training, the masks adapt to the unique requirements of common downstream tasks such as forecasting and imputation, enabling the model to specialize effectively.

As shown in Figure 1 right top, given $\mathbf{x} \in \mathbb{R}^{K \times L}$, the random mask $\mathbf{M}^R$ can be generated by:

$$\mathbf{M}^R(x, r) = \begin{cases} 1 & z_{i,j} > r, z \in \mathbb{R}^{K \times L}, z \sim Uniform(0, 1) \\ 0, & otherwise, \end{cases} \tag{6}$$

where $r$ is the mask ratio. For task-specific training and inference, we allow the user to supply customized imputation masks, which replace the random position masks, that could handle the naturally missing data and multi-resolution cases. In addition, block mask $\mathbf{M}^B$ can be generated via:

$$\mathbf{M}^B(x, l) = \begin{cases} 1 & j < L - l, \\ 0, & otherwise, \end{cases} \tag{7}$$

---

**Algorithm 1** Physics-Informed TimeDiT through Energy-based Sampling

---

1: $\mathbf{x}_T \sim \mathcal{N}(\mathbf{0}, \mathbf{I})$
2: **for** $t = T, \ldots, 1$ **do**
3:    $\mathbf{z} \sim \mathcal{N}(\mathbf{0}, \mathbf{I})$ if $t > 1$, else $\mathbf{z} = \mathbf{0}$
4:    $\mathbf{x}_{t-1} = \frac{1}{\sqrt{\alpha_t}} \left( \mathbf{x}_t - \frac{1-\alpha_t}{\sqrt{1-\bar{\alpha}_t}} \boldsymbol{\epsilon}_\theta(\mathbf{x}_t, t) \right) + \sigma_t \mathbf{z}$
5: **end for**
6: **for** $j = 0, 1, .., k-1$ **do**
7:    $\mathbf{x}_{j+1}^{tar} = \mathbf{x}_j^{tar} + \epsilon \nabla K(\mathbf{x}_j^{tar}; \mathbf{x}^{obs}) + \alpha \epsilon \nabla \log p(\mathbf{x}_j^{tar} | \mathbf{x}^{obs}) + \sqrt{2\epsilon}\sigma, \sigma \sim \mathcal{N}(0, 1)$
8: **end for**
9: **return** $\mathbf{x}_k^{tar}$

---

where $l$ is the predicted length. This mask offers flexibility across different stages of model development and application: during pre-training, a random $l$ exposes the model to various forecasting horizons, while in fine-tuning and inference, a fixed $l$ aligns with specific task requirements. Moreover, stride mask $\mathbf{M}^{\text{S}}$, a variant of $\mathbf{M}^{\text{B}}$, is designed for intermittent placement within time series during task-agnostic pretraining:

$$\mathbf{M}^{\text{S}}(x, n_{\text{blocks}}) = \begin{cases} 1 & \lfloor \frac{j}{b} \rfloor \bmod 2 = 0 \\ 0 & \text{otherwise,} \end{cases} \tag{8}$$

where $n_{block}$ is the number of blocks; $b = \left\lceil \frac{L}{n_{\text{blocks}}} \right\rceil$ is the length of each block; $j$ is the index of the sequence. It improves the modeling of temporal and inter-correlated dependencies by integrating information across non-contiguous parts of time series, leveraging neighboring values as additional context. In addition, reconstruction mask $\mathbf{M}^{\text{Rec}} = 0$ is employed for tasks such as synthetic data generation and anomaly detection. It allows the direct generation of synthetic data or calculation of anomaly scores for each temporal position by comparing the original and reconstructed series.

### 4.3 PHYSICS-INFORMED TIMEDIT

Physics principles are fundamental in shaping the evolution of temporal signals observed in real-world phenomena, such as climate patterns and oceanographic data. Therefore, it is essential to integrate physical knowledge into foundational time series models. In this section, we aim at developing a decoding method that can ensure the $\mathbf{x}^{\text{tar}}$ generated by TimeDiT to satisfy our prior knowledge to the physical laws. To this end, we propose a strategy to incorporate physics knowledge as an energy-based prior for TimeDiT during inference, which iteratively refines the reverse diffusion process. By guiding the denoising process during inference with gradients derived from physical laws represented by partial differential equations (PDEs), the integration of this knowledge can ensure $\mathbf{x}^{\text{tar}}$ to satisfy the PDEs and significantly enhance the quality of the generated samples.

We first start with a brief introduction to physical laws and PDE. A generic form of a physical law represented as a PDE that describes the evolution of a continuous temporal signal $\mathbf{x}(\mathbf{u}, t)$ over a spatial coordinate $\mathbf{u}$ is given by:

$$\frac{\partial \mathbf{x}}{\partial t} = F\left(t, \mathbf{x}, \mathbf{u}, \frac{\partial \mathbf{x}}{\partial \mathbf{u}_i}, \frac{\partial^2 \mathbf{x}}{\partial \mathbf{u}_i \partial \mathbf{u}_j}, \ldots\right) \tag{9}$$

Based on this PDE representation of physical knowledge, the consistency between the predicted time series $\mathbf{x}^{\text{tar}}$ and the physics knowledge can be quantified using the following squared residual function:

$$K(\mathbf{x}^{\text{tar}}; F) = -\|\frac{\partial \mathbf{x}^{\text{tar}}}{\partial t} - F(t, \mathbf{x}^{\text{tar}}, \mathbf{u}, \frac{\partial \mathbf{x}^{\text{tar}}}{\partial \mathbf{u}_i}, \frac{\partial^2 \mathbf{x}^{\text{tar}}}{\partial \mathbf{u}_i \partial \mathbf{u}_j}, \ldots)\|_2^2 \tag{10}$$

This function reaches its maximum when the predicted time series is perfectly consistent with the physical model, resulting in a residual of 0. Using this metric $K$, physics knowledge can be integrated into a probabilistic time series foundation model $p(\mathbf{x}^{\text{tar}}|\mathbf{x}^{\text{con}})$ as an explicit regularization by solving the following optimization problem to obtain a refined model $q(\mathbf{x}^{\text{tar}}|\mathbf{x}^{\text{con}})$:

$$q(\mathbf{x}^{\text{tar}}|\mathbf{x}^{\text{con}}) = \arg\max_q \mathbb{E}_{\mathbf{x}^{\text{tar}} \sim q} K(\mathbf{x}^{\text{tar}}; F) - \alpha D_{KL}(q(\mathbf{x}^{\text{tar}}|\mathbf{x}^{\text{con}})||p(\mathbf{x}^{\text{tar}}|\mathbf{x}^{\text{con}})) \tag{11}$$

where the first term represents the aforementioned physics knowledge metric, and the second term controls the divergence between $q(\mathbf{x}^{tar}|\mathbf{x}^{con})$ and $p(\mathbf{x}^{tar}|\mathbf{x}^{con})$. However directly updating the model parameters to optimize the above function is resource-consuming. To solve this issue, we derived the closed-form solution, which does not need updating the model parameters. The above optimization problem has a closed-form solution as provided by the following theorem:

**Theorem 4.1.** *The optimal* $q(\mathbf{x}^{tar}|\mathbf{x}^{con})$ *in Eq.11 is the Boltzmann distribution defined on the following energy function:*

$$E(\mathbf{x}^{tar}; \mathbf{x}^{con}) = K(\mathbf{x}^{tar}; F) + \alpha \log p(\mathbf{x}^{tar}|\mathbf{x}^{con}) \tag{12}$$

*in other words, the optimal* $q(\mathbf{x}^{tar}|\mathbf{x}^{con})$ *is:*

$$q(\mathbf{x}^{tar}|\mathbf{x}^{con}) = \frac{1}{Z} \exp(K(\mathbf{x}^{tar}; F) + \alpha \log p(\mathbf{x}^{tar}|\mathbf{x}^{con})), \tag{13}$$

*where* $Z = \int \exp(K(\mathbf{x}^{tar}; F) + \alpha \log p(\mathbf{x}^{tar}|\mathbf{x}^{con}))d\mathbf{x}^{tar}$ *is the partition function.*

The theorem illustrates that sampling from the Boltzmann distribution defined in Eq. 12, is analogous to incorporating physics knowledge into model edition. In the context of diffusion models, this distribution can be effectively sampled using Langevin dynamics (Stoltz et al., 2010):

$$\begin{aligned}
\mathbf{x}_{j+1}^{tar} &= \mathbf{x}_j^{tar} + \epsilon \nabla \log q(\mathbf{x}^{tar}|\mathbf{x}^{con}) + \sqrt{2\epsilon}\sigma, \sigma \sim \mathcal{N}(0,1) \\
&= \mathbf{x}_j^{tar} + \epsilon \nabla K(\mathbf{x}_j^{tar}; \mathbf{x}^{con}) + \alpha\epsilon \nabla \log p(\mathbf{x}_j^{tar}|\mathbf{x}^{con}) + \sqrt{2\epsilon}\sigma, \sigma \sim \mathcal{N}(0,1)
\end{aligned} \tag{14}$$

In diffusion model, precisely calculate the likelihood $\log p(\mathbf{x}^{tar}|\mathbf{x}^{con})$ is intractable. To tackle this issue, following previous works (Kollovieh et al., 2023), we approximate likelihood with the objective to edit the pre-trained diffusion model: $\log p(\mathbf{x}^{tar}|\mathbf{x}^{con}) = -\mathbb{E}_{\epsilon,t}[||\epsilon_\theta(\mathbf{x}^{tar}, t; \mathbf{x}^{con}) - \epsilon||^2]$. The approximation presented above constitutes the optimizable component of the evidence lower bound(ELBO). Algorithm 1 summarizes the comprehensive model editing process.

## 5 EXPERIMENTS

We evaluate our time series foundation model on diverse tasks that mirror real-world challenges. Our assessment covers practical scenarios such as handling missing data and performing multi-resolution forecasting on custom datasets, including Air Quality from climate, MIMIC-III and PhysioNet from healthcare, and NASDAQ from finance. Additionally, we demonstrate the model's capability in physics-informed modeling by accurately processing six complex partial differential equations (PDEs) (Yuan & Qiao, 2024). We then assess the model's capabilities in well-established benchmarking tasks. These tasks include zero-shot forecasting on Solar, Electricity, Traffic, Taxi, and Exchange datasets(Tashiro et al., 2021) to evaluate temporal dependency modeling, imputation on ETTh, ETTm, Weather and Electricity datasets(Zhou et al., 2021) to assess the handling of missing data, anomaly detection on MSL, SMAP, SWaT, SMD, and PSM datasets(Xu et al., 2021; Zhao et al., 2020) to gauge sensitivity to unusual patterns, and synthetic data generation on Stock, Air Quality, and Energy datasets(Yoon et al., 2019; Desai et al., 2021) to test understanding of underlying distributions. By evaluating these diverse tasks, we can demonstrate that our model truly serves as a foundation for various time series applications, potentially reducing the need for task-specific models.

### 5.1 PRACTICAL SCENARIOS: MISSING DATA AND MULTI-RESOLUTION FORECASTING

To evaluate TimeDiT's performance in realistic scenarios, we conducted experiments incorporating three real-world challenges: missing values (validated on Air Quality, MIMIC), irregularly sampled time series (Jeon et al., 2022; Naiman et al., 2024) with varying time intervals between observations (evaluated on PhysioNet), multi-resolution data (tested on NASDAQ). We evaluated forecasting accuracy using Mean Absolute Error (MAE) and Mean Squared Error (MSE), while uncertainty quantification (UQ) was assessed using Continuous Ranked Probability Score (CRPS) and CRPS_sum. Results in Table 1 demonstrate that TimeDiT not only achieves high accuracy in point forecasts but also provides well-calibrated probabilistic forecasts, effectively capturing the inherent uncertainties in complex time series data. The model's strong performance in probabilistic metrics indicates its ability to generate reliable prediction intervals and accurately represent the full predictive distribution. This robust UQ capability, coupled with TimeDiT's ability to handle missing values and irregular samples without additional designs for interpolation, positions it as a powerful tool for decision-making in uncertain environments.

Table 1: Forecasting results on practical scenarios with both deterministic metric (MAE/MSE) for *accuracy* evaluation and probabilistic metric (CRPS/CRPS_sum) for *uncertainty quantification*. **Bold** indicates best result, Underline indicates the second best result.

| | Air Quality | MIMIC-III | PhysioNet(a) | PhysioNet(b) | PhysioNet(c) | NASDAQ |
|---|---|---|---|---|---|---|
| | MAE/MSE | MAE/MSE | MAE/MSE | MAE/MSE | MAE/MSE | MAE/MSE |
| DLinear | 0.683/0.685 | 0.786/1.000 | 0.686/0.758 | 0.733/0.922 | 0.715/0.813 | 2.715/8.137 |
| Neural ODE | 0.678/0.679 | 0.784/0.999 | 0.685/0.756 | 0.732/0.918 | 0.713/0.811 | 3.227/11.155 |
| Neural CDE | 0.683/0.685 | 0.787/1.002 | 0.688/0.754 | 0.733/0.921 | 0.713/0.814 | 3.319/11.816 |
| PatchTST | 0.685/0.683 | 0.778/0.987 | 0.699/0.780 | 0.733/0.932 | 0.714/0.802 | 3.182/10.635 |
| GPT2-3(OFA) | 0.696/0.701 | 0.750/0.921 | 0.697/0.772 | 0.734/0.921 | 0.713/0.817 | 3.176/10.873 |
| CSDI | 0.539/0.554 | 0.551/0.681 | **0.548/0.548** | 0.665/0.792 | 0.665/0.695 | 0.524/**0.388** |
| DiffTS | 0.521/0.538 | 0.677/0.908 | 0.610/0.742 | 0.701/0.880 | 0.678/0.872 | 1.951/9.515 |
| TimeMixer | 0.691/0.697 | 0.769/0.981 | 0.692/0.775 | 0.734/0.920 | 0.707/0.805 | 3.267/11.511 |
| TimeLLM | 0.701/0.705 | 0.787/1.020 | 0.687/0.761 | 0.731/0.931 | 0.713/0.800 | 3.125/10.276 |
| MG-TSD | 0.471/0.364 | - | - | - | - | 0.522/3.324 |
| TimeDiT | **0.457/0.354** | **0.517/0.534** | 0.577/0.620 | **0.659/0.766** | **0.543/0.561** | **0.516**/0.418 |
| | CRPS/_sum | CRPS/_sum | CRPS/_sum | CRPS/_sum | CRPS/_sum | CRPS |
| DLinear | 0.662/0.544 | 0.770/0.748 | 0.764/0.812 | 0.794/0.793 | 0.767/0.797 | 0.342 |
| Neural ODE | 0.657/0.529 | 0.769/0.733 | 0.763/0.806 | 0.792/0.789 | 0.765/0.793 | 0.426 |
| Neural CDE | 0.659/0.551 | 0.771/0.754 | 0.763/0.799 | 0.792/0.786 | 0.765/0.791 | 0.439 |
| PatchTST | 0.664/0.564 | 0.771/0.771 | 0.769/0.812 | 0.791/0.775 | 0.766/0.777 | 0.410 |
| GPT2-3(OFA) | 0.666/0.584 | 0.751/0.690 | 0.767/0.809 | 0.795/0.798 | 0.770/0.768 | 0.419 |
| CSDI | 0.598/0.620 | **0.504**/0.798 | 0.620/0.641 | 0.725/0.787 | 0.669/0.748 | 0.096 |
| DiffTS | 0.649/0.719 | 0.633/0.676 | 0.628/0.668 | 0.720/0.724 | 0.679/0.719 | 0.283 |
| TimeMixer | 0.667/0.576 | 0.776/0.724 | 0.763/0.805 | 0.794/0.798 | 0.757/0.784 | 0.432 |
| TimeLLM | 0.664/0.571 | 0.785/0.700 | 0.752/0.797 | 0.795/0.795 | 0.757/0.754 | 0.405 |
| MG-TSD | 0.579/0.564 | - | - | - | - | 0.275 |
| TimeDiT | **0.554/0.522** | 0.599/**0.649** | **0.616/0.640** | **0.708/0.710** | **0.668/0.708** | **0.091** |

Table 2: Physics-informed TimeDiT results for PDE forecasting, including both mean error and error bars. Lower values indicate better performance and closer adherence to physical laws.

| | MSE | RMSE | MAE | CRPS | MSE | RMSE | MAE | CRPS |
|---|---|---|---|---|---|---|---|---|
| | Advection | | | | Navier-Stokes | | | |
| DDPM | 0.011(0.000) | 0.106(0.001) | 0.084(0.001) | 0.472(0.007) | 0.309(0.004) | 0.556(0.004) | 0.332(0.005) | 0.415(0.006) |
| DDIM | 0.015(0.000) | 0.122(0.002) | 0.096(0.001) | 0.559(0.009) | 0.350(0.014) | 0.591(0.011) | 0.377(0.009) | 0.470(0.013) |
| TSDiff | 0.011(0.000) | 0.106(0.022) | 0.085(0.001) | 0.472(0.011) | 0.399(0.008) | 0.556(0.007) | 0.331(0.006) | 0.414(0.007) |
| TimeDiT | **0.010(0.000)** | **0.103(0.002)** | **0.082(0.001)** | **0.464(0.008)** | **0.299(0.006)** | **0.546(0.006)** | **0.322(0.06)** | **0.403(0.007)** |
| | Burgers | | | | Vorticity | | | |
| DDPM | 0.016(0.001) | 0.128(0.004) | 0.101(0.003) | 1.787(0.040) | 1.917(0.020) | 1.385(0.007) | 0.851(0.009) | 0.476(0.005) |
| DDIM | 0.018(0.000) | 0.136(0.001) | 0.116(0.001) | 1.858(0.015) | 1.567(0.031) | 1.252(0.012) | **0.754(0.012)** | **0.401(0.006)** |
| TSDiff | 0.017(0.001) | 0.129(0.005) | 0.102(0.004) | 1.800(0.055) | 1.966(0.073) | 1.402(0.026) | 0.866(0.010) | 0.485(0.005) |
| TimeDiT | **0.011(0.001)** | **0.104(0.005)** | **0.083(0.003)** | **1.395(0.053)** | **1.524(0.523)** | **1.234(0.021)** | 0.772(0.009) | 0.445(0.006) |
| | Diffusion Sorption | | | | CFD | | | |
| DDPM | 0.309(0.004) | 0.556(0.004) | 0.332(0.005) | 0.415(0.006) | 0.004(0.000) | 0.065(0.001) | 0.054(0.000) | 0.082(0.000) |
| DDIM | 0.349(0.013) | 0.591(0.011) | 0.377(0.009) | 0.470(0.013) | 0.039(0.002) | 0.194(0.006) | 0.188(0.006) | 0.313(0.012) |
| TSDiff | 0.309(0.008) | 0.556(0.007) | 0.331(0.006) | **0.414(0.007)** | N/A | N/A | N/A | N/A |
| TimeDiT | **0.284(0.005)** | **0.533(0.005)** | **0.327(0.005)** | 0.423(0.007) | **0.004(0.000)** | **0.062(0.001)** | **0.051(0.001)** | **0.080(0.001)** |

## 5.2 DOMAIN KNOWLEDGE INTEGRATION: PHYSICS-INFORMED TIMEDIT

Our approach enables the direct incorporation of physics knowledge into the pre-trained foundation model without fine-tuning. In this section, we evaluate how effectively our pre-trained foundation model can integrate physics-informed knowledge into time series forecasting. We study six 1D partial differential equations (PDEs) forecasting from (Takamoto et al., 2022): general Navier-Stokes Equations, Kolmogorov Flow (a specific case of Navier-Stokes Equations), Advection Equations, Burgers Equations, Diffusion Soeption and Computational Fluid Dynamics (CFD). Table 2 clearly demonstrates that our proposed model editing solution, which incorporates physics knowledge, significantly outperforms previous sampling strategies introduced in DDPM (Ho et al., 2020), DDIM (Song et al., 2021), and TS Diffusion, which proposes the Self-Guidance (Kollovieh et al., 2023) to improve sampling quality. By leveraging domain-specific physical information, our approach achieves substantial performance improvements over these baselines, highlighting the effectiveness of integrating physics-informed priors into the diffusion model sampling process. This represents a novel advance in scientific machine learning, enabling rapid adaptation to specific physical systems.

## 5.3 FORECASTING ON ZERO-SHOT SETTING

We evaluate TimeDiT's performance as a foundation model in a zero-shot forecasting setting, comparing it to leading transformer-based time series models. This crucial assessment tests the model's ability to generalize and adapt to entirely new datasets without prior exposure, highlighting its robustness and versatility. In our experiments, TimeDiT is benchmarked against open-sourced foundation models including TEMPO (Cao et al., 2023b), which employs a Student's t-distribution head for

Table 3: Zero-shot Forecasting results on CRPS_sum. Zero-shot implies that the model did not encounter any samples from the evaluating datasets during training.

| | Solar | Electricity | Traffic | Taxi | Exchange |
|---|---|---|---|---|---|
| TEMPO | 0.581(0.002) | 0.081(0.003) | 0.147(0.000) | 0.400(0.001) | 0.030(0.001) |
| Moirai(S) | 0.884(0.005) | 0.079(0.002) | 0.215(0.000) | 0.463(0.001) | **0.007(0.000)** |
| Moirai(B) | 0.948(0.002) | 0.072(0.002) | 0.191(0.001) | 0.428(0.000) | 0.012(0.000) |
| Moirai(L) | 1.042(0.002) | 0.039(0.001) | **0.111(0.000)** | 0.597(0.000) | 0.011(0.000) |
| LagLLaMA | 0.690(0.005) | 0.065(0.005) | 0.275(0.001) | 0.620(0.003) | 0.024(0.001) |
| TimeDiT | **0.457(0.002)** | **0.026(0.001)** | 0.185(0.010) | **0.398(0.001)** | 0.021(0.002) |

Table 4: Imputation result on 96-length multivariate time series averaged over the four mask ratios.

| Datasets | ETTh1 | | ETTh2 | | ETTm1 | | ETTm2 | | Weather | | Electricity | | 1st Pl |
|---|---|---|---|---|---|---|---|---|---|---|---|---|---|
| | MSE | MAE | MSE | MAE | MSE | MAE | MSE | MAE | MSE | MAE | MSE | MAE | Count |
| DLinear | 0.201 | 0.306 | 0.142 | 0.259 | 0.093 | 0.206 | 0.096 | 0.208 | 0.052 | 0.110 | 0.132 | 0.260 | 0 |
| LightTS | 0.284 | 0.373 | 0.119 | 0.250 | 0.104 | 0.218 | 0.046 | 0.151 | 0.055 | 0.117 | 0.131 | 0.262 | 0 |
| ETSformer | 0.202 | 0.329 | 0.367 | 0.436 | 0.120 | 0.253 | 0.208 | 0.327 | 0.076 | 0.171 | 0.214 | 0.339 | 0 |
| FEDformer | 0.117 | 0.246 | 0.163 | 0.279 | 0.062 | 0.177 | 0.101 | 0.215 | 0.099 | 0.203 | 0.130 | 0.259 | 0 |
| Autoformer | 0.103 | 0.214 | 0.055 | 0.156 | 0.051 | 0.150 | 0.029 | 0.105 | 0.031 | 0.057 | 0.101 | 0.225 | 0 |
| PatchTST | 0.115 | 0.224 | 0.065 | 0.163 | 0.047 | 0.140 | 0.029 | 0.102 | 0.034 | 0.055 | 0.072 | 0.183 | 0 |
| TimesNet | 0.078 | 0.187 | 0.049 | 0.146 | 0.027 | 0.107 | 0.022 | 0.088 | **0.030** | 0.054 | 0.092 | 0.210 | 1 |
| GPT2(3) | 0.069 | 0.173 | 0.048 | 0.141 | 0.028 | 0.105 | 0.021 | 0.084 | 0.031 | 0.056 | 0.090 | 0.207 | 1 |
| Timer | 0.145 | 0.243 | 0.077 | 0.172 | 0.051 | 0.141 | 0.035 | 0.105 | 0.108 | 0.168 | 0.097 | 0.194 | 0 |
| TimeMixer | 0.119 | 0.226 | 0.064 | 0.157 | 0.051 | 0.143 | 0.028 | 0.093 | 0.031 | 0.049 | 0.061 | 0.164 | 0 |
| iTransformer | 0.149 | 0.270 | 0.150 | 0.271 | 0.071 | 0.185 | 0.083 | 0.192 | 0.053 | 0.116 | 0.099 | 0.224 | 0 |
| TimeDiT | **0.042** | **0.135** | **0.042** | **0.139** | **0.023** | **0.098** | 0.024 | **0.083** | 0.031 | **0.036** | 0.069 | 0.174 | 10 |

probabilistic outputs, as well as Moirai (Woo et al., 2024b) and LagLLama (Rasul et al., 2023). The results, presented in Table 3, demonstrate TimeDiT's superior performance in most cases. This noteworthy achievement suggests that TimeDiT can be effectively applied to a wide range of time series forecasting tasks across diverse domains, underscoring its potential as a versatile foundation model for time series analysis.

## 5.4 IMPUTATION TASK

We conduct experiments on six benchmark time-series datasets: ETTh1, ETTh2, ETTm1, ETTm2, Electtricity, and Weather. We use random mask ratios $\{12.5\%, 25\%, 37.5\%, 50\%\}$ following previous studies' settings with sequence length set to 96. Table 4 shows the imputation result averaged over the four mask ratios. TimeDiT is finetuned using pre-trained checkpoints, which have already been encountered and learned from a wide range of data scenarios, including those with missing values. TimeDiT demonstrates superior performance, achieving the best results in 10 out of 12 evaluations, while all other baselines combined secured only 2 top positions. Notably, TimeDiT achieved a 39% reduction in MSE and 22% reduction in MAE compared to the strongest baseline on the ETTh1 dataset. For full result on each mask ratio, please refer to section D.2.

Table 5: Anomaly Detection result on 100-length multivariate time series. We calculate F1 score as % for each dataset. '.' notation in model name stands for transformer.

| Methods | TimeDiT | TimeMixer | iTrans. | GPT2(6) | TimesNet | PatchTS. | ETS. | FED. | LightTS | DLinear | Auto. | Anomaly. |
|---|---|---|---|---|---|---|---|---|---|---|---|---|
| MSL | **89.33** | 81.95 | 72.54 | 82.45 | 81.84 | 78.70 | 85.03 | 78.57 | 78.95 | 84.88 | 79.05 | 83.31 |
| SMAP | **95.91** | 67.63 | 66.76 | 72.88 | 69.39 | 68.82 | 69.50 | 70.76 | 69.21 | 69.26 | 71.12 | 71.18 |
| SWaT | **96.46** | 88.84 | 92.63 | 94.23 | 93.02 | 85.72 | 84.91 | 93.19 | 93.33 | 87.52 | 92.74 | 83.10 |
| SMD | 83.28 | 78.33 | 82.08 | **86.89** | 84.61 | 84.62 | 83.13 | 85.08 | 82.53 | 77.10 | 85.11 | 85.49 |
| PSM | **97.57** | 93.11 | 95.32 | 97.13 | 97.34 | 96.08 | 91.76 | 97.23 | 97.15 | 93.55 | 93.29 | 79.40 |
| 1st Pl Count | 4 | 0 | 0 | 1 | 0 | 0 | 0 | 0 | 0 | 0 | 0 | 0 |

## 5.5 ANOMALY DETECTION TASK

We conduct experiments on five real-world datasets from industrial applications: MSL, SMAP, SWaT, SMD, and PSM. As shown in Table 5, TimeDiT outperforms baseline models on four of the five datasets. Notably, on the SMAP dataset, TimeDiT achieves a remarkable 23.03-point improvement in F1 score compared to the previous best baseline. These results demonstrate the effectiveness of our approach in handling real-world anomaly detection scenarios across various industrial applications.

## 5.6 SYNTHETIC GENERATION TASK

We conduct experiments to synthesize multivariate time series and evaluate performance using the discriminative score and predictive score metrics under a "train on synthetic test on real" experimental

Table 6: Synthetic Generation results on 24-length multivariate time series. We calculate discriminative and predictive scores according to (Yoon et al., 2019).

| Metric | Methods | Sine | Stocks | Air Quality | Energy |
|---|---|---|---|---|---|
| Discriminative Score | TimeGAN | 0.1217(0.039) | 0.2038(0.057) | 0.3913(0.039) | 0.4969 (0.000) |
| | TimeVAE | 0.0489(0.0562) | 0.1987(0.037) | 0.2869(0.053) | 0.4993(0.001) |
| | Diffusion-TS | 0.0099(0.003) | 0.1869(0.0159) | **0.1227(0.006)** | 0.2301(0.006) |
| | TimeDiT | **0.0086(0.004)** | **0.0087(0.006)** | 0.1923(0.003) | **0.0053(0.002)** |
| Predictive Score | TimeGAN | 0.2797(0.015) | 0.0481(0.002) | 0.035(0.002) | 0.3305(0.003) |
| | TimeVAE | 0.2285(0.000) | 0.0485(0.000) | 0.0269(0.001) | 0.2878(0.001) |
| | Diffusion-TS | 0.2262(0.000) | **0.042(0.000)** | 0.022(0.002) | 0.2506(0.000) |
| | TimeDiT | **0.1915(0.000)** | 0.0445(0.000) | **0.0217(0.000)** | **0.2489(0.000)** |

setup with sequence length set to 24 (Yuan & Qiao, 2024). Table 6 shows the result on synthetic generation where TimeDiT, in general, consistently generates more realistic synthetic samples compared to baselines, even on challenging energy datasets. This demonstrates TimeDiT's strength in complex time series synthesis. PCA visualization of synthesis performance in Appendix D.3 shows that TimeDiT's samples markedly overlap the original data distribution better than other methods. Qualitative and quantitative results confirm TimeDiT's superior ability to model intricate characteristics for realistic time series synthesis, even on multidimensional, complex datasets.

### 5.7 MULTIMODAL TIMEDIT

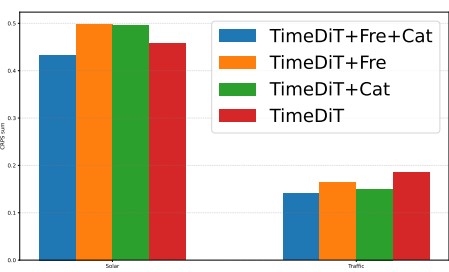

Figure 2: TimeDiT with textual information.

While textual information is intuitively crucial for precise time series analysis, effectively aligning textual and numerical data has remained challenging. To address this, we explore the integration of textual information as classifiers in TimeDiT, incorporating two key elements as guidance ($c$ in Figure 1): TSD's frequency (Fre) for capturing temporal periodicity, and TSD's categories (Cat) for representing domain-specific features. We pre-train three variants of TimeDiT and apply them in a zero-shot setting on Solar and Traffic datasets. The results demonstrate that utilizing both types of information significantly boosts zero-shot performance, indicating TimeDiT's capacity to leverage external information for rapid adaptation to both learned and specific representations. Comparing single-term guidance with the combined TimeDiT+Fre+Cat model reveals that precise, multi-faceted information is necessary to achieve optimal results. These experiments highlight that TimeDiT's integration of textual context improves forecasting accuracy, enabling more informed decision-making in real-world time series applications.

### 6 CONCLUSION

In this paper, we introduce TimeDiT, a pioneering approach to creating a versatile and robust foundation model for various time series tasks under practical scenarios. By integrating transformer inductive bias with diffusion model, TimeDiT effectively captures temporal dependencies and addresses real world challenges unique to time series regarding multi-resolution and missing values as well as incorporating external knowledge. Our innovative masking strategies allow for a consistent training framework adaptable to diverse tasks such as forecasting, imputation, and anomaly detection and synthetic data generation. We recognize some limitations of current work: first, we primarily explored common sequence lengths and did not assess TimeDiT's performance on very long sequences. While we have introduced randomness in prediction length and feature numbers up to a maximum, we aim to develop more scalable solutions for highly variable multivariate time series. Furthermore, our understanding of how different types of domain information contribute to performance improvement is still under investigation. In addition, we acknowledge the importance of sequence-level classification and are actively collecting datasets to extend TimeDiT's capabilities to classification tasks in future work. Lastly, there is a high demand for deeply developing foundation models for multi-modal time series, allowing TimeDiT to utilize diverse data sources for enhanced performance.

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

# Appendix

## A TimeDiT Paradigm on Training and Inference

**Position of TimeDiT** Rather than pursuing novelty through architectural complexity, our architectural choices reflect a careful balance between incorporating domain knowledge and maintaining general-purpose computational capabilities. TimeDiT exhibits the key characteristics of a foundation model - general-purpose architecture, multi-task capability, domain adaptability, and strong performance across diverse applications - making it a legitimate time series foundation model. First, it handles variable channel sizes and sequence lengths natively through its unified mask mechanism, allowing it to process diverse types of time series data without requiring task-specific architectures. Second, the model supports multiple downstream tasks including forecasting, imputation, anomaly detection, and synthetic data generation within a single framework. Third, TimeDiT incorporates physics-informed sampling through an energy-based approach, allowing it to integrate domain knowledge during inference without requiring model retraining. This combination of flexible architecture, task-agnostic design, and the ability to incorporate external knowledge positions TimeDiT as a powerful foundation model capable of addressing diverse time series challenges across various fields.

**Standardized pipeline** The TimeDiT paradigm introduces a novel approach to time series analysis, integrating information across continuous temporal segments to enhance the modeling of complex dependencies. Its core diffusion model establishes global statistical characteristics across domains, allowing flexible historical context without retraining. To handle heterogeneous data, TimeDiT employs an adaptive input processing mechanism, managing varying channel numbers and sequence lengths through intelligent padding and segmentation. Combining with mask units, we pre-define a maximum channel number $K_{max}$ and length $L_{max}$. Inputs with $k < K_{max}$ channels are padded to $K_{max}$, while those exceeding $K_{max}$ are segmented into $\lceil \frac{k}{K_{max}} \rceil$ blocks, each containing $K_{max}$ channels for independent processing. We apply front-padding to achieve uniform input dimensions up-to $L_{max}$. This approach efficiently handles high-dimensional data while maintaining positional integrity. The framework's versatility supports tasks like imputation, forecasting, and anomaly detection while providing confidence intervals for predictions. For example, with an input of 75 channels and $K_{max} = 40$, TimeDiT processes it in $\lceil 75/40 \rceil = 2$ blocks: Block 1 processes channels 1-40 directly, while Block 2 handles channels 41-75 with padding to 40 channels (35 actual + 5 padded). During sampling, the 5 padded channels in Block 2 are masked to prevent false information, and results from both blocks are integrated to reconstruct the full 75-channel output. This segmentation strategy ensures efficient processing while maintaining the integrity of the original data structure.

**Masking mechanism in practice** For the pretraining stage, we random select one conditional mask type from $\mathbf{M} = \{\mathbf{M}^{R}, \mathbf{M}^{S}, \mathbf{M}^{B}, \mathbf{M}^{Rec}\}$ for each instance. Our masking mechanism serves dual purposes: it enables both representation learning and downstream task design. The model's ability to handle varying sampling rates, incorporate physical constraints, and adapt to multiple tasks through a unified architecture demonstrates that this seemingly straightforward adaptation required non-trivial solutions to time series-specific challenges. TimeDiT's goal is to reconstruct the $\mathbf{x}^{tar}$, defined as $\mathbf{x}_0 \times (J - \mathbf{M})$ where $J$ is all-ones matrix, and $\mathbf{x}^{con}$ is defined as $\mathbf{x}_0 \times \mathbf{M}$. For each input, we randomly select one mask type (stride, random, or block) with randomly chosen parameters (Bengio et al., 2015). The prediction target spans 20-60% of the input length, ensuring adequate context. Stride masks improve representation, random masks enhance imputation for missing values and multi-resolution data, and block masks develop future prediction skills. We process each instance only once to prevent overfitting. The mask's stride number or block length is randomly determined, with the prediction length constrained to provide sufficient information. In addition, we randomly vary the mask ratio for each training instance. While increasing training complexity, this approach forces the model to learn robust patterns rather than memorizing specific mask configurations. To further enhance training, we could explore adaptive masking strategies, curriculum learning, or domain-specific masking patterns.

**Training details** Similar to the previous DiT work (Peebles & Xie, 2022), TimeDiT is available in four sizes: small (S, 33M parameters), big (B, 130M parameters), large (L, 460M parameters), and extra large (XL, 680M parameters). A comprehensive comparison in Table 8 shows TimeDiT's

expanded task coverage relative to existing general-purpose time series models, including anomaly detection, imputation and data generation. In our training process, we utilized the Adam optimizer with a learning rate of 0.0001 and the loss function is from Equantion 4. Batch sizes of 256 or 512 were employed, depending on model size. The ideal epoch to convergence is over 100 as the complexity of training data, but we choose to use the earlier checkpoint for the case of downstream purpose of anomaly detection and synthetic generation because the two tasks are very dataset-specific and do not necessarily benefit from learning distributions beyond the target dataset. In practice, the maximum channel number ($K_{max}$) was set to 40, with a maximum sequence length of 198, unless otherwise specified. All experiments were conducted on NVIDIA A100 GPUs with 40G GPU memory. Importantly, our zero-shot foundation model was trained without exposure to any data from the evaluated downstream tasks or datasets. For example, the forecasting foundation model was trained on multivariate datasets including ETT, weather, illness, air quality, cloud, and M4. In future work, we plan to incorporate a wider range of time series datasets to develop an even more robust foundation model, enhancing its generalization capabilities across diverse time series tasks.

**Inference** In the finetuning and inference stage, the choice of mask is tailored to align with the specific requirements of the user. This flexibility allows TimeDiT to apply the most appropriate masking strategy based on the context of the task and application. During inference, while the mask type and parameters are fixed for a given task to ensure consistency, TimeDiT's generative task architecture allows for flexible transformation of various downstream tasks. This adaptability enables us to address a wide range of time series challenges within a unified framework. Let $n$ represent the number of samples generated for each prediction, which we set to $n = 10$ ($n = 30$ for forecasting tasks) in our experimental setup at inference time. We use the median of these $n$ predictions as the final prediction, providing the added benefit of obtaining a confidence interval for TimeDiT's predictions. To prevent channel padding from affecting the generated samples, we mask out the invalid channels during sampling at each diffusion timestep so that TimeDiT does not falsely treat the information in the non-valid channels as meaningful information. Padding is applied at the beginning of the temporal dimension to ensure that the most relevant information remains at the end, thereby mitigating the effect of padding. We have included inference time comparisons for single-sample generation, where TimeDiT demonstrates superior computational efficiency, requiring only 1 second for single-sample generation, making it more practical for real-world applications.

Table 7: Comparison of inference times for single-sample generation.

| Model | Inference Time (mm:ss) |
| --- | --- |
| Diffusion-TS | 00:06 |
| CSDI | 00:02 |
| TimeDiT | 00:01 |

**Data usage strategy** Due to limited resources, we streamline our pre-training process by overlapping datasets to maximize reuse without compromising task-specific integrity. Specifically, to maintain comparability with current mainstream foundation models, we employ extensive pre-training data that may include datasets pertinent to imputation tasks. For instance, while our forecasting tasks utilize more practically relevant datasets excluding ETT, the ETT dataset itself is reserved exclusively for our final prediction models. In imputation tasks, we ensure that the pre-training datasets do not encompass ETT data. Furthermore, fine-tuning is performed on specific datasets without introducing additional external data. In essence, rather than selecting subsets from the pre-training datasets, we incrementally incorporate training data in a sequential manner to ensure fair and unbiased comparisons across tasks. In practice, once we specify the datasets for pre-training, our data loader randomly samples batches from the entire pool of available data. This means that during any training iteration, the model can encounter samples from any of the included datasets, ensuring truly randomized training. The final pre-trained model, therefore, learns from all datasets simultaneously, not sequentially.

Table 8: A comparable analysis of representative general purposes time series models

| Model | Parameter Size | Model Architecture | Channel Setting | Task Type | Pretrain Dataset | Data Size |
|---|---|---|---|---|---|---|
| Lag-LLama | - | Transformer | Univariate | Forecasting | Monash (Godahewa et al., 2021b) | 300 Million Time Points |
| Moriai | S: 14M
B: 91M
L: 311M | Transformer | Univariate | Forecasting | LOTSA (Woo et al., 2024b) | 27 Billion Time Points |
| TimeDiT | S: 33M
B: 130M
L: 460M
XL: 680M | Transformer + Diffusion | Multivariate | Forecasting,
Imputation,
Anomaly Detection,
Data Generation | Academic Public Dataset | 152 Million Time Points |

Table 9: Training Details. Imp stands for Imputation. SG stands for Syntheric Generation. AD stands for Anomaly Detection. FC stands for Forecasting

| Dataset | Task | Model Size | Hidden Size | Attention Head | Depth |
|---|---|---|---|---|---|
| ETTh | Imp test, FC pretrain | S | 384 | 6 | 12 |
| ETTm | Imp test, FC pretrain | S | 384 | 6 | 12 |
| Weather | Imp test, FC pretrain | S | 384 | 6 | 12 |
| Electricity | Imp test, FC pretrain | S | 384 | 6 | 12 |
| Air Quality | Imp test, FC pretrain | S | 384 | 6 | 12 |
| Sine | SG test, FC pretrain | S | 384 | 6 | 12 |
| Stock | SG test, FC pretrain | S | 384 | 6 | 12 |
| Energy | SG test, FC pretrain | S | 384 | 6 | 12 |
| MSL | AD test, AD pretrain | S | 384 | 6 | 12 |
| PSM | AD test, AD pretrain | S | 384 | 6 | 12 |
| SMAP | AD test, AD pretrain | S | 384 | 6 | 12 |
| SMD | AD test, AD pretrain | S | 384 | 6 | 12 |
| SWaT | AD test, AD pretrain | S | 384 | 6 | 12 |
| Air Quality | FC test, FC pretrain | S | 384 | 6 | 12 |
| MIMIC III | FC test, FC pretrain | S | 384 | 6 | 12 |
| PhysioNet | FC test, FC pretrain | S | 384 | 6 | 12 |
| NASDAQ | FC test, FC pretrain | S | 384 | 6 | 12 |
| Solar | FC zero shot test | B | 768 | 12 | 12 |
| Taxi | FC zero shot test | B | 768 | 12 | 12 |
| Traffic | FC zero shot test | B | 768 | 12 | 12 |
| Exchange | FC zero shot test | B | 768 | 12 | 12 |
| Electricity | FC zero shot test | B | 768 | 12 | 12 |

# B   EXPERIMENTS SETTING

## B.1   DATASETS

1. The ETT (Electricity Transformer Temperature) datasets (Zhou et al., 2021)[1] include electricity load data at various resolutions (ETTh & ETTm) from two different electricity stations.

2. The Weather dataset (Zhou et al., 2021)[2] comprises 21 meteorological indicators collected in Germany over the span of one year.

3. The Electricity (ECL, Electricity Consuming Load) (Zhou et al., 2021)[3] dataset provides information on electricity consumption.

4. PEMS-SF (Lai et al., 2018)[4] This dataset includes the San Francisco Traffic data, which comprises 862 hourly time series, depicting road occupancy rates on the San Francisco Bay Area freeways from 2015 to 2016.

5. The SMD dataset (Su et al., 2019) includes multivariate time-series data collected from server machines in a data center. It typically contains metrics such as CPU usage, memory usage, and disk activity.

---

[1]ETT: https://github.com/zhouhaoyi/ETDataset

[2]Weather:https://www.ncei.noaa.gov/data/local-climatological-data/

[3]ECL: https://archive.ics.uci.edu/ml/datasets/ElectricityLoadDiagrams20112014

[4]PEMS-SF: https://zenodo.org/records/4656132

6. The PSM dataset (Abdulaal et al., 2021) is used for predictive maintenance and includes sensor data from industrial machines. It often contains readings such as temperature, pressure, and vibration over time.

7. The MSL dataset (Hundman et al., 2018) comes from the Mars Science Laboratory mission, specifically the Curiosity rover. It includes telemetry data from the rover's sensors and systems.

8. The SWaT dataset (Mathur & Tippenhauer, 2016) originates from a scaled-down water treatment testbed designed to reflect a real-world water treatment process. It includes sensor and actuator data collected over time.

9. The SMAP dataset (Hundman et al., 2018) comes from NASA's Soil Moisture Active Passive (SMAP) mission, which measures soil moisture and freeze/thaw state. It includes time-series data from multiple sensors aboard the SMAP satellite.

10. The Sine dataset (Yoon et al., 2019) is synthetically generated by sinusoidal waves.

11. The Air Quality dataset (Yi et al., 2016) [5] contains hourly averaged readings from five metal oxide chemical sensors integrated into an Air Quality Chemical Multisensor Device. This device was positioned at road level in a highly polluted area of an Italian city. Data were collected from March 2004 to February 2005, making it the longest freely available record of on-field air quality chemical sensor responses.

12. The Stock dataset (Yoon et al., 2019) [6] contains daily historical Google stocks data from 2004 to 2019.

13. The UCI Appliances Energy prediction dataset (Yoon et al., 2019) [7] consists of multivariate, continuous-valued measurements including numerous temporal features measured at close intervals.

14. The Cloud dataset: The Huawei cloud datasets contain serverless traces (Joosen et al., 2023). Following (Rasul et al., 2023), we selected 8 time series containing metrics based on the minute-frequency occurrences of the top 10 functions over a period of 141 days. The metrics included in these series are: Function delay; Platform delay; CPU usage; Memory usage; CPU limit; Memory limit; Instances; Requests. The functions were chosen based on their median occurrences throughout the dataset.

15. The Weather_2 dataset (Godahewa et al., 2021a): The Weather_2 dataset comprises hourly climate TSD collected near Monash University, Clayton, Victoria, Australia, from January 2010 to May 2021. It includes series for temperature, dewpoint temperature, wind speed, mean sea level pressure, relative humidity, surface solar radiation, surface thermal radiation, and total cloud cover.

16. The PhysioNet dataset (Silva et al., 2012) [8] contains clinical time series data from 12,000 ICU patients, each with 42 vital variables recorded over 48 hours with naturally missing values. Patients are evenly divided into three groups of 4,000 each. For benchmarking purposes, we select 7 out of 42 variables. To address varying scales, we apply standard normalization, resulting in features with zero mean and unit variance.

17. MIMIC-III (Bica et al., 2020) [9]: MIMIC-III dataset contains 5000 patient ICU records with 19 variables from the lab events table including 'anion gap, albumin, bands, bicarbonate, bilirubin, creatinine, chloride, glucose, hematocrit, hemoglobin, lactate, platelet, potassium, PTT, INR, PT, sodium, BUN, WBC'. They are irregularly sampled and we process them following the previous works (Bica et al., 2020; Cao et al., 2023a), which have naturally missing values.

18. NASDAQ: NASDAQ Top 10 Stocks dataset comprises time series data for the ten largest companies by market capitalization listed on the NASDAQ stock exchange. The dataset includes daily and 5-day price data for each stock from 2014-2024, offering two temporal resolutions for comprehensive analysis. We predict the close prices of each company in the multi-resolution forecasting task.

---

[5] Air Quality: https://archive.ics.uci.edu/dataset/360/air+quality

[6] Stock: https://finance.yahoo.com/quote/GOOG

[7] Energy: https://archive.ics.uci.edu/ml/datasets

[8] The PhysioNet: https://physionet.org/content/challenge-2012/1.0.0/

[9] MIMIC-III: MIMIC-III:https://physionet.org/content/mimiciii/1.4/

19. Monash dataset archive (Godahewa et al., 2021b): The Monash repository contains 30 datasets, including publicly available time series datasets in various formats and those curated by us. Many datasets have different versions based on frequency and the inclusion of missing values. We use their multivariate time series version for pre-training and evaluation (specified if needed).

Table 10: Dataset details

| Dataset | Domain | Length | Dimension | Frequency |
|---|---|---|---|---|
| ETTh | Energy | 17420 | 7 | 1 hour |
| ETTm | Energy | 69680 | 7 | 15 min |
| Weather | Nature | 52696 | 21 | 10 min |
| Electricity | Energy | 26304 | 321 | 1 hour |
| Air Quality | Nature | 9357 | 13 | 1 hour |
| Sine | Synthetic | 10000 | 5 | N/A |
| Stock | Finance | 3685 | 6 | 1 day |
| Energy | Energy | 19745 | 28 | 10 min |
| MSL | Space | 132046 | 55 | 1 min |
| PSM | Cloud | 220322 | 25 | 1 min |
| SMAP | Space | 562800 | 25 | 1 min |
| SMD | Cloud | 1416825 | 38 | 1 min |
| SWaT | Energy | 944920 | 51 | 1 second |
| Requests Minute | Cloud | 64800 | 10 | 1 min |
| Function Delay Minute | Cloud | 64800 | 10 | 1 min |
| Platform Delay Minute | Cloud | 64800 | 10 | 1 min |
| Memory Usage Minute | Cloud | 64800 | 10 | 1 min |
| CPU Limit Minute | Cloud | 64800 | 10 | 1 min |
| Memory Limit Minute | Cloud | 64800 | 10 | 1 min |
| Instances Minute | Cloud | 64800 | 10 | 1 min |
| Weather_2 | Climate | 3001 | 695 | 1 day |
| PEMS_SF | Traffic | 4320 | 852 | 1 hour |
| PhysioNet(b) | Health Care | - | 7 | Irregular |
| PhysioNet(b) | Health Care | - | 7 | Irregular |
| PhysioNet(c) | Health Care | - | 7 | Irregular |
| MIMIC-III | Health Care | - | 19 | 1 day |
| NASDAQ | Finance | 2516 | 20 | Multiresolution |

## B.2 METRICS

**MAE** describes the mean absolute error that measures the absolute difference between ground truth and prediction.

$$\text{MAE} = \frac{1}{n} \sum_{i=1}^{n} |y_i - \hat{y}_i| \tag{15}$$

**MSE** describes the mean squared difference between ground truth and prediction.

$$\text{MSE} = \frac{1}{n} \sum_{i=1}^{n} (y_i - \hat{y}_i)^2 \tag{16}$$

**RMSE** is the sqaure root of MSE.

$$\text{RMSE} = \sqrt{\frac{1}{n} \sum_{i=1}^{n} (y_i - \hat{y}_i)^2} \tag{17}$$

**Discriminative score** Following TimeGAN, we train a post-hoc time-series classification model (by optimizing a 2-layer LSTM) to distinguish between sequences from the original and generated datasets. First, each original sequence is labeled real, and each generated sequence is labeled not real. Then, an off-the-shelf (RNN) classifier is trained to distinguish between the two classes as a standard supervised task. We then report the classification error on the held-out test set.

**Predictive Score**  Following TimeGAN, we train a post-hoc sequence-prediction model (by optimizing a 2-layer LSTM) to predict next-step temporal vectors over each input sequence. Then, we evaluate the trained model on the original dataset. Performance is measured in terms of the mean absolute error (MAE); for event-based data, the MAE is computed as the absolute value of 1 - estimated probability that the event occured.

**Computations of CRPS**  We explain the definition and calculation of the CRPS metric. The continuous ranked probability score (CRPS) assesses how well an estimated probability distribution $F$ aligns with an observation $x$. It is defined as the integral of the quantile loss $\Lambda_\alpha(q, z) := (\alpha - \mathbf{1}_{z < q})(z - q)$ over all quantile levels $\alpha \in [0, 1]$:

$$\text{CRPS}(F^{-1}, x) = \int_0^1 2\Lambda_\alpha(F^{-1}(\alpha), x)\, d\alpha \tag{18}$$

where $\mathbf{1}$ represents the indicator function. We then calculated quantile losses for quantile levels discretized in 0.05 increments. Thus, we approximated CRPS as follows:

$$\text{CRPS}(F^{-1}, x) \approx \frac{1}{19} \sum_{i=1}^{19} 2\Lambda_{i \cdot 0.05}(F^{-1}(i \cdot 0.05), x). \tag{19}$$

Next, we computed the normalized average CRPS for all features and time steps:

$$\text{CRPS Score} = \frac{\sum_{k,l} \text{CRPS}(F_{k,l}^{-1}, x_{k,l})}{\sum_{k,l} |x_{k,l}|} \tag{20}$$

where $k$ and $l$ denote the features and time steps of the imputation targets, respectively. The lower the CRPS, the more accurate the model, i.e., the closer the predicted probability is to the observed outcome.

**Computations of CRPS_sum**  CRPS_sum measures CRPS for the distribution $F$ of the sum of all $K$ features, calculated by:

$$\text{CRPS\_sum Score} = \frac{\sum_l \text{CRPS}(F^{-1}, \sum_k x_{k,l})}{\sum_{k,l} |x_{k,l}|} \tag{21}$$

where $\sum_k x_{k,l}$ is the total of the forecasting targets for all features at time point $l$.

**Precision**  Precision measures the accuracy of positive predictions made by a model. It is defined as the ratio of true positives (TP) to the total number of predicted positives, which includes both true positives and false positives (FP). Mathematically, precision is expressed as:

$$\text{Precision} = \frac{TP}{TP + FP} \tag{22}$$

**Recall**  Recall, also known as sensitivity, measures a model's ability to correctly identify true positive instances. It is calculated as the ratio of true positives (TP) to the sum of true positives and false negatives (FN). In the context of anomaly detection, failing to detect an anomalous timestamp can have serious consequences, making recall a critical metric. Mathematically, recall is defined as:

$$\text{Recall} = \frac{TP}{TP + FN} \tag{23}$$

**F1-score**  The F1-score is a balanced measure of model performance that combines Recall and Precision. It is calculated as the harmonic mean of these two metrics, giving equal importance to both. This score effectively captures the trade-off between Recall and Precision, penalizing significant disparities between them. By providing a single, comprehensive metric, the F1-score offers a more holistic view of a model's effectiveness, particularly useful when dealing with imbalanced datasets.

$$\text{F1} = 2 \times \frac{Precision \times Recall}{Precision + Recall} \tag{24}$$

### B.3  BASELINES

We conduct a comprehensive comparative analysis, benchmarking TimeDiT against a diverse array of leading models in the field. Our analysis extends to state-of-the-art probabilistic models, encompassing TimeGAN (Yoon et al., 2019), TimeVAE (Desai et al., 2021), Diffusion-TS (Yuan & Qiao, 2024), CSDI (Tashiro et al., 2021), TimeGrad (Rasul et al., 2021), TransMAF (Rasul et al., 2020), GP-copula (Salinas et al., 2019), and TSDiff (Kollovieh et al., 2023). We also evaluate against cutting-edge deterministic models, including DLinear (Zeng et al., 2023), GPT-2 (Zhou et al., 2023b), TimesNet (Wu et al., 2023), PatchTST (Nie et al., 2023), ETSformer (Woo et al., 2022), FEDformer (Zhou et al., 2022), LightTS (Zhang et al., 2022), Autoformer (Wu et al., 2021), and Anomaly Transformer (Xu et al., 2021), LatentODE and LatentCDE(Rubanova et al., 2019), etc. Furthermore, we include comparisons with recent forecasting foundation models, such as TEMPO (Cao et al., 2023b), Moirai (Woo et al., 2024b), and LagLLama (Rasul et al., 2023). This extensive comparison allows us to thoroughly evaluate TimeDiT's performance across a wide spectrum of methodologies and architectures in time series modeling.

### B.4  PHYSICS EQUATIONS IN PHYSICS-INFORMED TIMEDIT

The Burgers Equation is:

$$\frac{\partial u}{\partial t} + u\frac{\partial u}{\partial x} - v\frac{\partial^2 u}{\partial x^2} = 0 \tag{25}$$

where $v$ is the diffusion term. We set the $v$ (diffusion term) as 0.1 and randomly sample a combination of sine waves as initial status

The Advection Equation is:

$$\frac{\partial u}{\partial t} + c\frac{\partial u}{\partial x} = 0 \tag{26}$$

where $c$ is the advection speed. We set the $c$ as 1.0 and randomly placed Gaussian peaks as initial status

The diffusion-reaction Equation is:

$$\frac{\partial u}{\partial t} - D\frac{\partial^2 u}{\partial x^2} - R(u) = 0 \tag{27}$$

where $D$ is the diffusion coefficient and $R(u)$ is the reaction term. Here, we apply a linear reaction term $R(u) = -k \cdot u$, where $k$ is the reaction speed. We set the $D$ as 1.0, $k$ as 0.1, and a Gaussian distribution with random parameters as initial status.

The Kolmogrov Flow is a specific case of NS equation. More specifically, it is described by:

$$\mathbf{u}(x, y, z, t) = \left( -\frac{\partial \psi}{\partial y}, \frac{\partial \psi}{\partial x}, 0 \right) \tag{28}$$

where the $psi$ is the flow function. It is usually set as:

$$\psi(x, y, z, t) = A\sin(kx)\cos(zy + \omega t) \tag{29}$$

where $A, k, w$ are hyperparameters.

## C    FURTHER DISCUSSION ON PHYSICS-INFORMED TIMEDIT

### C.1    PROOF OF PHYSICS-INFORMED TIMEDIT THEOREM 4.1

*Proof.* Let us consider the objective function:

$$
\begin{aligned}
O(q(y|x)) &= \mathbb{E}_{y \sim q(y|x)} K(y) - \alpha D_{KL}(q(y|x) || p(y|x)) \\
&= \mathbb{E}_{y \sim q(y|x)} K(y) - \alpha \int_y q(y|x) \log(\frac{q(y|x)}{p(y|x)}) dy \\
&= \int_y q(y|x)[K(y) + \alpha \log p(y|x) - \alpha \log q(y|x)] dy
\end{aligned}
\tag{30}
$$

We try to find the optimal $q(y|x)$ through Lagrange multipliers. The constraint of the above objective function is that $q(y|x)$ is a valid $\int_y q(y|x) dy = 1$. Thus, the Lagrangian is:

$$
\begin{aligned}
L(q(y|x), \lambda) &= \int_y q(y|x)[K(y) + \alpha \log p(y|x) - \alpha \log q(y|x)] dy - \lambda(\int_y q(y|x) dy - 1) \\
&= \int_y q(y|x)[K(y) + \alpha \log p(y|x) - \alpha \log q(y|x) - \lambda q(y|x)] dy + \lambda
\end{aligned}
\tag{31}
$$

We define $f(q(y|x), y, \lambda) = q(y|x)[K(y) + \alpha \log p(y|x) - \alpha \log q(y|x) - \lambda] + \lambda h(y)]$, where $h(y)$ can be the density function of any fixed distribution defined on the support set of $y$. Therefore, $L(q(y|x), \lambda) = \int_y f(q(y|x), y, \lambda) dy$. According to Euler-Lagrange equation, when the above Lagrangian achieve extreme point, we have:

$$
\frac{\partial f}{\partial q} = K(y) + \alpha \log p(y|x) - \alpha \log q(y|x) - \lambda - \alpha = 0
\tag{32}
$$

Thus, we have:

$$
\begin{aligned}
\alpha \log q(y|x) &= K(y) + \alpha \log p(y|x) - \log q(y|x) - \lambda - \alpha \\
q(y|x) &= \exp(\frac{1}{\alpha} K(y) + \log p(y|x) - \frac{\lambda}{\alpha} - 1) \\
&= \frac{1}{\exp(\frac{\lambda}{\alpha} + 1)} \exp(\frac{1}{\alpha} K(y) + \log p(y|x))
\end{aligned}
\tag{33}
$$

Meanwhile, since $\int_y q(y|x) dy = 1$, we have:

$$
\begin{aligned}
\int_y \exp(\frac{1}{\alpha} K(y) + \log p(y|x) - \frac{\lambda}{\alpha} - 1) dy &= 1 \\
\frac{1}{\exp(\frac{\lambda}{\alpha} + 1)} \int_y \exp(\frac{1}{\alpha} K(y) + \log p(y|x)) dy &= 1
\end{aligned}
\tag{34}
$$

Thus, we have $\exp(\frac{\lambda}{\alpha} + 1) = \int_y \exp(\frac{1}{\alpha} K(y) + \log p(y|x)) dy = Z$, leading to:

$$
q(y|x) = \frac{1}{Z} \exp(K(y) + \alpha \log p(y|x)), Z = \int \exp(K(y) + \alpha \log p(y|x)) dy
\tag{35}
$$

$\square$

### C.2    PHYSICS-INFORMED TIMEDIT VS. DIRECT PDE-BASED GENERATION TRAINING

The tension between physical constraints and learned distributions in TimeDiT is managed through a sophisticated energy-based optimization framework that combines two key components:

- the physics knowledge represented by function $K(x^{\text{tar}}; F)$, which measures PDE residuals for physical law conformity
- the learned probabilistic distribution $p(x^{\text{tar}} | x^{\text{con}})$ from the diffusion model

Table 11: Comparison on the physics informed zero-shot TimeDiT with fully trained baselines.

| | | MSE | RMSE | MAE | CRPS |
|---|---|---|---|---|---|
| **Burgers** | | | | | |
| Full-shot | DLinear | 0.031(0.002) | 0.175(0.001) | 0.12610.005) | 1.400(0.057) |
| | PatchTST | 0.029(0.001) | 0.170(0.001) | 0.125(0.004) | 1.411(0.051) |
| | NeuralCDE | 0.031(0.002) | 0.176(0.002) | 0.126(0.005) | 1.397(0.061) |
| Zero-shot | TimeDiT | **0.011(0.001)** | **0.104(0.005)** | **0.083(0.003)** | **1.395(0.053)** |
| **Vorticity** | | | | | |
| | | MSE | RMSE | MAE | CRPS |
| Full-shot | DLinear | 2.650(0.003) | 1.628(0.001) | 1.459(0.010) | 0.695(0.005) |
| | PatchTST | 2.651(0.002) | 1.628(0.002) | 1.460(0.012) | 0.700(0.001) |
| | NeuralCDE | 2.631(0.001) | 1.622(0.001) | 1.453(0.010) | 0.691(0.005) |
| Zero-shot | TimeDiT | **1.524(0.523)** | **1.234(0.021)** | **0.772(0.009)** | **0.445(0.006)** |

This balance is achieved through an energy function:

$$E(x^{\text{tar}}; x^{\text{con}}) = K(x^{\text{tar}}; F) + \alpha \log p(x^{\text{tar}}|x^{\text{con}})$$

where the parameter $\alpha$ controls the trade-off between physical consistency and distribution fidelity.

Rather than directly modifying model parameters, TimeDiT implements this balance through an iterative sampling procedure that:

1. starts with samples from the learned distribution
2. gradually refines them using physical gradients while maintaining probabilistic characteristics

This approach allows the model to generate samples that respect both the learned patterns in the data and the underlying physical laws without significantly compromising either aspect, ultimately resolving the tension through a theoretically-grounded Boltzmann distribution as the optimal solution.

Physics-informed machine learning represents an active research area where physical constraints guide model outputs toward realistic solutions (Meng et al., 2022). Our physics-informed TimeDiT offers a novel approach that addresses key limitations of traditional PDE-based training methods. While direct use of PDE-based solvers to generate samples and then training is possible, TimeDiT provides crucial advantages in efficiency and flexibility. Our model incorporates physical knowledge during inference through energy-based sampling that guides the reverse diffusion process. This means we can flexibly integrate different physical constraints without any model retraining or parameter updates. We conduct an experimental comparison with direct PDE-based training methods. Using PDE solvers, we generated 5,000 training samples per scenario and trained three baseline models: DLinear, PatchTST, and NeuralCDE. Notably, zero-shot TimeDiT outperformed these models. For 6 PDE equations, the traditional approach required training 18 distinct models, resulting in significant computational overhead - approximately 18 times more training time - and extensive code modifications. This approach becomes increasingly impractical in real-world applications where multiple physical laws interact, as each new constraint would require training additional dedicated models. In contrast, TimeDiT's unified framework incorporates various physical constraints during inference while maintaining a single trained model, providing a more efficient and scalable solution.

# D  DETAILED EXPERIMENT RESULTS

## D.1  FORECASTING

### D.1.1  PRACTICAL FORECASTING SETTING

**Setting of Table 1.** The Nasdaq dataset features two resolutions (daily and 5-day intervals), using 168 historical steps to predict 30 future steps. The Air Quality dataset, containing natural missing values, also uses 168 steps to predict 30. For healthcare datasets, we group and normalize patient records individually. In PhysioNet, we select trajectories longer than 10 steps, using 96 to predict 24.

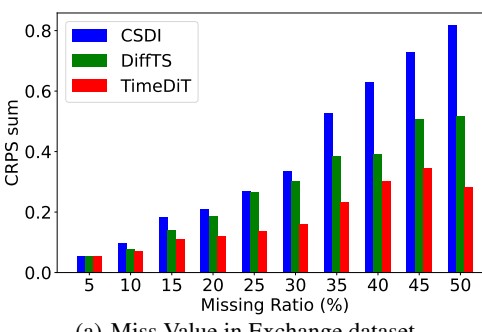 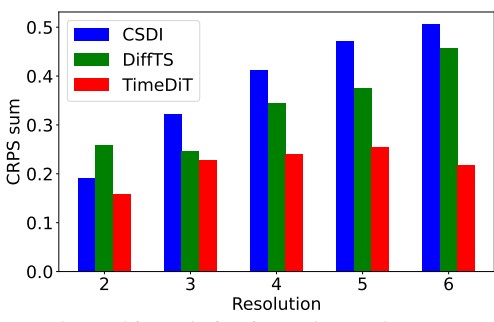

(a) Miss Value in Exchange dataset     (b) Multi-resolution in Exchange dataset

Figure 3: Visualization of miss value (a) and multiresolution (b) forecasting results on the Exchange dataset and miss value (c) and multiresolution (d) forecasting results on the Traffic dataset. Compared between our model TimeDiT and state-of-the-art diffusion-based methods. The x-axis number in (b) is the sampling skip in the resolutions in the multivariate input.

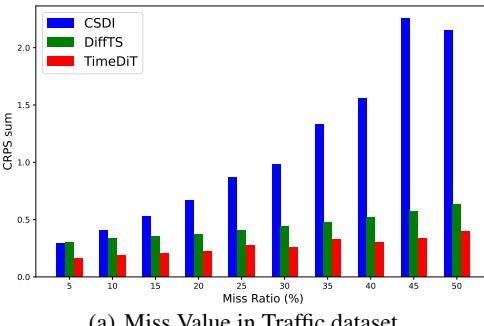 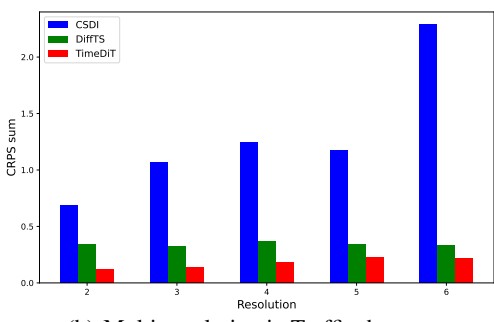

(a) Miss Value in Traffic dataset     (b) Multi-resolution in Traffic dataset

Figure 4: Visualization of miss value (a) and multi resolution (b) forecasting results on the Traffic (PEMS-SF) dataset. Compared between our model TimeDiT and state-of-the-art diffusion-based methods. The x-axis number in (b) is the sampling skip in the resolutions in the multivariate input.

For MIMIC-III, we choose trajectories between 10 and 40 steps, using 27 to predict 3 due to shorter lengths. This diverse dataset collection enables comprehensive evaluation of TimeDiT across various temporal resolutions and domain-specific challenges, spanning financial forecasting, environmental monitoring, and healthcare predictive modeling. We compare TimeDiT with state-of-the-art models in two categories: deterministic forecasting models adapted with a Student's t-distribution head for probabilistic outputs, and inherently diffusion-based probabilistic time series forecasting SOTA models. All baseline models are trained in a full-shot setting, while TimeDiT leverages a pre-trained foundation model, fine-tuning it on realistic datasets. Notably, TimeDiT can naturally handle input data with missing values, eliminating the need for additional imputation methods. This capability allows TimeDiT to perform forecasting directly using learned representations, even in the presence of incomplete data.

### D.1.2 MORE PRACTICAL FORECASTING RESULTS

**More results on miss-value and multi-resolution setting.** To further evaluate the practical ability of our proposed TimeDiT, we built two cases based on the previous dataset: the missing value scenario, where we created datasets with various missing ratios, simulating incomplete data often encountered in practice. In the multi-resolution setting, we sampled each individual time series within the multivariate dataset at different resolutions, reflecting the diverse sampling frequencies often present in real-world data collection. Figure 3 and Figure 4 illustrate TimeDiT's performance in realistic scenarios, showcasing its effectiveness across different sampling frequencies on the Exchange dataset. In Figure 3(a) and Figure 4(a), we observe TimeDiT's superior performance in handling missing data. As the missing ratio increases from 5% to 50%, TimeDiT maintains the lowest CRPS_sum across all scenarios, indicating its robustness to data gaps. The performance gap between TimeDiT and other models widens as the missing ratio increases, highlighting its effectiveness in more challenging conditions. Figure 3(b) and Figure 4(b) demonstrate TimeDiT's ability to manage multi-resolution data, where it maintains a clear performance advantage as the number of different sampling resolutions increases from 2 to 6. This demonstrates its ability to effectively integrate and forecast TSD sampled at varying frequencies.

**More results on advanced models.** As shown in Table 12, TimeDiT demonstrates superior performance against state-of-the-art models across diverse paradigms, consistently outperforming both TimeMixer (Wang et al., 2024) and TimeLLM (Jin et al., 2024a) across all evaluated datasets. The model shows particularly remarkable improvements in challenging scenarios, achieving substantially lower Mean Absolute Error (MAE) on MIMIC-III (0.517 versus 0.769/0.787) and NASDAQ (0.516 versus 3.267/3.125). In comparison with the diffusion-based MG-TSD model (Fan et al., 2024b), TimeDiT achieves comparable or superior performance on compatible datasets (Air Quality: 0.457 versus 0.471 MAE; NASDAQ: 0.516 versus 0.522 MAE). Notably, TimeDiT's architectural flexibility enables it to process irregular sampling patterns and heterogeneous inputs in datasets like MIMIC-III and PhysioNet, which exceed MG-TSD's capabilities. Furthermore, TimeDiT exhibits enhanced probabilistic forecasting capabilities, as evidenced by improved Continuous Ranked Probability Score (CRPS) metrics across all datasets. These comprehensive results validate our unified approach to time series modeling, demonstrating that TimeDiT not only competes with specialized models but often surpasses them while offering broader applicability and enhanced flexibility.

Table 12: Forecasting results on practical scenarios

| Model | Air Quality | MIMIC-III | PhysioNet(a) | PhysioNet(b) | PhysioNet(c) | NASDAQ |
|---|---|---|---|---|---|---|
| | **MAE/MSE** | **MAE/MSE** | **MAE/MSE** | **MAE/MSE** | **MAE/MSE** | **MAE/MSE** |
| TimeMixer | 0.691/0.697 | 0.769/0.981 | 0.692/0.775 | 0.734/0.920 | 0.707/0.805 | 3.267/11.511 |
| TimeLLM | 0.701/0.705 | 0.787/1.020 | 0.687/0.761 | 0.731/0.931 | 0.713/0.800 | 3.125/10.276 |
| MG-TSD | 0.471/0.364 | - | - | - | - | 0.522/3.324 |
| TimeDiT | **0.457/0.354** | **0.517/0.534** | **0.577/0.620** | **0.659/0.766** | **0.543/0.561** | **0.516/0.418** |
| | **CRPS/_sum** | **CRPS/_sum** | **CRPS/_sum** | **CRPS/_sum** | **CRPS/_sum** | **CRPS** |
| TimeMixer | 0.667/0.576 | 0.776/0.724 | 0.763/0.805 | 0.794/0.798 | 0.757/0.784 | 0.432 |
| TimeLLM | 0.664/0.571 | 0.785/0.700 | 0.752/0.797 | 0.795/0.795 | 0.757/0.754 | 0.405 |
| MG-TSD | 0.579/0.564 | - | - | - | - | 0.275 |
| TimeDiT | **0.554/0.522** | **0.599/0.649** | **0.616/0.640** | **0.708/0.710** | **0.668/0.708** | **0.091** |

Table 13: Evaluate time series dataset for forecasting tasks.

| | Features | Time Steps | History Length ($L_1$) | Prediction Horizon ($L_2$) | Rolling Windows | Frequency | Domain |
|---|---|---|---|---|---|---|---|
| Solar | 137 | 7009 | 168 | 24 | 7 | 1 hour | $\mathbb{R}^+$ |
| Electricity | 370 | 5833 | 168 | 24 | 7 | 1 hour | $\mathbb{R}^+$ |
| Traffic | 963 | 4001 | 168 | 24 | 7 | 1 hour | $(0, 1)$ |
| Taxi | 1214 | 1488 | 48 | 24 | 56 | 30 mins | $\mathbb{N}$ |
| Exchange | 8 | 6071 | 60 | 30 | 5 | 1 day | $\mathbb{R}^+$ |

Table 14: Forecasting results on CRPS_sum with full shot setting.

| | | Solar | Electricity | Traffic | Taxi | Exchange |
|---|---|---|---|---|---|---|
| Deterministic | DLinear | 0.432(0.002) | 0.033(0.000) | 0.070(0.001) | 0.177(0.000) | 0.011(0.001) |
| | PatchTST | 0.457(0.019) | 0.037(0.002) | 0.405(0.001) | 0.190(0.005) | 0.026(0.001) |
| | Latent ODE | 0.445(0.002) | 0.140(0.017) | 0.095(0.004) | 0.181(0.006) | 0.013(0.001) |
| | GPT2(6) | 0.467(0.002) | 0.033(0.001) | 0.069(0.001) | 0.187(0.001) | 0.013(0.001) |
| Probabilistic | GP-copula | 0.337(0.024) | 0.024(0.002) | 0.078(0.002) | 0.208(0.183) | 0.007(0.000) |
| | TransMAF | 0.301(0.014) | 0.021(0.011) | 0.056(0.001) | 0.179(0.002) | 0.005(0.003) |
| | TimeGrad | 0.287(0.020) | 0.021(0.001) | 0.044(0.006) | **0.114(0.020)** | 0.006(0.001) |
| | CSDI | 0.298(0.004) | **0.017(0.000)** | 0.020(0.001) | 0.123(0.003) | 0.007(0.001) |
| | Diffusion-TS | 0.286(0.003) | 0.019(0.002) | 0.097(0.001) | 0.303(0.004) | 0.009(0.001) |
| | TimeDiT | **0.278(0.001)** | **0.017(0.000)** | **0.019(0.000)** | 0.123(0.001) | **0.005(0.001)** |

### D.1.3 FULL-SHOT FORECASTING SETTING

For the full-shot benchmarking forecasting and zero-shot forecasting task, we utilized five widely-used open datasets to evaluate probabilistic time series forecasting performance. These datasets were collected in GluonTS (Alexandrov et al., 2020) and have been previously employed in (Tashiro et al., 2021; Salinas et al., 2019):

- Solar[10]: Hourly solar power production records from 137 stations in Alabama State, as used in (Lai et al., 2018).

- Electricity[11]: Hourly time series of electricity consumption for 370 customers, as used in (Asuncion & Newman, 2007).

- Traffic[12]: Hourly occupancy rates of 963 San Francisco freeway car lanes, with values between 0 and 1 (Asuncion & Newman, 2007).

- Taxi[13]: Half-hourly spatio-temporal time series of New York taxi rides taken at 1,214 locations, using data from January 2015 for training and January 2016 for testing, as proposed in (Tlc, 2017).

- Exchange rate[14]: Daily exchange rates between 8 currencies, namely Australia, the United Kingdom, Canada, Switzerland, China, Japan, New Zealand, and Singapore, as used in (Lai et al., 2018).

Table 13 summarizes the characteristics of each dataset. The task for these datasets is to predict the future $L_2$ steps given the observed $L_1$ steps. We set $L_1$ and $L_2$ values based on previous studies (Tashiro et al., 2021; Salinas et al., 2019). For training, we randomly selected $L_1 + L_2$ consecutive time steps as a single time series and designated the last $L_2$ steps as forecasting targets. We adhered to the train/test splits used in previous studies and utilized the last five samples of the training data as validation data. For the full-shot setting, we trained separate models on different datasets. Due to the large number of features in multivariate time series, we adopted subset sampling of features for training. For each input, we split them into subsets based on their order. If the last subset was smaller than the fixed shape, we applied padding to ensure equal input sizes across all subsets.

---

[10]Solar: https://www.nrel.gov/grid/solar-power-data.html

[11]Electricity:https://archive.ics.uci.edu/ml/datasets/ElectricityLoadDiagrams20112014

[12]Traffic_nips: https://archive.ics.uci.edu/dataset/204/pems_sf

[13]Taxi: https://www1.nyc.gov/site/tlc/about/tlc-trip-record-data

[14]Exchange: https://github.com/laiguokun/multivariate-time-series-data

### D.1.4 Full-shot Forecasting Results

In the full-shot forecasting task, we evaluate TimeDiT against various baselines using separate training and testing datasets to assess performance on conventional time series forecasting tasks. Table 14 presents the results, comparing TimeDiT with state-of-the-art models in two categories: deterministic forecasting models adapted with a Student's t-distribution head for probabilistic outputs, and inherently probabilistic time series forecasting models, including both diffusion-based (e.g., CSDI) and non-diffusion-based (e.g., GP-copula) approaches. Our model achieves the lowest CRPS_sum on four out of five datasets, securing the second-best performance on the Taxi dataset, demonstrating TimeDiT's robust performance across diverse time series forecasting scenarios and its ability to effectively learn and generalize from complete data.

### D.1.5 Additional zero-shot Forecasting Results

In zero-shot forecasting scenarios, as shown in Table 15, TimeDiT demonstrates remarkable performance advantages over contemporary models, including TimeMixer, TimeLLM, and Timer (Liu et al., 2024b). To ensure a fair comparison with these models, which were originally designed for deterministic

Table 15: Forecasting results on zero shot setting

| Model | Solar | Electricity | Traffic |
|---|---|---|---|
| TimeMixer | 0.999(0.001) | 0.302(0.003) | 0.403(0.015) |
| TimeLLM | 0.997(0.001) | 0.303(0.003) | 0.368(0.007) |
| Timer | 1.101(0.002) | 0.301(0.003) | 0.384(0.008) |
| TimeDiT | **0.457(0.002)** | **0.026(0.001)** | **0.185(0.010)** |

time series forecasting, we conducted pre-training using identical datasets under consistent conditions. The experimental results reveal substantial performance improvements across all evaluated datasets: TimeDiT achieves significantly lower error rates on Solar (0.457 compared to 0.999, 0.997, and 1.101 for TimeMixer, TimeLLM, and Timer respectively), Electricity (0.026 versus 0.302, 0.303, and 0.301), and Traffic datasets (0.185 compared to 0.403, 0.368, and 0.384). These consistent performance gains across diverse datasets underscore TimeDiT's superior capability in capturing and generalizing temporal patterns without task-specific fine-tuning, demonstrating its effectiveness as a robust zero-shot forecasting framework.

Table 16: Full result of imputation task.

| Methods / Mask Ratio | | TimeDiT MSE | TimeDiT MAE | Timer MSE | Timer MAE | TimeMixer MSE | TimeMixer MAE | iTransformer MSE | iTransformer MAE | GPT2(3) MSE | GPT2(3) MAE | TimesNet MSE | TimesNet MAE | PatchTST MSE | PatchTST MAE | ETSformer MSE | ETSformer MAE | LightTS MSE | LightTS MAE | DLinear MSE | DLinear MAE | FEDformer MSE | FEDformer MAE | Stationary MSE | Stationary MAE | Autoformer MSE | Autoformer MAE | Informer MSE | Informer MAE | Reformer MSE | Reformer MAE |
|---|---|---|---|---|---|---|---|---|---|---|---|---|---|---|---|---|---|---|---|---|---|---|---|---|---|---|---|---|---|---|---|
| ETTh1 | 12.5% | **0.025** | **0.107** | 0.119 | 0.222 | 0.094 | 0.203 | 0.099 | 0.221 | 0.043 | 0.140 | 0.057 | 0.159 | 0.093 | 0.201 | 0.126 | 0.263 | 0.240 | 0.345 | 0.151 | 0.267 | 0.070 | 0.190 | 0.060 | 0.165 | 0.074 | 0.182 | 0.114 | 0.234 | 0.074 | 0.194 |
| | 25% | **0.034** | **0.122** | 0.133 | 0.235 | 0.111 | 0.219 | 0.125 | 0.249 | 0.054 | 0.156 | 0.069 | 0.178 | 0.107 | 0.217 | 0.169 | 0.304 | 0.265 | 0.364 | 0.180 | 0.292 | 0.106 | 0.236 | 0.080 | 0.189 | 0.090 | 0.203 | 0.140 | 0.262 | 0.102 | 0.227 |
| | 37.5% | **0.047** | **0.143** | 0.151 | 0.249 | 0.124 | 0.233 | 0.158 | 0.281 | 0.072 | 0.180 | 0.084 | 0.196 | 0.120 | 0.230 | 0.220 | 0.347 | 0.296 | 0.382 | 0.215 | 0.318 | 0.124 | 0.258 | 0.102 | 0.212 | 0.109 | 0.222 | 0.174 | 0.293 | 0.135 | 0.261 |
| | 50% | **0.063** | **0.166** | 0.176 | 0.267 | 0.144 | 0.249 | 0.214 | 0.328 | 0.107 | 0.216 | 0.102 | 0.215 | 0.141 | 0.248 | 0.293 | 0.402 | 0.334 | 0.404 | 0.257 | 0.347 | 0.165 | 0.299 | 0.133 | 0.240 | 0.137 | 0.248 | 0.215 | 0.325 | 0.179 | 0.298 |
| | Avg | **0.042** | **0.135** | 0.145 | 0.243 | 0.119 | 0.226 | 0.149 | 0.270 | 0.069 | 0.173 | 0.078 | 0.187 | 0.115 | 0.224 | 0.202 | 0.329 | 0.284 | 0.373 | 0.201 | 0.306 | 0.117 | 0.246 | 0.094 | 0.201 | 0.103 | 0.214 | 0.161 | 0.279 | 0.122 | 0.245 |
| ETTh2 | 12.5% | **0.025** | **0.104** | 0.070 | 0.163 | 0.056 | 0.156 | 0.099 | 0.221 | 0.039 | 0.125 | 0.040 | 0.130 | 0.057 | 0.152 | 0.187 | 0.319 | 0.101 | 0.231 | 0.100 | 0.216 | 0.095 | 0.212 | 0.042 | 0.133 | 0.044 | 0.138 | 0.305 | 0.431 | 0.163 | 0.289 |
| | 25% | **0.037** | **0.129** | 0.074 | 0.168 | 0.063 | 0.157 | 0.130 | 0.251 | 0.044 | 0.135 | 0.046 | 0.141 | 0.061 | 0.158 | 0.279 | 0.390 | 0.115 | 0.246 | 0.127 | 0.247 | 0.137 | 0.258 | 0.049 | 0.147 | 0.050 | 0.149 | 0.322 | 0.444 | 0.206 | 0.331 |
| | 37.5% | **0.046** | **0.149** | 0.079 | 0.174 | 0.064 | 0.158 | 0.158 | 0.281 | 0.051 | 0.147 | 0.052 | 0.151 | 0.067 | 0.166 | 0.400 | 0.465 | 0.126 | 0.257 | 0.158 | 0.276 | 0.187 | 0.304 | 0.056 | 0.158 | 0.060 | 0.163 | 0.353 | 0.462 | 0.252 | 0.370 |
| | 50% | 0.062 | 0.173 | 0.085 | 0.182 | 0.071 | 0.168 | 0.214 | 0.328 | 0.059 | 0.158 | 0.060 | 0.162 | 0.073 | 0.174 | 0.602 | 0.572 | 0.136 | 0.268 | 0.183 | 0.299 | 0.232 | 0.341 | 0.065 | 0.170 | 0.068 | 0.173 | 0.369 | 0.472 | 0.316 | 0.419 |
| | Avg | **0.042** | **0.139** | 0.077 | 0.172 | 0.064 | 0.157 | 0.150 | 0.271 | 0.048 | 0.141 | 0.049 | 0.146 | 0.065 | 0.163 | 0.367 | 0.436 | 0.119 | 0.250 | 0.142 | 0.259 | 0.163 | 0.279 | 0.053 | 0.152 | 0.055 | 0.156 | 0.337 | 0.452 | 0.234 | 0.352 |
| ETTm1 | 12.5% | **0.016** | **0.083** | 0.044 | 0.131 | 0.046 | 0.136 | 0.045 | 0.147 | 0.017 | 0.085 | 0.023 | 0.101 | 0.041 | 0.130 | 0.096 | 0.229 | 0.093 | 0.206 | 0.080 | 0.193 | 0.052 | 0.166 | 0.032 | 0.119 | 0.046 | 0.144 | 0.063 | 0.180 | 0.042 | 0.146 |
| | 25% | **0.019** | **0.091** | 0.048 | 0.136 | 0.048 | 0.137 | 0.060 | 0.172 | 0.022 | 0.096 | 0.023 | 0.101 | 0.044 | 0.135 | 0.096 | 0.229 | 0.093 | 0.206 | 0.080 | 0.193 | 0.052 | 0.166 | 0.032 | 0.119 | 0.046 | 0.144 | 0.063 | 0.180 | 0.042 | 0.146 |
| | 37.5% | **0.025** | **0.102** | 0.053 | 0.144 | 0.059 | 0.155 | 0.078 | 0.190 | 0.029 | 0.111 | 0.029 | 0.111 | 0.049 | 0.143 | 0.133 | 0.271 | 0.113 | 0.231 | 0.103 | 0.219 | 0.069 | 0.191 | 0.039 | 0.131 | 0.057 | 0.161 | 0.079 | 0.200 | 0.063 | 0.182 |
| | 50% | **0.032** | **0.115** | 0.061 | 0.154 | 0.053 | 0.145 | 0.102 | 0.220 | 0.040 | 0.128 | 0.036 | 0.124 | 0.055 | 0.151 | 0.186 | 0.323 | 0.134 | 0.255 | 0.132 | 0.248 | 0.089 | 0.218 | 0.047 | 0.145 | 0.067 | 0.174 | 0.093 | 0.218 | 0.082 | 0.208 |
| | Avg | **0.023** | **0.098** | 0.051 | 0.141 | 0.051 | 0.143 | 0.071 | 0.185 | 0.028 | 0.105 | 0.027 | 0.107 | 0.047 | 0.140 | 0.120 | 0.253 | 0.104 | 0.218 | 0.093 | 0.206 | 0.062 | 0.177 | 0.036 | 0.126 | 0.051 | 0.150 | 0.071 | 0.188 | 0.055 | 0.166 |
| ETTm2 | 12.5% | **0.016** | **0.065** | 0.032 | 0.098 | 0.024 | 0.086 | 0.052 | 0.156 | 0.017 | 0.076 | 0.018 | 0.080 | 0.026 | 0.094 | 0.108 | 0.239 | 0.034 | 0.127 | 0.062 | 0.166 | 0.056 | 0.159 | 0.021 | 0.088 | 0.023 | 0.092 | 0.133 | 0.270 | 0.108 | 0.228 |
| | 25% | **0.022** | **0.078** | 0.034 | 0.102 | 0.026 | 0.090 | 0.071 | 0.170 | 0.020 | 0.080 | 0.020 | 0.085 | 0.028 | 0.099 | 0.164 | 0.294 | 0.042 | 0.143 | 0.085 | 0.196 | 0.080 | 0.195 | 0.024 | 0.096 | 0.026 | 0.101 | 0.135 | 0.272 | 0.136 | 0.262 |
| | 37.5% | 0.027 | **0.089** | 0.036 | 0.106 | 0.029 | 0.094 | 0.091 | 0.204 | 0.022 | 0.087 | 0.023 | 0.091 | 0.030 | 0.104 | 0.237 | 0.356 | 0.051 | 0.159 | 0.106 | 0.222 | 0.110 | 0.231 | 0.027 | 0.103 | 0.030 | 0.108 | 0.155 | 0.293 | 0.175 | 0.300 |
| | 50% | 0.031 | 0.099 | 0.040 | 0.112 | 0.032 | 0.101 | 0.117 | 0.232 | 0.025 | 0.095 | 0.026 | 0.098 | 0.034 | 0.110 | 0.323 | 0.421 | 0.059 | 0.174 | 0.131 | 0.247 | 0.156 | 0.276 | 0.030 | 0.108 | 0.035 | 0.119 | 0.200 | 0.333 | 0.211 | 0.329 |
| | Avg | 0.024 | **0.083** | 0.035 | 0.105 | 0.028 | 0.093 | 0.083 | 0.192 | 0.021 | 0.084 | 0.022 | 0.088 | 0.029 | 0.102 | 0.208 | 0.327 | 0.046 | 0.151 | 0.096 | 0.208 | 0.101 | 0.215 | 0.026 | 0.099 | 0.029 | 0.105 | 0.156 | 0.292 | 0.157 | 0.280 |
| ECL | 12.5% | **0.051** | **0.148** | 0.077 | 0.174 | 0.047 | 0.145 | 0.073 | 0.190 | 0.080 | 0.194 | 0.085 | 0.202 | 0.055 | 0.160 | 0.196 | 0.321 | 0.102 | 0.229 | 0.092 | 0.214 | 0.107 | 0.237 | 0.093 | 0.210 | 0.089 | 0.210 | 0.218 | 0.326 | 0.190 | 0.308 |
| | 25% | **0.061** | **0.163** | 0.087 | 0.184 | 0.055 | 0.156 | 0.090 | 0.159 | 0.087 | 0.203 | 0.089 | 0.206 | 0.065 | 0.175 | 0.207 | 0.332 | 0.121 | 0.252 | 0.118 | 0.247 | 0.120 | 0.251 | 0.097 | 0.214 | 0.096 | 0.220 | 0.219 | 0.326 | 0.197 | 0.312 |
| | 37.5% | **0.074** | **0.181** | 0.101 | 0.199 | 0.064 | 0.169 | 0.107 | 0.234 | 0.094 | 0.211 | 0.094 | 0.213 | 0.076 | 0.189 | 0.219 | 0.344 | 0.141 | 0.273 | 0.144 | 0.276 | 0.136 | 0.266 | 0.102 | 0.220 | 0.104 | 0.229 | 0.222 | 0.328 | 0.203 | 0.315 |
| | 50% | **0.090** | **0.202** | 0.121 | 0.219 | 0.079 | 0.185 | 0.127 | 0.257 | 0.101 | 0.220 | 0.100 | 0.221 | 0.091 | 0.208 | 0.235 | 0.357 | 0.160 | 0.293 | 0.175 | 0.305 | 0.158 | 0.284 | 0.108 | 0.228 | 0.113 | 0.239 | 0.228 | 0.331 | 0.210 | 0.319 |
| | Avg | **0.069** | **0.174** | 0.097 | 0.194 | 0.061 | 0.164 | 0.099 | 0.224 | 0.090 | 0.207 | 0.092 | 0.210 | 0.072 | 0.183 | 0.214 | 0.339 | 0.131 | 0.262 | 0.132 | 0.260 | 0.130 | 0.259 | 0.100 | 0.218 | 0.101 | 0.225 | 0.222 | 0.328 | 0.200 | 0.313 |
| Weather | 12.5% | **0.029** | **0.033** | 0.107 | 0.168 | 0.030 | 0.054 | 0.040 | 0.090 | 0.026 | 0.049 | 0.025 | 0.045 | 0.057 | 0.141 | 0.047 | 0.101 | 0.039 | 0.084 | 0.041 | 0.107 | 0.027 | 0.051 | 0.026 | 0.047 | 0.037 | 0.093 | 0.031 | 0.076 | | |
| | 25% | 0.031 | **0.033** | 0.108 | 0.167 | 0.029 | 0.043 | 0.047 | 0.108 | 0.028 | 0.052 | 0.029 | 0.052 | 0.031 | 0.053 | 0.065 | 0.155 | 0.052 | 0.111 | 0.048 | 0.103 | 0.064 | 0.163 | 0.029 | 0.056 | 0.030 | 0.054 | 0.042 | 0.100 | 0.035 | 0.082 |
| | 37.5% | 0.034 | **0.037** | 0.108 | 0.167 | 0.032 | 0.047 | 0.055 | 0.121 | 0.033 | 0.060 | **0.031** | 0.057 | 0.035 | 0.058 | 0.081 | 0.180 | 0.057 | 0.117 | 0.057 | 0.117 | 0.107 | 0.229 | 0.033 | 0.062 | 0.032 | 0.060 | 0.049 | 0.111 | 0.040 | 0.091 |
| | 50% | 0.031 | 0.041 | 0.109 | 0.168 | 0.035 | 0.051 | 0.070 | 0.145 | 0.037 | 0.065 | 0.034 | 0.062 | 0.038 | 0.063 | 0.102 | 0.207 | 0.065 | 0.133 | 0.066 | 0.134 | 0.183 | 0.312 | 0.037 | 0.068 | 0.037 | 0.067 | 0.053 | 0.114 | 0.046 | 0.099 |
| | Avg | **0.031** | **0.036** | 0.108 | 0.168 | 0.031 | 0.049 | 0.053 | 0.116 | 0.031 | 0.056 | **0.030** | 0.054 | 0.060 | 0.144 | 0.076 | 0.171 | 0.055 | 0.117 | 0.052 | 0.110 | 0.099 | 0.203 | 0.032 | 0.059 | 0.031 | 0.057 | 0.045 | 0.104 | 0.038 | 0.087 |

## D.2 Imputation

### D.2.1 Full Imputation Results

The imputation task results, presented in Table D.1.5, demonstrate TimeDiT's superior performance across various datasets and missing data ratios. All baseline models are trained in a full-shot setting, while TimeDiT leverages a pre-trained foundation model, fine-tuning it on realistic datasets. TimeDiT

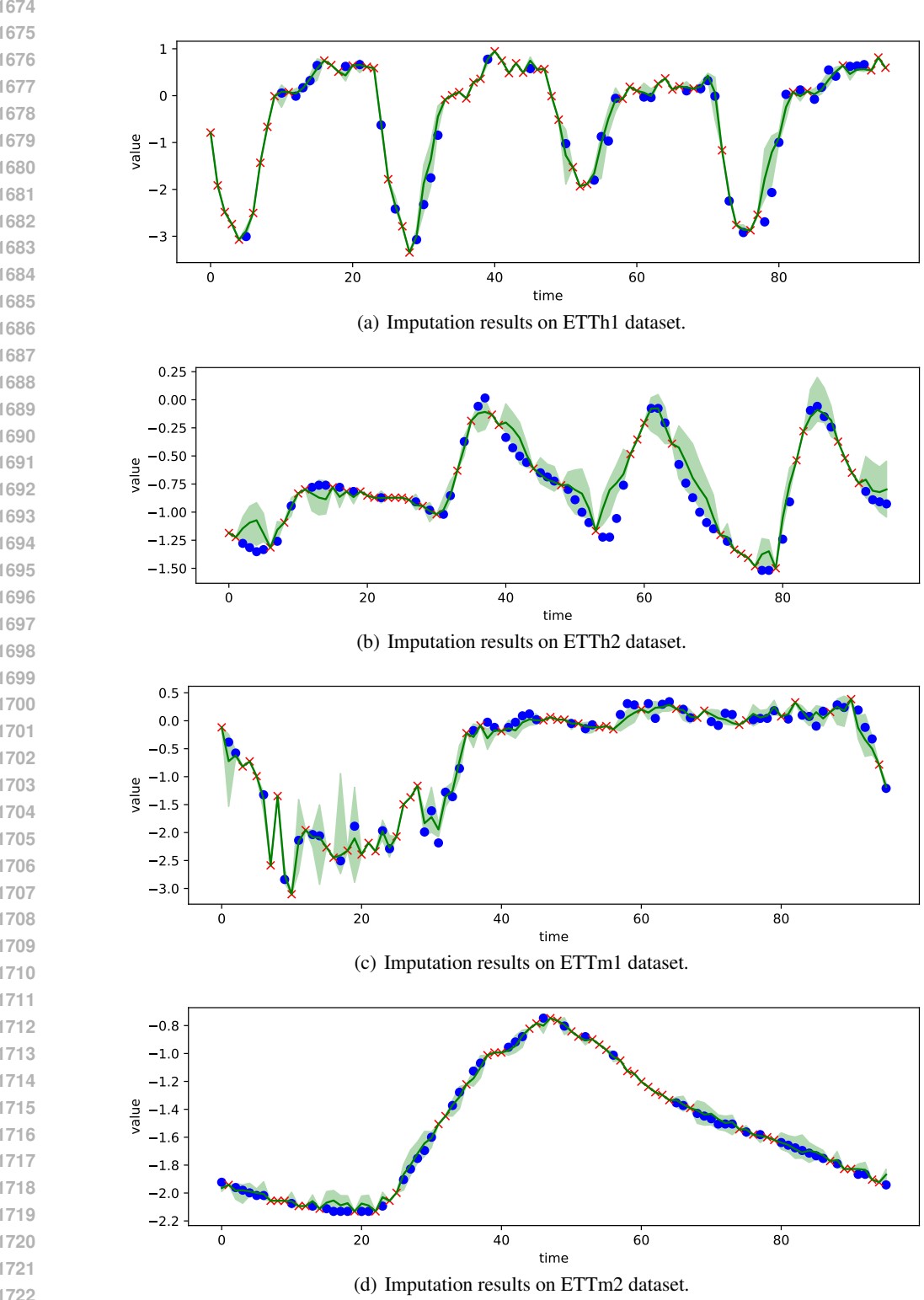

(a) Imputation results on ETTh1 dataset.

(b) Imputation results on ETTh2 dataset.

(c) Imputation results on ETTm1 dataset.

(d) Imputation results on ETTm2 dataset.

Figure 5: Visualization of imputation task on ETT datasets. This figure illustrates TimeDiT's performance, with red ×'s marking observed values, blue dots showing ground truth points for interpolation, a green line representing TimeDiT's mean of interpolation, and green shading indicating its estimated uncertainty intervals.

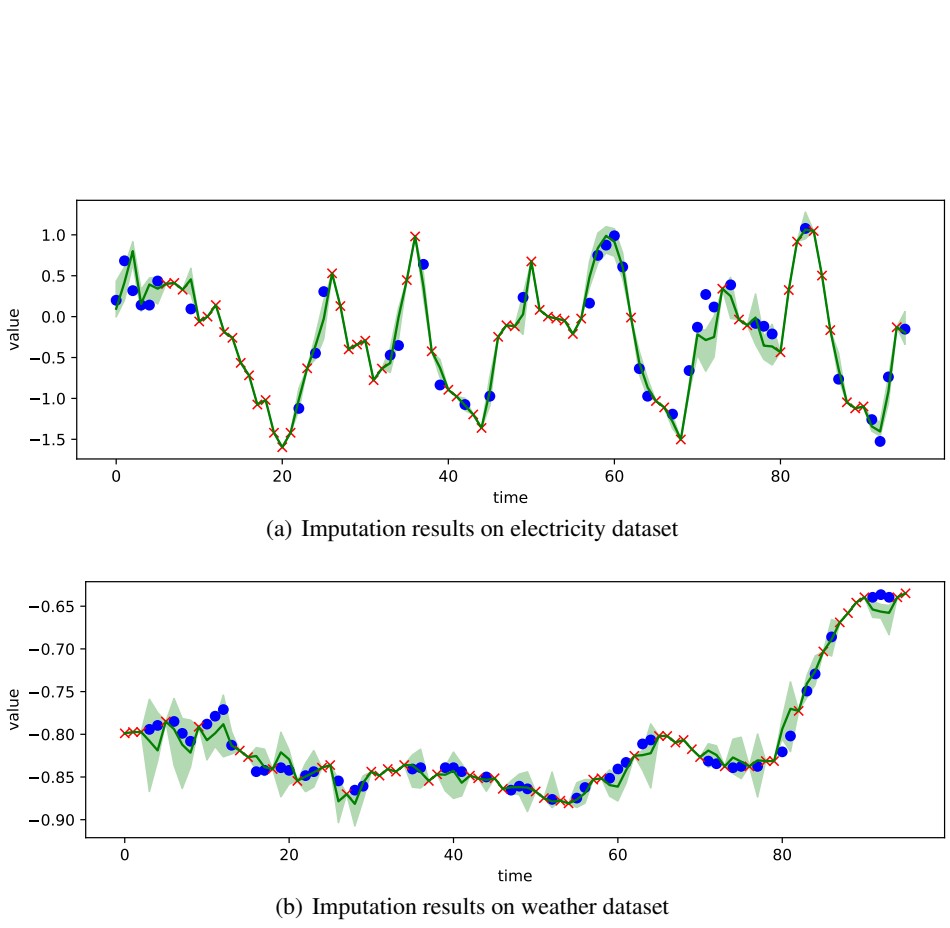

(a) Imputation results on electricity dataset

(b) Imputation results on weather dataset

Figure 6: Visualization of imputation task on electricity and weather datasets. This figure illustrates TimeDiT's performance, with red ×'s marking observed values, blue dots showing ground truth points for interpolation, a green line representing TimeDiT's mean of interpolation, and green shading indicating its estimated uncertainty intervals.

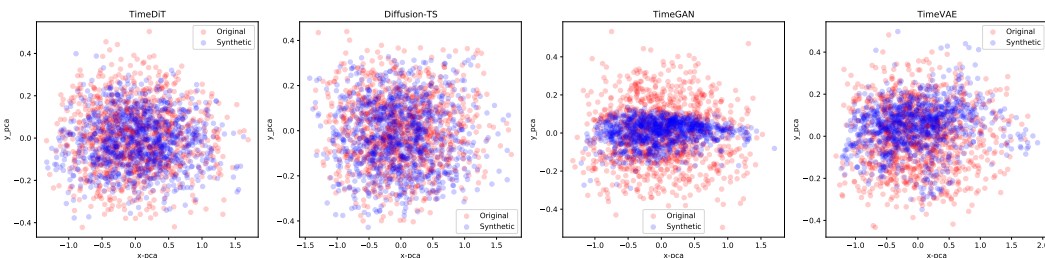

Figure 7: PCA Evaluation of Synthetic TSD from TimeDiT and other baselines on the sine dataset.

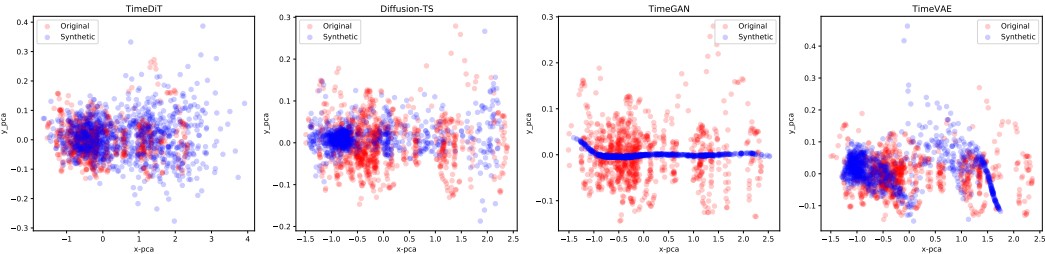

Figure 8: PCA Evaluation of Synthetic TSD from TimeDiT and other baselines on the stock dataset.

consistently achieves the lowest Mean Squared Error (MSE) and Mean Absolute Error (MAE) scores in most scenarios, outperforming state-of-the-art models such as GPT2, TimesNet, and PatchTST. Notably, TimeDiT's performance remains robust even as the proportion of missing data increases from 12.5% to 50%, showcasing its ability to handle substantial data gaps effectively. The model's imputation accuracy is particularly impressive for the ETTh1, ETTh2, ETTm1, and ETTm2 datasets, where it maintains a significant lead over other methods. imeDiT demonstrates superior performance on most datasets, achieving significant improvements over Timer, TimeMixer, and iTransformer, particularly on ETT datasets where we see reductions in MSE by up to 60%. TimeDiT maintains strong overall performance while offering greater versatility

### D.2.2 IMPUTATION VISUALIZATION

For visual representation of TimeDiT's imputation capabilities, we have plotted the results in Figure 5 and Figure 6, which clearly illustrates the model's accuracy in reconstructing missing data points across different datasets and missing data ratios.

### D.3 SYNTHETIC GENERATION

### D.3.1 SYNTHETIC GENERATION VISUALIZATION

We use 80% of all data for training and evaluation of the same data. For the air quality dataset, previous methods did not carefully use the -200 values as a placeholder for missing values. In our experiment, we masked all the -200 values for TimeDiT and baselines that support masks. For baselines that do not support mask, we replace -200 with the mean value. Minmax scaler is used for all models. Figure 7, 8,9,10 shows the PCA plots for all datasets and baselines. The visual comparison also validates the superiority of TimeDiT.

### D.3.2 LIMITED SYNTHETIC GENERATION

We also run the generation experiments with the limited data fine-tuning in Table 17. The generation experiments with limited data fine-tuning demonstrate TimeDiT's superior performance across various datasets and evaluation metrics. Comparing TimeGAN, TimeVAE, Diffusion-TS, and TimeDiT on sine, air, and energy datasets with 5% and 10% training data, TimeDiT consistently achieves the lowest Discriminative Scores, indicating its ability to generate the most realistic time series. In terms

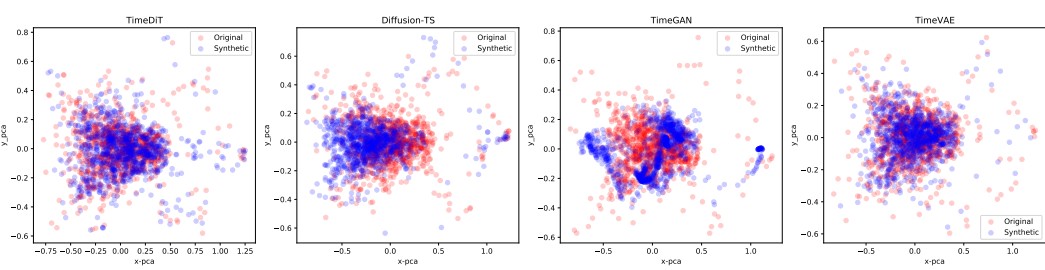

Figure 9: PCA plot for air quality dataset.

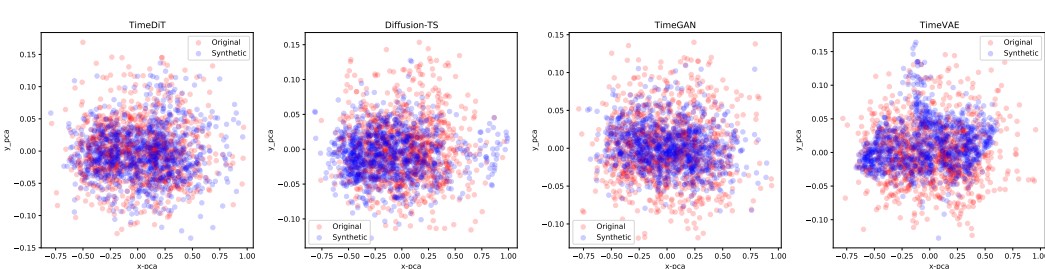

Figure 10: PCA plot for energy dataset.

of Predictive Scores, TimeDiT outperforms or matches other models, particularly excelling in the air dataset. Notably, TimeDiT's performance remains robust or improves when increasing from 5% to 10% training data, showcasing its effectiveness in data-scarce scenarios. These results highlight TimeDiT's capability to capture complex temporal patterns and generate high-quality time series data, even with limited training samples, making it a promising tool for various time series generation tasks.

### D.4 ANOMALY DETECTION

We conduct experiments on five real-world datasets from industrial applications: MSL, SMAP, SWaT, SMD, and PSM. The diffusion model, renowned for its proficiency in distribution learning, may inadvertently overfit by reconstructing anomalies alongside normal data points. To counteract this, we opted to bypass pretraining and introduced spectral residue (SR) transformation at the preprocessing stage of TimeDiT. This transformation helps to conceal points most likely to be anomalies and their immediate neighbors. The number of neighbors affected is controlled by the hyperparameter $n_{neighbor}$. The SR method utilizes Fourier Transformation to convert the original time series into a saliency map, thereby amplifying abnormal points, as detailed in (Ren et al., 2019; Zhao et al., 2020). Consistent with prior methodologies, we set the sequence length to be 100 identify anomalies using the 99th percentile of reconstruction errors. During evaluations, we apply standard anomaly adjustments as suggested by (Xu et al., 2018). As demonstrated in Table 5, TimeDiT outperforms baseline models on four of the five datasets. In particular, TimeDiT 23.03 points of improvement in terms of F1 score on the SMAP dataset compared to the previous best baseline. In addition, TimeDiT consistently outperforms both TimeMixer and iTransformer across all datasets, with particularly notable improvements on SMAP (95.91 vs 67.63/66.76) and SWAT (97.57 vs 88.84/92.63). These comprehensive comparisons against the latest models demonstrate TimeDiT's effectiveness as a unified framework for time series analysis, often achieving state-of-the-art performance while maintaining broader applicability across diverse tasks.

**Anomaly Detection Threshold** Our comprehensive analysis of threshold selection in Table 19 revealed that higher percentile thresholds, particularly the 99th and 99.5th percentiles, consistently yield superior performance. While we observed a systematic degradation in detection accuracy as threshold values decrease, we maintained the 99th percentile threshold to ensure fair comparison

Table 17: Limited observation data Synthetic Generation results on 24-length multivariate time series. Discriminative and predictive scores are calculated as described in (Yoon et al., 2019).

| Metric | Methods | 0.05 | | | 0.1 | | |
|---|---|---|---|---|---|---|---|
| | | Sine | Air Quality | Energy | Sine | Air Quality | Energy |
| Discriminative Score | TimeGAN | 0.120(0.043) | 0.500(0.003) | 0.500(0.000) | 0.067(0.028) | 0.492(0.003) | 0.500(0.000) |
| | TimeVAE | 0.220(0.224) | 0.498(0.001) | 0.500(0.000) | 0.499(0.002) | 0.495(0.002) | 0.499(0.001) |
| | Diffusion-TS | 0.037(0.013) | 0.496(0.003) | 0.498(0.005) | 0.031(0.012) | 0.494(0.001) | 0.494(0.011) |
| | TimeDiT | **0.031(0.007)** | **0.456(0.003)** | **0.472(0.000)** | **0.030(0.009)** | **0.437(0.004)** | **0.447(0.002)** |
| Predictive Score | TimeGAN | 0.231(0.007) | 0.148(0.029) | 0.308(0.006) | 0.200(0.002) | 0.130(0.029) | 0.302(0.004) |
| | TimeVAE | 0.251(0.003) | 0.328(0.008) | **0.296(0.001)** | 0.238(0.002) | 0.308(0.014) | **0.288(0.001)** |
| | Diffusion-TS | 0.196(0.003) | 0.111(0.004) | 0.333(0.018) | 0.188(0.001) | 0.102(0.010) | 0.340(0.019) |
| | TimeDiT | **0.194(0.001)** | **0.089(0.005)** | 0.335(0.008) | **0.192(0.000)** | **0.070(0.007)** | 0.318(0.005) |

with existing methodologies. This decision reflects our commitment to methodological rigor, as optimizing threshold values based on test set performance would introduce bias in the comparative analysis. Our approach prioritizes consistent experimental conditions across all evaluated methods, enabling meaningful benchmark comparisons while acknowledging the impact of threshold selection on detection performance.

**Spectral Residue processing for Anomaly Detection.** The SR Transformation involves the following equations. Table D.4 shows the full anomaly detection results.

$$A(f) = \text{Amplitude}(F(x)) \tag{36}$$

$$P(f) = \text{Phase}(F(x)) \tag{37}$$

$$L(f) = \log(A(f)) \tag{38}$$

$$AL(f) = h_q(f) \cdot L(f) \tag{39}$$

$$R(f) = L(f) - AL(f) \tag{40}$$

$$S(x) = F^{-1}(\exp(R(f) + iP(f))) \tag{41}$$

# E ANALYSIS ON TIMEDIT

## E.1 ABLATION STUDY

Our comprehensive ablation studies, detailed in Sections E1, E2, and E3, systematically evaluate TimeDiT's architectural choices. In Section E1, with particular emphasis on the Transformer design strategy, we explore TimeDiT's temporal-wise attention mechanism and compare it against alternative approaches, including channel-wise attention and dual attention mechanisms (as discussed in (Yu et al., 2024)). The analysis demonstrates that temporal-wise processing significantly outperforms

Table 18: Anomaly Detection result on 100-length multivariate time series. We calculate Precision, Recall, and F1 score as % for each dataset. '.' notation in model name stands for transformer. **Bold** indicates best result, Underline indicates the second best result. We replace the joint criterion in Anomaly Transformer with reconstruction error for fair comparison.

| Methods | MSL | | | SMAP | | | SWaT | | | SMD | | | PSM | | | 1st Pl |
|---|---|---|---|---|---|---|---|---|---|---|---|---|---|---|---|---|
| Metrics | P | R | F1 | P | R | F1 | P | R | F1 | P | R | F1 | P | R | F1 | Count |
| TimeDiT | **91.54** | 87.23 | **89.33** | **93.35** | **98.61** | **95.91** | **93.64** | **99.46** | **96.46** | 78.83 | **88.26** | 83.28 | 97.36 | **97.79** | **97.57** | 11 |
| GPT(6) | 82.00 | 82.91 | 82.45 | 90.60 | 60.95 | 72.88 | 92.20 | 96.34 | 94.23 | 88.89 | 84.98 | **86.89** | 98.62 | 95.68 | 97.13 | 1 |
| TimesNet | 89.54 | 75.36 | 81.84 | 90.14 | 56.40 | 69.39 | 90.75 | 95.40 | 93.02 | 87.91 | 81.54 | 84.61 | 98.51 | 96.20 | 97.34 | 0 |
| PatchTST | 88.34 | 70.96 | 78.70 | 90.64 | 55.46 | 68.82 | 91.10 | 80.94 | 85.72 | 87.26 | 82.14 | 84.62 | 98.84 | 93.47 | 96.08 | 0 |
| ETSformer | 85.13 | 84.93 | 85.03 | 92.25 | 55.75 | 69.50 | 90.02 | 80.36 | 84.91 | 87.44 | 79.23 | 83.13 | **99.31** | 85.28 | 91.76 | 1 |
| FEDformer | 77.14 | 80.07 | 78.57 | 90.47 | 58.10 | 70.76 | 90.17 | 96.42 | 93.19 | 87.95 | 82.39 | 85.08 | 97.31 | 97.16 | 97.23 | 0 |
| LightTS | 82.40 | 75.78 | 78.95 | 92.58 | 55.27 | 69.21 | 91.98 | 94.72 | 93.33 | 87.10 | 78.42 | 82.53 | 98.37 | 95.97 | 97.15 | 0 |
| DLinear | 84.34 | 85.42 | 84.88 | 92.32 | 55.41 | 69.26 | 80.91 | 95.30 | 87.52 | 83.62 | 71.52 | 77.10 | 98.28 | 89.26 | 93.55 | 0 |
| Autoformer | 77.27 | 80.92 | 79.05 | 90.40 | 58.62 | 71.12 | 89.85 | 95.81 | 92.74 | 88.06 | 82.35 | 85.11 | 99.08 | 88.15 | 93.29 | 0 |
| Anomaly. | 79.61 | **87.37** | 83.31 | 91.85 | 58.11 | 71.18 | 72.51 | 97.32 | 83.10 | **88.91** | 82.23 | 85.49 | 68.35 | 94.72 | 79.40 | 2 |
| TimeMixer | 89.72 | 75.42 | 81.95 | 89.51 | 54.34 | 67.63 | 91.56 | 86.28 | 88.84 | 86.60 | 71.50 | 78.33 | 99.18 | 87.74 | 93.11 | 0 |
| iTransformer | 86.16 | 62.64 | 72.54 | 90.69 | 52.82 | 66.76 | 92.21 | 93.06 | 92.63 | 86.92 | 77.75 | 82.08 | 97.98 | 92.81 | 95.32 | 0 |

Table 19: Threshold Sensitivity Analysis on Anomaly Detection Performance evaluated on F1 score

| Threshold | 99.5 | 99 | 98 | 97 | 96 | 95 |
|-----------|------|------|------|------|------|------|
| MSL | 83.9 | 89.33 | **90.1** | 88.17 | 85.28 | 82.84 |
| PSM | 96.32 | **97.57** | 96.78 | 95.72 | 94.66 | 93.61 |
| SMAP | **97.08** | 95.91 | 93.23 | 90.33 | 87.64 | 85.09 |
| SMD | **83.28** | 82.07 | 76.61 | 70.73 | 65.71 | 61.24 |
| SWAT | **97.6** | 96.46 | 93.49 | 90.74 | 88.0 | 85.42 |

Table 20: Ablation Study on the Model Design Space.

| Dataset | TimeDiT | Physics-Informed | Dual-attention | Channel-wise | Patch Token |
|---------|---------|------------------|----------------|--------------|-------------|
| Solar | 0.457(0.002) | 0.452(0.001) | 0.467(0.002) | 0.461(0.003) | 0.874(0.010) |
| Electricity | 0.026(0.001) | 0.024(0.000) | 0.029(0.001) | 0.028(0.000) | 0.105(0.013) |
| Traffic | 0.185(0.010) | 0.153(0.005) | 0.187(0.007) | 0.164(0.006) | 0.258(0.021) |

traditional patch-based tokenization approaches, achieving substantially lower error rates (0.457 versus 0.874 on Solar dataset).

This performance disparity can be attributed to two key factors: First, while channel relationships exhibit model-specific variations, temporal patterns provide more universal characteristics across time series data, enabling better generalization. Second, patch-based approaches introduce additional hyperparameter dependencies (patch length and stride settings) that compromise the model's universal applicability. These findings validate our design choice of temporal-wise processing as a more robust and generalizable approach for time series modeling. The empirical results strongly support our architectural decisions, demonstrating that TimeDiT's temporal-focused design effectively captures universal temporal dynamics while maintaining model flexibility across diverse applications and domains. In addition, the Physics-Informed component yields consistent performance improvements across all datasets, with notable enhancements in Traffic (0.153 versus 0.185), Electricity (0.024 versus 0.026), and Solar (0.452 versus 0.457) predictions, underscoring the value of incorporating physical constraints during inference.

## E.2 HANDLING MISSING VALUES

Table 21: Mask mechanisms for TimeDiT, compared on the zero-shot forecasting task.

| Dataset | TimeDiT | w/o Random Mask | w/o Stride Mask | w/o Future Mask |
|---------|---------|-----------------|-----------------|-----------------|
| Solar | **0.457(0.002)** | 0.463(0.002) | 0.465(0.002) | 0.843(0.005) |
| Electricity | **0.026(0.001)** | 0.029(0.001) | 0.030(0.001) | 0.095(0.006) |
| Traffic | **0.185(0.010)** | 0.191(0.007) | 0.188(0.007) | 0.201(0.011) |

Our experimental design leverages naturally occurring missing values inherent in real-world datasets, primarily arising from irregular sampling rates and multi-resolution data collection processes. This approach authentically validates model robustness against genuine missing data patterns rather than artificially generated scenarios. TimeDiT incorporates a comprehensive masking strategy that aligns with three well-established missing data mechanisms: Missing Completely at Random (MCAR) using uniform distribution-based random masks, Missing at Random (MAR) employing block and stride masks to capture structured patterns and dependencies between non-contiguous observations, and Missing Not at Random (MNAR) utilizing reconstruction masks with physics-informed sampling for scenarios where missing patterns correlate with unobserved variables. These mechanisms are simultaneously applied through self-supervised learning, enabling robust representation learning without requiring explicit knowledge of the underlying missing data processes. Our comprehensive ablation studies in Table 21 demonstrate the criticality of each masking strategy, where the removal of any mask type leads to performance degradation, with future masks showing the most significant impact. These findings validate our integrated approach to handling diverse missing data scenarios in time-series analysis.

### E.3 CONDITION SCHEME FOR TIMEDIT

Table 22: Condition scheme for TimeDiT, compared on the zero-shot forecasting task.

| Dataset | AdaLN | Additive | Cross-attention | Token concatenation |
|---|---|---|---|---|
| Solar | **0.457(0.002)** | 0.671(0.002) | 0.721(0.002) | 0.463(0.001) |
| Electricity | **0.026(0.001)** | 0.068(0.004) | 0.079(0.003) | 0.041(0.003) |
| Traffic | **0.185(0.010)** | 0.224(0.001) | 0.216(0.000) | 0.188(0.008) |

As mentioned in Section 4.2, AdaLN's superior performance stems from its ability to dynamically adjust feature distributions across different layers while maintaining computational efficiency. This approach aligns well with the inherent nature of time series data, where temporal dependencies typically exhibit gradual rather than dramatic changes in both seen and unseen time steps. We conducted comparative experiments to evaluate different conditioning mechanisms in TimeDiT:

- Additive conditioning, which adds conditional information directly to the diffusion input;

- Cross-attention, which uses conditional time series as keys/values and noisy time series as queries to fuse conditional information;

- Token concatenation, which concatenates conditional time series with noisy time series at the input level before TimeDiT processing.

The experimental results (Table E.3) across Solar, Electricity, and Traffic datasets consistently show that AdaLN achieves superior performance compared to the next best alternative. This significant performance gap validates our choice of AdaLN as TimeDiT's primary conditioning mechanism.

### E.4 NOISE EMBEDDING JUSTIFICATION

Table 23: Results in predicting the input of TimeDiT, compared on the zero-shot forecasting task.

| Dataset | TimeDiT | Predict the input |
|---|---|---|
| Solar | 0.457(0.002) | 0.462(0.003) |
| Electricity | 0.026(0.001) | 0.037(0.002) |
| Traffic | 0.185(0.010) | 0.199(0.007) |

TimeDiT's noise embedding approach plays multiple key roles in the diffusion modeling framework. The diffusion process operates directly in a continuous embedding space, allowing for smoother transitions between noise levels and better preserving the inherent time dependence, thus enabling the model to learn a more robust representation of the underlying time series structure. This approach has several technical advantages (Ho et al., 2020; Peebles & Xie, 2022; Lu et al., 2024): the embedding space provides a continuous representation in which the diffusion process can operate more efficiently. The direct embedding of noisy samples helps prevent the embedding space from collapsing during training. From a practical point of view, this approach allows for parallel processing of multiple time steps, handles varying degrees of noise through a unified framework, and makes the diffusion process more stable compared to traditional generation methods. In addition, the embedded noise representation allows for the seamless incorporation of physical constraints and maintains temporal continuity while progressively denoising, thus contributing to a better quantification of the uncertainty in the generated samples. Direct prediction of the input is also an option available, and we added new experiments as shown in Table 23. This also demonstrates the advantages of reconstructive noise.

### E.5 CHANNEL NUMBERS

imeDiT implements an adaptive architectural framework for processing variable-dimensional inputs through a sophisticated channel management system. The architecture employs a predefined maximum channel parameter $K_{max}$, where inputs with fewer channels ($k < K_{max}$) undergo appropriate padding, while those exceeding $K_{max}$ are automatically segmented into $\lceil k/K_{max} \rceil$ blocks of $K_{max}$ channels for independent processing. Based on comprehensive analysis across diverse multivariate time series datasets, we established $K_{max} = 40$ as an optimal parameter that balances computational efficiency with model performance across various domains. Empirical evaluations in Table 24 demonstrate that while performance significantly degrades with limited channels ($k \leq 10$),

Table 24: Difference channel number's influence on the zero-shot performance.

| Channel Number | 10 | 20 | 30 | 40 | 50 |
|---|---|---|---|---|---|
| Solar | 0.471(0.002) | 0.462(0.001) | 0.459(0.002) | 0.457(0.002) | 0.458(0.002) |
| Electricity | 0.030(0.001) | 0.029(0.002) | 0.028(0.001) | 0.026(0.001) | 0.027(0.001) |
| Traffic | 0.192(0.008) | 0.183(0.007) | 0.177(0.007) | 0.185(0.010) | 0.165(0.006) |

the model maintains robust performance across larger channel configurations, indicating the architecture's effectiveness in handling diverse multivariate scenarios without compromising computational efficiency.

### E.6 SAMPLING STEPS

Table 25: CRPS and CRPS_sum for Solar and Traffic datasets with different sampling steps.

|  | 50 | 100 | 150 | 200 | 250 | 300 | 350 | 400 | 450 | 500 |
|---|---|---|---|---|---|---|---|---|---|---|
| Solar (CRPS) | 0.440 | 0.443 | 0.439 | 0.430 | 0.435 | 0.431 | 0.430 | 0.431 | **0.428** | 0.434 |
| Solar (CRPS_sum) | 0.427 | 0.430 | 0.425 | 0.418 | 0.422 | 0.418 | **0.410** | 0.414 | 0.409 | 0.419 |
| Traffic (CRPS) | 0.425 | 0.369 | 0.350 | 0.342 | 0.330 | 0.330 | 0.334 | 0.330 | 0.328 | **0.327** |
| Traffic (CRPS_sum) | **0.135** | 0.136 | 0.141 | 0.138 | 0.140 | 0.140 | 0.138 | 0.142 | 0.141 | 0.141 |

To further understand TimeDiT's behavior and optimize its performance, we conducted additional experiments on the impact of sampling steps. These experiments are crucial as they reveal the model's sensitivity to this hyperparameter and its implications for different datasets and evaluation metrics. For the Solar dataset, increasing the number of sampling steps generally improves performance, with the best CRPS achieved at 450 steps and the best CRPS_sum at 350 steps. The Traffic dataset shows a different trend: CRPS improves with more sampling steps, reaching its best at 500 steps, while CRPS_sum achieves its optimum at the lowest sampling step of 50. These results suggest that the optimal number of sampling steps is dataset-dependent and can differ based on the chosen metric. The variation in performance across sampling steps is relatively small, indicating that TimeDiT is robust to this hyperparameter within the tested range. However, the trade-off between computational cost and marginal performance gains should be considered when selecting the number of sampling steps for practical applications.

### E.7 FAILURE SCENARIOS ANALYSIS

TimeDiT's performance shows notable degradation in three key scenarios: highly irregular sampling rates deviating from training distributions, complex non-stationary patterns underrepresented in pretraining data, and domain-specific patterns requiring expert knowledge beyond general time series characteristics. As shown by SMD dataset for anomaly detection (Table 5) where it achieves 83.28% F1 score versus GPT2's 86.89%. This dataset represents cloud server machine metrics with high-frequency sampling and complex feature interdependencies. Additionally, when dealing with extremely short-term patterns or highly localized anomalies, specialized architectures like GPT2 that focus intensively on recent temporal context may outperform TimeDiT's more holistic approach, as our diffusion-based generation process may occasionally smooth over abrupt local changes. These limitations, primarily stemming from the model's dependence on learned foundational patterns, become particularly relevant in specialized industrial applications and unique financial scenarios. Understanding these boundaries is crucial for informed model deployment decisions and highlights promising directions for future research.

### E.8 DYNAMIC ON MODEL SIZE

The experimental results demonstrate a clear correlation between TimeDiT's model size and its imputation performance across different datasets and missing data ratios. As shown in Table 26, as the model size increases from Small (S) to Big (B) to Large (L), we observe consistent improvements in both averaged Mean Squared Error (MSE) and averaged Mean Absolute Error (MAE) metrics.

Table 26: Performance metrics for weather and ecl datasets on different model size.

| | | S | | B | | L | |
|---|---|---|---|---|---|---|---|
| | | MSE | MAE | MSE | MAE | MSE | MAE |
| Weather | 0.125 | 0.029 | 0.033 | 0.029 | 0.026 | 0.025 | 0.024 |
| | 0.250 | 0.031 | 0.033 | 0.033 | 0.029 | 0.028 | 0.027 |
| | 0.375 | 0.034 | 0.037 | 0.036 | 0.033 | 0.031 | 0.031 |
| | 0.500 | 0.031 | 0.041 | 0.042 | 0.039 | 0.036 | 0.036 |
| | **Avg** | 0.031 | 0.036 | 0.035 | 0.032 | 0.030 | 0.029 |
| ECL | 0.125 | 0.051 | 0.148 | 0.050 | 0.144 | 0.048 | 0.140 |
| | 0.250 | 0.061 | 0.163 | 0.060 | 0.158 | 0.058 | 0.154 |
| | 0.375 | 0.074 | 0.181 | 0.071 | 0.175 | 0.069 | 0.170 |
| | 0.500 | 0.090 | 0.202 | 0.087 | 0.197 | 0.084 | 0.190 |
| | **Avg** | 0.069 | 0.174 | 0.067 | 0.169 | 0.065 | 0.163 |

The Large model consistently outperforms the Small and Big variants across all scenarios, with the most significant gains observed in the weather dataset. Notably, larger models (B and L) show better resilience to increased proportions of missing data compared to the Small model. The improvement is more pronounced for the weather dataset than for the ecl dataset, suggesting that the benefits of increased model size may vary depending on the nature and complexity of the time series data. The consistent performance gains from S to B to L models indicate that TimeDiT's architecture scales well with increased model size. These findings suggest that increasing TimeDiT's model size is an effective strategy for improving imputation accuracy, particularly for complex datasets or scenarios with higher proportions of missing data. However, the performance may remain relatively consistent across all model sizes for both the weather and ecl datasets, even as the proportion of missing data increases from 12.5% to 50%. This stability in performance suggests that TimeDiT's architecture may achieve its optimal capacity for these imputation tasks even at smaller model sizes. Thus, the trade-off between computational resources and performance gains should be considered when selecting the appropriate model size for specific applications.

### E.9 LEARNED REPRESENTATION

We randomly sampled 4000 training samples from each of the Solar and Traffic datasets and got their representation from the foundation model with and without textual condition, which is the zero-shot setting. To visualize the distribution of these datasets, we employ t-SNE dimensionality reduction. As depicted in Figure 11, the t-SNE plot clearly distinguishes between the Solar and Traffic datasets, highlighting their unique characteristics. The Solar dataset samples form a distinct cluster, likely reflecting the periodic patterns and seasonal variations inherent in solar power generation. In contrast, the Traffic dataset samples create a separate cluster, capturing the complex temporal dynamics of traffic flow, which may include daily commute patterns and irregular events. This clear separation in the t-SNE visualization underscores the fundamental differences in the underlying structures and patterns of these two time series datasets. Such distinction is crucial for understanding the diverse nature of temporal data and highlights the importance of developing versatile models like TimeDiT that can effectively capture and generate a wide range of time series patterns.

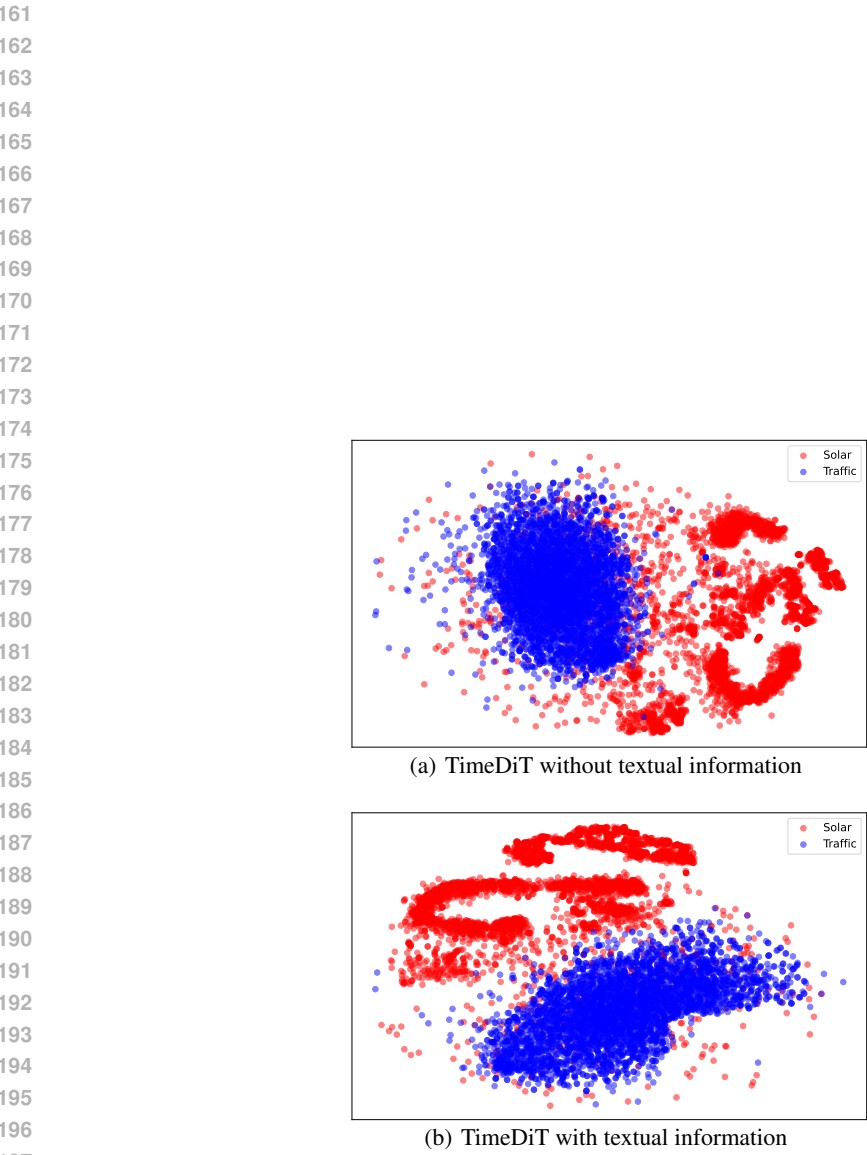

(a) TimeDiT without textual information

(b) TimeDiT with textual information

Figure 11: Repreasentation Visualization of TimeDiT when the input is two new datasets and it can modeling them separately with the capability.

