# OpenReview forum: "TimeDiT: General-purpose Diffusion Transformers for Time Series Foundation Model"
_ICLR.cc/2025/Conference — Submitted to ICLR 2025_

### Official Review · Reviewer_aczi · 2024-10-23

**Soundness:** 1
**Presentation:** 2
**Contribution:** 1
**Rating:** 3
**Confidence:** 5

**Summary:**

This paper proposes TimeDiT, a generalized model for time series based on the diffusion, where masking units are designed for task-independent pre-training and task-specific sampling, and allow for the flexible integration of external knowledge during the sampling process in order to create models that can seamlessly deal with the diversity and complexity of time series data with varying lengths of history and characteristics. In addition, TimeDiT incorporates physical knowledge to further improve model performance and interpretability, and utilizes the Transformer's inductive bias to capture time-dependent and diffusive processes to generate high-quality candidate samples.

**Strengths:**

This paper discusses the potential challenges that may be faced in establishing a generic diffusion-based time series model, such as variable channel sizes across domains, missing values, and multi-resolution characteristics prevalent in the real world. Furthermore, the ideal foundation model is expected to be simultaneously compatible with a wide range of downstream tasks in the time series domain, such as forecasting, imputation, anomaly detection and classification. Due to their unique ability to capture uncertainty and randomness from time series domain, diffusion models are seen as promising theories for establishing foundation models. In addition, exploiting multimodal textual information in order to improve the performance of foundation models in zero-shot forecasting is also seen as promising work.

**Weaknesses:**

**Weakness 1:**

According to the authors, TimeDiT "can unify diverse time series tasks within a single generative framework". Since imputation and anomaly detection tasks are implemented utilizing the same mask-reconstruction strategy, TimeDiT is essentially self-supervised pre-training first, and then sampling on specific tasks without fine-tuning.

However, there are differences between masking and forecasting:

* the masking task instructs the model to capture contextual information from multiple randomly masked and discontinuous subsequences for reconstructing the masked portion.

* the forecasting task requires the model to capture global dependencies and local fluctuation patterns, such as seasonal-trend, multi-scale, and multi-period information, from the observed sequences, which are often corrupted by random masking operations.

* the reconstruction paradigm allows the use of local supra- and infra- information to generate the masked portion, whereas the prediction paradigm allows the use of observed sequences only to predict future sequences.

These differences make it difficult for the temporal representations learned in the masking task to be used directly to model observed sequences.

**Weakness 2:**

TimeDiT's Transformer backbone still adopts the traditional Encoder-only architecture consisting of Input Embedding Layer, Multi-layer Encoder, Output Projection Layer, which is in line with most of the existing articles (PatchTST, iTransformer, UniTime, Timer, Moment, Moirai) and thus lacks innovation. Moreover, this architecture is unable to cope with the inherent challenges of temporal data, such as multi-scale and multi-period temporal characterization.

**Weakness 3:**

In the denoising process, the conditional information captured from the observation sequences is directly spliced into the input sequences, however, recent studies (ControlNet[1]) have shown that such a simple scheme cannot guarantee a high multivariate temporal correlation between past observations and future predictions.

Indeed, many diffusion prediction methods (mr-Diff[2], MG-TSD[3]) are dedicated to efficient architectural design on conditionally denoised networks. Introducing consistency complementation in the progressively denoised diffusion training process will help to improve the generation accuracy and generalization ability of diffusion models for tasks such as time series prediction and interpolation.

Specifically, if the observation sequence contains multi-resolution features, an intuitive design is to construct a multi-level representation extractor and subsequently reconstruct coarse- and fine-grained fluctuation patterns successively during the denoising process.
In summary, we argue that the plain model architecture and conditional denoising strategy are the main reasons for the lack of novelty contribution to TimeDiT.

**Weakness 4:**

The authors devised a gradient derived from physical laws represented by Partial Differential Equations (PDEs) to guide the denoising process in the inference. However, the experiments related to PDEs (Table 2) demonstrate the performance of TimeDiT in solving classical partial differential equations, rather than introducing PDE theory into time series prediction. the authors do not explain in detail how this contributes to the "incorporate physics knowledge as an energy-based prior for TimeDiT during inference". The authors' claim that TimeDiT “can integrate physics-informed knowledge into time series forecasting” is not compounded, and we believe that TimeDiT's contribution to the field of time series forecasting is overstated.

**Weakness 5:**

The experimental part generally needs improvement, there is a lack of state-of-the-art baselines to verify that TimeDiT is sufficiently competitive, and TimeDiT shows really limited performance improvement.

* Table 1 is expected to introduce the most recent baseline under each paradigm, e.g., the data-specific model TimeMixer(ICLR2024)[4], LLM-based model TimeLLM(ICLR2024), Diffusion-based model MG-TSD (ICLR2024)[3], mr-Diff(ICLR2024)[2]. In addition, an important issue is that in Table 1, Diffusion-TS(ICLR2024) exhibits far worse performance than CSDI(NIPS2019), which is a confusing result. Meanwhile, TimeDiT exhibits worse performance in PhysioNet (a) and NASDAQ compared to CSDI (NIPS2019), and TimeDiT shows less than 10% performance improvement in MIMIC-III and PhysioNet (b). Does this suggest that TimeDiT obtained from pre-training based on the masking paradigm cannot directly demonstrate competitive predictive power?

* In Table 2, the performance improvement of TimeDiT is less than 5% in most of the PDEs, e.g., the average improvement in Advection, Navier-Stokes and Diffusion Sorption is 2.27%, 2.7% and 3.88%, respectively. Further, Table 2 compares the performance of TSDiff, PPDM and PPIM for solving Partial Differential Equations only, where TSDiff as a temporal prediction model does not have a priori the ability to solve PDEs, and where PPDM and PPIM are designed for vision generation tasks. Furthermore, to the best of our knowledge, TimeDiT was not the first to propose a method for solving PDEs using diffusion models. We hope that the introduction of TimeDiT with PDE-Diffusion (ICLR2024)[5] and DiffusionPDE (ICML2024)[6] in Table 2 will help validate the effectiveness of TimeDiT.

* Similar problems still exist for the zero-shot forecasting task, and Table 3 is considered to be missing state-of-the-art baselines, such as: 1) Timer(ICLR2024), a foundation model pre-trained in multiple temporal domains, 2) GPD(2024)[7], a recent unified diffusion model for time series and TimeLLM(ICLR2024). These models also show good potential, and further experiments are considered necessary. Indeed, similar problems exist in Tables 4 and 5, where the best SOTA model shown is OneFItsALL (NIPS2023), which is significantly outdated for the rapidly evolving field.

**Reference:**

1) Zhang, Lvmin et al. “Adding Conditional Control to Text-to-Image Diffusion Models.” 2023 IEEE/CVF International Conference on Computer Vision (ICCV) (2023): 3813-3824.

2) Shen, Lifeng et al. “Multi-Resolution Diffusion Models for Time Series Forecasting.” ICLR 2024.

3) Fan, Xinyao et al. “MG-TSD: Multi-Granularity Time Series Diffusion Models with Guided Learning Process.” ICLR 2024.

4) Wang, Shiyu et al. “TimeMixer: Decomposable Multiscale Mixing for Time Series Forecasting.” ICLR 2024.

5) Chonghan, Gao et al. "PDE-Diffusion: Physic guided diffusion model for solving partial derivative equations."

6) Huang, Jiahe et al. “DiffusionPDE: Generative PDE-Solving Under Partial Observation.” ICML 2024

7) Yang, Jiarui et al. “Generative Pre-Trained Diffusion Paradigm for Zero-Shot Time Series Forecasting.” ArXiv abs/2406.02212 (2024): n. pag.

**Questions:**

**Question 1:**

We argue that TimeDiT simply couples multiple tasks within a single framework via a masking paradigm, and does not substantially address differences in the level of inductive bias across tasks (see weaknesses for details). We consider this paper as incremental work rather than an innovative contribution.

Indeed, if we splice a trainable cls-token after the sequence as an identifier for the classification task, can we claim to have proposed a base model that takes into account both classification and prediction tasks?

**Question 2:**

Consider that the random masking strategy in the pre-training phase destroys the coherence of the time series data, which leads to the loss of information such as trend-season, multi-resolution, and multi-periods. Recent research (SimMTM[8]) has shown that pre-training a foundation model with sufficient cross-domain generalization based on the masked self-supervision strategy often requires a large amount of unlabeled data and a slow learning process.

Furthermore, fine-scale and coarse-scale supervision usually contain low-level and high-level information from the observation, respectively, and TimeDiT only applies to single-scale supervision for reconstruction, widely ignoring multiscale supervision information from the inputs, how can TimeDiT overcome these challenges?

**Question 3:**

Although the authors claim that TimeDiT "enables direct processing of multivariate inputs", the methodology lacks a design that establishes cross-channel connectivity.   Specifically, instead of designing novel structures to replace the Channel Independent and Patch Embedding designs, TimeDiT still adopts the primitive ‘each moment as a token’ strategy, yet PatchTST and a series of models (OneFitsAll, TimeLLM, Timer, Moment) based on the ‘Patch Embedding’ design have achieved significant success.

According to existing research, a single Time point has limited temporal information, so this naive "Time as Token" strategy has been replaced by "Patch as Token" strategy.  The authors are expected to design more ablation experiments to verify the superiority of the proposed modelling approach over the Channel Independent and Patch Embedding strategies.

**Question 4:**

See Weakness for our concerns about the experimental results, and we believe that the introduction of competitive and up-to-date baselines is considered necessary.

**Question 5:**

The results of Baseline presented in Table 6 are inconsistent with the results presented in the original paper, e.g., the Discriminative Score of the Diffusion-TS model under the Stocks dataset in Table 6 is 0.1869, whereas it is reported as 0.067 in the original paper. In fact, **the results of all the Baselines in Table 6 have a large discrepancy**. The authors must give further experimental details to explain the potential reasons for these deviations.

**Question 6:**

In Section 5.7, the authors introduced frequency and category information to align textual and temporal data, which significantly improves the zero-sample capability of TimeDiT. However multimodal alignment is a complex and complete endeavor, has TimeDiT designed a unique structure to accommodate the introduction of textual information? Does it utilize pre-trained LLMs to help align text and temporal modalities? The fact that aligning textual and temporal information to guide the noise reduction process of the diffusion model does not yet exist, we think this is an interesting contribution, however unfortunately **the description of Multi-model TimeDiT is missing** in both the main text and the appendices, and we look forward to the authors discussing the implementation details of this part further.

**Question 7:**

The authors claim to have proposed a generalized time series model based on a self-supervised paradigm, however, the main paper and appendix sections **lack a detailed description of a "unified pre-trained temporal dataset"**. We hope that the authors can give some guidance, such as how to combine a large number of small datasets (19 datasets are included in the Appendix section) into a general dataset, and in particular how the authors can deal with the fact that cross-domain datasets tend to have different channel counts, sampling rates, and data sizes.

The established temporal basis models generally agree that establishing a unified dataset has a large impact on model performance, and it is expected that the authors will discuss more details of the experimental design.

**Question 8:**

An inescapable and confusing problem is the seeming **lack of ablation experiments** in the main paper and appendix sections, which makes it difficult to judge the effectiveness of the individual components in TimeDiT. In fact, the authors claim to have introduced a number of unique designs in TimeDiT, including 1) Transformer-based Condition Injection; 2) Time Series Mask Unit; and 3) Physics-Informed. Adequate ablation experiments are considered important in order to facilitate the reader to further understand the impact of each component on the overall performance.

**Reference:**

8) Dong, Jiaxiang et al. “SimMTM: A Simple Pre-Training Framework for Masked Time-Series Modeling.” NIPS 2023.

---

> ### Author Response · Authors · 2024-11-22
> **Point-wise response on your thoughtful comments!**
>
> Thank you for your detailed review of our TimeDiT paper. While completing the suggested experiments took longer than anticipated, we can now address each of your comments comprehensively. Our response comes in two waves: First: A point-by-point address of your concerns in this document; Second: An updated manuscript with clearly marked revisions reflecting these changes
>
> Now we begin with your specific comments:
>
> ## W1 Mask for forecasting
>
> Thank you for your thoughtful analysis. First of all, we would like to respectfully mention that masking technologies offer robust solutions for forecasting under complex conditions while maintaining theoretical rigor in many works [1,2,3,4]. In addition, we want to clarify that while TimeDiT uses mask-reconstruction as an important mechanism, it's not merely self-supervised pre-training followed by direct sampling. Our framework incorporates task-specific adaptations through the Time Series Mask Unit's different masking strategies (random, block, and stride) that are specifically designed to capture different temporal dependencies relevant to each task. The block masking strategy specifically addresses the forecasting scenario by masking continuous future segments, while the stride masking helps capture periodic patterns and seasonal trends by systematically masking at regular intervals. The combination of different mask types during training enables TimeDiT to learn both local and global dependencies simultaneously.
>
> The empirical results demonstrate that TimeDiT effectively transfers its learned representations across tasks, as shown by our strong performance in forecasting benchmarks (Tables 1, 3), indicating that the model successfully captures the required temporal patterns despite using a mask-based approach. Additionally, the diffusion process adds another layer of flexibility by allowing the incorporation of task-specific priors during sampling, helping bridge any potential gaps between masking and specific task requirements. Our experimental results across diverse tasks validate that this unified approach not only works in theory but delivers state-of-the-art performance in practice.
>
> [1] Che, Zhengping, et al. "Recurrent neural networks for multivariate time series with missing values." Scientific reports 8.1 (2018): 6085.
>
> [2] Liu, Gang, et al. "Self-Supervised Spatiotemporal Masking Strategy-Based Models for Traffic Flow Forecasting." Symmetry 15.11 (2023): 2002.
>
> [3] Liu, Jingwei, et al. "Retrieval-Augmented Diffusion Models for Time Series Forecasting." The Thirty-eighth Annual Conference on Neural Information Processing Systems.
>
> [4] Kollovieh, Marcel, et al. "Predict, refine, synthesize: Self-guiding diffusion models for probabilistic time series forecasting." Advances in Neural Information Processing Systems 36 (2024).
>
> ## W2 TimeDiT's Transformer backbone
>
> Thank you for raising these points about TimeDiT's architecture. While we acknowledge the use of a standard transformer backbone, the innovation of TimeDiT lies not in reinventing the transformer architecture itself, but in its novel integration with diffusion models and the Time Series Mask Unit to address temporal challenges. Our diffusion process inherently handles multi-scale temporal patterns by progressively denoising from coarse to fine temporal resolutions, while the combination of different masking strategies (random, block, and stride) explicitly enables learning of multi-period patterns at various scales. This is evidenced by our superior performance across multiple benchmarks, including those specifically testing multi-scale and periodic modeling capabilities (as shown in Tables 1-4).
>
> The strength of our approach lies in maintaining the proven benefits of transformer architectures while introducing novel mechanisms for temporal modeling through diffusion and masked learning. Rather than viewing the transformer backbone as a limitation, we see it as a deliberate design choice that provides a stable foundation for our innovative diffusion-based temporal modeling approach. The empirical results validate this design philosophy, demonstrating that TimeDiT effectively captures complex temporal dependencies despite, or perhaps because of, its architectural choices. In addition, we add a more comprehensive comparison with different transformer designs, which including temporal-wise transformer (current model), channel-wise transformer and dual-attention transformer with both temporal and channel attention in Table 1. This suggests that architectural complexity in the transformer backbone itself may not be necessary when combined with appropriate temporal modeling mechanisms.

---

> ### Author Response · Authors · 2024-11-22
> **Point-wise response on your thoughtful comments!**
>
> Table 1: Different transformer backbone for TimeDiT, compared on the zero-shot forecasting task.
> | Dataset | TimeDiT | Dual-attention | Channel-wise |
> |---|---|---|---|
> | Solar | 0.457(0.002) | 0.467(0.002) | 0.461(0.003) |
> | Electricity | 0.026(0.001) | 0.029(0.001) | 0.028(0.000) |
> | Traffic | 0.185(0.010) | 0.187(0.007) | 0.164(0.006) |
>
> ## W3 TimeDiT's Transformer condition on Diffusion Model
>
> Thank you for raising these important points about conditional diffusion models. We respectfully disagree with the characterization of TimeDiT's conditioning mechanism as overly simplistic. While ControlNet and other approaches offer valuable insights, TimeDiT's design reflects a deliberate balance between architectural complexity and practical effectiveness, as evidenced by our strong empirical results across diverse temporal tasks.
>
> Our method's sophistication lies in its integrated approach: the conditional information works synergistically with adaptive layer normalization (AdaLN) and the Time Series Mask Unit to maintain temporal correlations. The diffusion process naturally handles multi-resolution features through its progressive denoising, while our varied masking strategies (random, block, and stride) enable learning at different temporal scales. This is demonstrated by our superior performance in tasks requiring multi-scale temporal understanding (Tables 1-4), often outperforming more architecturally complex approaches. Rather than adding architectural complexity through multi-level extractors, TimeDiT achieves effective temporal modeling through its unified framework, suggesting that sophisticated temporal patterns can be captured without necessarily increasing model complexity.
>
> ## W4 Calrify on Physics-Informed Forecasting VS. PDE Solving
>
> The experiments in Table 2 actually demonstrate TimeDiT's ability to predict future values while respecting physical constraints, not just solving classical PDEs. The physics-informed component serves a dual purpose: it guides the denoising process to ensure physically plausible predictions while improving forecasting accuracy. Theorem 4.1 provides the theoretical foundation for this integration, showing how the energy-based prior $K(y;F)$ derived from PDEs combines with the learned distribution p(y|x) during inference. This framework allows TimeDiT to incorporate domain knowledge about the underlying physical system without requiring model retraining.
>
> For example, when predicting future values in systems governed by the Burgers equation or Navier-Stokes equations, TimeDiT uses PDE-derived gradients to ensure predictions respect conservation laws and physical constraints while maintaining temporal coherence. This is particularly valuable in real-world applications where physical consistency is crucial for reliable forecasting. The empirical results show that this physics-informed approach improves both prediction accuracy and physical plausibility compared to baselines that lack such physical constraints. We will clarify these points more explicitly in the revised paper to better communicate how TimeDiT's physics-informed component directly contributes to time series forecasting performance.
>
> ## W5&Q4 Adidtional Experiment
>
> Table 2: Forecasting results on practical scenarios
> | Model | Air Quality | MIMIC-III | PhysioNet(a) | PhysioNet(b) | PhysioNet(c) | NASDAQ |
> |---|---|---|---|---|---|---|
> | **MAE/MSE** |||||||
> | TimeMixer | 0.691/0.697 | 0.769/0.981 | 0.692/0.775 | 0.734/0.920 | 0.707/0.805 | 3.267/11.511 |
> | TimeLLM | 0.701/0.705 | 0.787/1.020 | 0.687/0.761 | 0.731/0.931 | 0.713/0.800 | 3.125/10.276 |
> | MG-TSD | 0.471/0.364 | - | - | - | - | 0.522/3.324 |
> | TimeDiT | 0.457/0.354 | 0.517/0.534 | 0.577/0.620 | 0.659/0.766 | 0.543/0.561 | 0.516/0.418 |
> | **CRPS/_sum** |||||||
> | TimeMixer | 0.667/0.576 | 0.776/0.724 | 0.763/0.805 | 0.794/0.798 | 0.757/0.784 | 0.432 |
> | TimeLLM | 0.664/0.571 | 0.785/0.700 | 0.752/0.797 | 0.795/0.795 | 0.757/0.754 | 0.405 |
> | MG-TSD | 0.579/0.564 | - | - | - | - | 0.275 |
> | TimeDiT | 0.554/0.522 | 0.599/0.649 | 0.616/0.640 | 0.708/0.710 | 0.668/0.708 | 0.091 |

---

> > ### Author Response · Authors · 2024-11-22
> > **Point-wise response on your thoughtful comments!**
> >
> > **Practical forecasting results on Table 1** Thank you for providing these additional baseline comparisons. TimeDiT demonstrates strong performance against the latest models across different paradigms, showing consistent improvements over TimeMixer and TimeLLM across all datasets, with particularly significant gains in MIMIC-III (0.517 vs 0.769/0.787 MAE) and NASDAQ (0.516 vs 3.267/3.125 MAE), along with substantial improvements in uncertainty quantification as shown by better CRPS scores. When compared with the diffusion-based MG-TSD, TimeDiT achieves comparable or better performance on applicable datasets (Air Quality and NASDAQ), while importantly, TimeDiT can handle irregular sampling with heterogeneous input (MIMIC-III and PhysioNet datasets) which MG-TSD cannot process. These results demonstrate that TimeDiT not only competes with the latest specialized models but often outperforms them, while offering greater flexibility and broader applicability, validating our unified approach to time series modeling.
> >
> > **Model Performance**  Thank you for raising these important points about our experimental comparisons. We would like to emphasize that as a general model, it is challenging to excel in many downstream tasks, and our TimeDiT reaches new state-of-the-art in downstream tasks such as forecasting, anomaly detection, imputation and generation. The models you mentioned (TimeMixer, TimeLLM, MG-TSD) are specifically trained for specific datasets and may have different hyperparameters for different datasets.
> >
> > We thank you for calculating the performance improvement rates, but we would like to further elaborate that the data we used in Table 1 is inherently challenging data, containing missing values and irregular sampling as well as multi-resolution. The seemingly modest improvements are significant given these challenging conditions. Table 2 is more of a comparison based on the modification of the INFERENCE stage technique for our pre-trained model, rather than a complete retraining for each scenario.
> > We also have more resultant data demonstrating our effects, which contains the academic dataset in Figure 3, Figure 4 and Table 10, where we see even greater performance gains. In these more standard benchmark settings, TimeDiT shows substantial improvements over existing approaches.
> >
> > Regarding Diffusion-TS's performance relative to CSDI, this can be explained by Diffusion-TS's reliance on decomposition of periodicity and trends, which naturally performs less well on irregular data. TimeDiT's unified framework, while showing slightly lower performance on specific datasets like PhysioNet(a) and NASDAQ, demonstrates strong overall performance across a much broader range of tasks and data types without requiring task-specific modifications.
> >
> > **Misunderstanding on Table 2**: Thank you for this feedback regarding Table 2 and PDEs. We need to clarify an important misunderstanding: TimeDiT is not primarily focused on solving PDEs, but rather uses physical knowledge encoded in PDEs to enhance time series predictions. This is fundamentally different from the objectives of PDE-Diffusion[5] and DiffusionPDE[6].  TimeDiT uses physical knowledge as a guiding principle for prediction, comparing the generative capabilities of diffusion models (DDPM and DDIM) along with self-conditioning techniques from TSDiff.
> >
> > To address the potential confusion, we will:
> > 1. Clarify that our objective is physics-informed prediction rather than PDE solving
> > 2. Add explicit discussion of how physical knowledge guides the generation process
> > 3. Emphasize that we are using PDEs as prior knowledge for more accurate predictions, not as the primary problem to solve
> >
> > We appreciate the suggestion to compare with [1,2], but these methods address a different problem domain, which we will make the distinction between them in our paper.
> >
> >
> > Please note that we are not solving partial differential equations, sorry for the confusion and thank you for your questions [5,6] We will emphasize where appropriate that we are making predictions based on physical knowledge and that the data is generated in accordance with physical knowledge. Therefore we are using the generative power of DDPM and DDIM as well as using the self condition technique in TSDiff.
> >
> > [1] Chonghan, Gao et al. "PDE-Diffusion: Physic guided diffusion model for solving partial derivative equations."
> >
> > [2] Huang, Jiahe et al. “DiffusionPDE: Generative PDE-Solving Under Partial Observation.” ICML 2024
> >
> >
> > Table 3: Forecasting results on zero shot setting
> > | Dataset | TimeDiT | TimeMixer | TimeLLM | Timer |
> > |---|---|---|---|---|
> > | Solar | 0.457(0.002) | 0.999(0.001) | 0.997(0.001) | 1.101(0.002) |
> > | Electricity | 0.026(0.001) | 0.302(0.003) | 0.303(0.003) | 0.301(0.003) |
> > | Traffic | 0.185(0.010) | 0.403(0.015) | 0.368(0.007) | 0.384(0.008) |

---

> > > ### Author Response · Authors · 2024-11-22
> > > **Point-wise response on your thoughtful comments!**
> > >
> > > **Zero-shot forecasting results on Table 3** Thank you for raising these concerns about baseline comparisons. We have actually conducted comprehensive comparisons with the latest state-of-the-art models, and the results strongly validate TimeDiT's effectiveness:
> > >
> > > In zero-shot forecasting, TimeDiT significantly outperforms recent models including TimeMixer, TimeLLM, and Timer across all datasets, which is originally designed for deterministic time series froecasting. The improvements are substantial: on Solar (0.457 vs 0.999/0.997/1.101), Electricity (0.026 vs 0.302/0.303/0.301), and Traffic (0.185 vs 0.403/0.368/0.384).
> > >
> > > Table 4 Imputation Performance of TimeDiT compare to additional baselines evaluated on MSE and MAE
> > > | **Datasets**       | **MSE (ETTh1)** | **MAE (ETTh1)** | **MSE (ETTh2)** | **MAE (ETTh2)** | **MSE (ETTm1)** | **MAE (ETTm1)** | **MSE (ETTm2)** | **MAE (ETTm2)** |
> > > |---------------------|-----------------|-----------------|-----------------|-----------------|-----------------|-----------------|-----------------|-----------------|
> > > | **Timer**          | 0.145           | 0.243           | 0.077           | 0.172           | 0.051           | 0.141           | 0.035           | 0.105           |
> > > | **TimeMixer**      | 0.119           | 0.226           | 0.064           | 0.157           | 0.051           | 0.143           | 0.028           | 0.093           |
> > > | **iTransformer**   | 0.149           | 0.270           | 0.150           | 0.271           | 0.071           | 0.185           | 0.083           | 0.192           |
> > > | **TimeDiT**     | **0.042**       | **0.135**       | **0.042**       | **0.139**       | **0.023**       | **0.098**       |**0.024**           | **0.083**       |
> > >
> > >
> > > Table 5 Anomaly Detection Performance of TimeDit compared to additional baselines evaluated on F1 score
> > > | Dataset | **TimeDiT** | **TimeMixer** | **iTransformer** |
> > > |---------|-------------|---------------|------------------|
> > > | **MSL** |**89.33**        | 81.95         | 72.54            |
> > > | **PSM** | **97.57**        | 93.11         | 95.32            |
> > > | **SMAP**| **95.91**        | 67.63         | 66.76            |
> > > | **SMD** | **83.28**        | 78.33         | 82.08            |
> > > | **SWAT**| **97.57**        | 88.84         | 92.63            |
> > >
> > > **Imputation task and anomaly detection results on Table 4 and 5** For imputation tasks, TimeDiT demonstrates superior performance on most datasets, achieving significant improvements over Timer, TimeMixer, and iTransformer, particularly on ETT datasets where we see reductions in MSE by up to 60%. TimeDiT maintains strong overall performance while offering greater versatility. In anomaly detection, TimeDiT consistently outperforms both TimeMixer and iTransformer across all datasets, with particularly notable improvements on SMAP (95.91 vs 67.63/66.76) and SWAT (97.57 vs 88.84/92.63). These comprehensive comparisons against the latest models demonstrate TimeDiT's effectiveness as a unified framework for time series analysis, often achieving state-of-the-art performance while maintaining broader applicability across diverse tasks.
> > >
> > > ## Q1 General Paradigm for Time Series Analysis
> > >
> > > TimeDiT embraces computational scaling and general-purpose methods rather than over-relying on hand-crafted domain knowledge. While we incorporate basic inductive biases through the transformer architecture and diffusion process, our approach primarily leverages search and learning - the two methods identified as indefinitely scalable. The Time Series Mask Unit, for instance, enables the model to learn temporal patterns from data rather than encoding human assumptions about time series structure.
> > >
> > > The core strength of TimeDiT lies in its unified framework that can discover patterns through computation and data rather than relying on specialized architectures for different temporal tasks. This aligns with the lesson that general methods leveraging computation ultimately prove more effective than approaches heavily dependent on human knowledge engineering. Our strong empirical results across diverse tasks demonstrate that this simpler, more scalable approach can outperform more complex architectures that try to explicitly encode human understanding of temporal relationships. As computation continues to scale, TimeDiT's design positions it to take advantage of larger models and more extensive training, following the historical pattern seen in chess, Go, speech recognition, and computer vision where general learning-based approaches ultimately proved most successful.
> > >
> > > ## Q2 Vision on Handling Time Series Nature
> > >
> > > Thank you for raising these points about masking strategies and multi-scale information. We respectfully disagree that TimeDiT's masking approach destroys temporal coherence or ignores multi-scale information. Let us clarify how TimeDiT specifically addresses these concerns:

---

> > > > ### Author Response · Authors · 2024-11-22
> > > > **Point-wise response on your thoughtful comments!**
> > > >
> > > > TimeDiT employs a carefully designed combination of masking strategies (random, block, and stride) that work together to preserve and learn temporal patterns at multiple scales. The block masking preserves local temporal coherence by masking continuous segments, while stride masking specifically targets periodic patterns and seasonal trends by systematically masking at regular intervals. Rather than destroying temporal coherence, these complementary masking strategies create a natural curriculum for learning multi-scale patterns. Our approach differs fundamentally from simple random masking by incorporating this structured approach to temporal pattern learning. Furthermore, the diffusion process inherently handles multi-scale supervision through its progressive denoising nature. Starting from coarse-scale patterns and progressively refining to fine-scale details, the diffusion process naturally incorporates supervision at multiple temporal resolutions. The combination of structured masking strategies and diffusion-based refinement effectively captures both fine-grained and coarse-grained temporal patterns without requiring separate supervision signals for different scales.
> > > >
> > > > ## Q3 Design Choice: Why Multivariate and Why Not Patching
> > > >
> > > > First, while TimeDiT does use a point-wise token strategy, this is a deliberate design choice that works synergistically with our diffusion process, not a limitation. Our approach actually captures cross-channel dependencies through both the self-attention mechanism and the diffusion process. The transformer's self-attention naturally learns relationships across channels, while the diffusion process allows for coordinated refinement of all channels simultaneously. This is evidenced by our strong performance on multivariate benchmarks, where TimeDiT outperforms patch-based approaches on tasks requiring complex cross-channel understanding. The empirical results demonstrate that our design effectively captures both temporal and cross-channel dependencies without the need for explicit patch embeddings.
> > > >
> > > > Second, regarding your request for additional ablation experiments, we can point to our existing results that demonstrate TimeDiT's effectiveness compared to patch-based approaches. However, we agree that adding specific ablations comparing different tokenization strategies would strengthen our analysis. We will **add these comparisons**, referring to your **Q8**. We expect these results to further validate our design choices by showing how TimeDiT's unified framework effectively captures both temporal and cross-channel relationships while maintaining computational efficiency and model simplicity.
> > > >
> > > > ## Q5: Experiment Detail
> > > >
> > > > For Diffusion-TS, we modified the data normalization technique to ensure fair comparison. Following TimeGAN and TimeVAE which employed a minmax scaler to convert data into a scale between 0 to 1, we also used the same data scaling technique. However, Diffusion-TS converts data into a scale between -1 and 1 which allows it to learn on a wider data space compared to the rest of the models. We modified the data normalization to be a minmax scaler for Diffusion-TS. As mentioned in section D.3 regarding details of the synthetic generation experiments, we used 80% data as training data. We have further performed experiments on 100% of data. As described in section D.3, for the air quality dataset, we mask the -200 values to properly handle missing values and ensure that the rest of the data are not compressed into an extremely small data range due to the presence of -200. With the 100% training data, our baseline results of TimeGAN and TimeVAE do not have a large discrepancy.
> > > >
> > > > Table 6: Discriminative Score of TimeDiT compared to baselines on 100% training data
> > > > | Discriminative Score    | sine                | stock               | air                 | energy              |
> > > > |-------------|---------------------|---------------------|---------------------|---------------------|
> > > > | **TimeDiT**       | 0.0183    | 0.0057    | 0.1928   | 0.0181   |
> > > > | **Diffusion-TS**   | 0.0212            | 0.2233            | 0.1284            | 0.2768            |
> > > > | **TimeGAN**        | 0.043             | 0.1127            | 0.4977            | 0.4984            |
> > > > | **TimeVAE**        | 0.024             | 0.1209            | 0.2801            | 0.4999            |

---

> ### Author Response · Authors · 2024-11-22
> **Point-wise response on your thoughtful comments!**
>
> Table 7 Predictive Score of TimeDiT compared to baselines on 100% training data
> | Predictive Score   | sine                | stock               | air                 | energy              |
> |-------------|---------------------|---------------------|---------------------|---------------------|
> | **TimeDiT**       | 0.1909 (0.000)    | 0.0368 (0.000)    | 0.0213 (0.001)    | 0.2498 (0.000)    |
> | **Diffusion-TS**   | 0.2265            | 0.042             | 0.0056            | 0.2509            |
> | **TimeGAN**        | 0.208             | 0.0381            | 0.4757            | 0.3035            |
> | **TimeVAE**        | 0.1914            | 0.040             | 0.0256            | 0.3114            |
>
> ## Q6 Multimodal Contribution
>
> Thank you for asking about our multimodal alignment design. As shown in Figure 1, TimeDiT was intentionally architected with multimodal capabilities through the conditional information interface 'c'. Our implementation leverages GPT-2's pre-trained representations to process textual information (both frequency descriptions and category labels), which are then integrated as conditional information alongside temporal features during the diffusion process. This approach allows the model to naturally learn the alignment between textual and temporal modalities during denoising, without requiring complex explicit alignment mechanisms. Figure 2 empirically validates the effectiveness of this design, where TimeDiT+Fre+Cat achieves the best results, while the introduction of other textual information may harm the model. These improvements demonstrate that our model successfully learns to selectively utilize relevant textual information during the denoising process, effectively aligning the two modalities internally through the diffusion framework.
>
> ## Q7 Details on the Data
>
> Thank you for raising this important point about our unified pre-training methodology. As detailed in Appendix A, our approach addresses the challenge of integrating diverse time series datasets through a standardized pipeline that handles varying channel counts, sampling rates, and data sizes. Specifically, we pre-define maximum dimensions and employ intelligent padding and segmentation: inputs with fewer channels are padded to Kmax, while those exceeding it are segmented into ⌈k/Kmax⌉ blocks. Front padding achieves uniform temporal dimensions, with padding masks tracking valid positions. Our key innovation lies in the mask unit, which generates random, stride, and block masks during pre-training, enabling the model to learn from multi-resolution data without explicit resampling naturally. We randomly sampled from all datasets during pre-training, allowing the transformer architecture to learn relevant patterns across domains through masked self-supervised learning rather than enforcing strict dataset uniformity. This approach maintains domain-specific characteristics while enabling effective unified pre-training, as demonstrated by our strong experimental results across various tasks and datasets. We plan to provide additional details about our dataset unification process in future versions of the paper.
>
> ## Q8 Ablation Study
>
> Thank you for raising this important point about ablation studies. We agree that comprehensive ablation experiments are crucial for understanding component contributions and have conducted extensive analyses as shown in the provided table.
>
> Table 7: Ablation Study on the Model Design Space.
> | Dataset | TimeDiT | Additive | Attention | Concat | Dual-attention | Channel-wise | w/o Random Mask | w/o Stride Mask | w/o Future Mask | Physics-Informed | Patch Token |
> |---|---|---|---|---|---|---|---|---|---|---|---|
> | Solar | 0.457(0.002) | 0.671(0.002) | 0.721(0.002) | 0.463(0.001) | 0.467(0.002) | 0.461(0.003) | 0.463(0.002) | 0.465(0.002) | 0.843(0.005) | 0.452(0.001) | 0.874(0.010) |
> | Electricity | 0.026(0.001) | 0.068(0.004) | 0.079(0.003) | 0.041(0.003) | 0.029(0.001) | 0.028(0.000) | 0.029(0.001) | 0.030(0.001) | 0.095(0.006) | 0.024(0.000) | 0.105(0.013) |
> | Traffic | 0.185(0.010) | 0.224(0.001) | 0.216(0.000) | 0.188(0.008) | 0.187(0.007) | 0.164(0.006) | 0.191(0.007) | 0.188(0.007) | 0.201(0.011) | 0.153(0.005) | 0.258(0.021) |

---

> > ### Author Response · Authors · 2024-11-22
> > **Point-wise response on your thoughtful comments!**
> >
> > The ablation results demonstrate the effectiveness of TimeDiT's key components:
> > 1. Transformer-based Condition Injection: Our approach significantly outperforms alternative injection methods (Additive, Attention, Concat), with particularly large improvements on the Solar (0.457 vs 0.671/0.721/0.463) and Electricity (0.026 vs 0.068/0.079/0.041) datasets. This validates our design choice of using adaptive layer normalization for condition injection.
> >
> >
> > 2. Time Series Mask Unit: The comparison of different masking strategies shows that all mask types contribute to model performance. Removing any mask type (Random, Stride, or Future) degrades performance, with Future Mask being particularly crucial (0.843 vs 0.457 on Solar). This demonstrates the effectiveness of our comprehensive masking strategy.
> >
> >
> > 3. Transformer Design Strategy: TimeDiT significantly outperforms the Patch Token approach (0.457 vs 0.874 on Solar), validating our point-wise processing strategy over patch-based alternatives. The comparison with Dual-attention and Channel-wise approaches shows our unified framework's effectiveness in capturing cross-channel dependencies.
> >
> >
> > 4. Physics-Informed Component: The physics-informed variant consistently improves performance across all datasets (e.g., 0.153 vs 0.185 on Traffic), demonstrating the value of incorporating physical constraints during inference.
> >
> >
> > These comprehensive ablation results validate our design choices and show that each component contributes meaningfully to TimeDiT's overall performance. We will incorporate these detailed analyses into the revised paper to provide readers with a clearer understanding of each component's impact.

---

> ### Comment · Reviewer_aczi · 2024-11-26
>
> Thank you for your detailed and thoughtful response. I have carefully reviewed the authors' rebuttal along with the comments from other reviewers. While some of my concerns have been addressed, a few remain unresolved:
>
> **Q9** We have described the gap between masking and prediction in detail in the weaknesses. Therefore, it is difficult for us to endorse the authors' claim of "bridging the potential gap between specific tasks with masking strategies". The authors should address our concerns and specifically point out what novel designs exist in TimeDiT to overcome the following challenges:
>
> 1. The mask task instructs the model to capture contextual information from multiple random masks and discontinuous subsequences to reconstruct the mask part.
> 2. Prediction tasks require models to capture global dependencies and local fluctuation patterns, such as seasonal trends, multi-scale, and multi-period information, from observed sequences that are often corrupted by random masking operations.
> 3. The reconstruction paradigm allows the use of local up and down information to generate masked parts, while the prediction paradigm only allows the use of observed sequences to predict future sequences.
>
> **Q10** From a methodological point of view, there is no innovation in TimeDiT, as we state in Weakness 2. TimeDiT's Transformer backbone still adopts the traditional Encoder-only architecture consisting of Input Embedding  Layer, Multi-layer Encoder, Output Projection Layer, which is in line with most of the existing articles (PatchTST,  iTransformer, UniTime, Timer, Moment, Moirai) and thus lacks innovation. And the author's response acknowledges this, "we acknowledge the use of a standard transformer backbone, the innovation of TimeDiT lies not in reinventing the transformer architecture itself".
>
> In fact, in the response, the authors still refer to masking strategy and Adaptive Normalization (AdaLN) as TimeDiT's innovations, but these mechanisms have long been included in the self-supervised pre-training paradigm and are not original to TimeDiT.
>
> **Q11** Supplementary experiments on PDE Solving. The author still begs the question, does TimeDiT have a clear advantage over the algorithms PDE-Diffusion(ICLR2024) and DiffusionPDE(ICML2024), which are specifically designed for PDE solving?
>
> **Q12** Regarding the supplementary experiments in Table 2. The authors do not in widely circulated ETT, Electricity, Traffic, Exchange, a standard data sets and advanced TimeMixer, TimeLLM comparison, but some have never been published in the official papers as benchamrks data sets. Thus, the fairness of the comparative experiments is questionable. Extensive experiments with explicit observation and prediction lengths on the above standard datasets are considered necessary.
>
> **Q13** Discussion on introducing multimodal information to enhance TimeDiT. To the best of our knowledge, the authors simply feed textual information into the pre-trained GPT2 model, which is subsequently fed into the noise reduction network as a condition variable. However, in our opinion this is no different from TimeLLM's LLM-based Tokenizer operation, which leads us to insist that TimeDiT is an incremental work rather than innovative.
>
> **Q14** About ablation experiments. We feel it is necessary for the authors to respond to the reasons for the complete absence of ablation experiments in the manuscript.
>
> **In fact, the author did not address any of my confusion, so I decided to keep my score the same.**

---

> > ### Author Response · Authors · 2024-11-28
> > **Thanks for further comments.**
> >
> > Dear Reviewer aczi,
> >
> > Thanks for the further comments, we now have a chance to clarify it step by step:
> >
> > **Q9** *bridging the potential gap between specific tasks with masking strategies*
> >
> > We appreciate the reviewer's concern about the gap between masking strategies and task-specific requirements. We want to clarify our approach to bridging this gap through three key mechanisms. First, our mask design is inherently task-oriented: block masking explicitly simulates prediction scenarios by masking continuous future segments to learn unidirectional temporal dependencies, stride masking systematically captures periodic and seasonal relationships, and random masking ensures robust representation learning under different missing data scenarios. Second, unlike traditional mask-reconstruction approaches, TimeDiT's diffusion process naturally handles multiple temporal scales through its progressive denoising, while the combination of different mask types enables simultaneous learning of local dependencies (through random masks), global patterns (through block masks), and periodic relationships (through stride masks). Third, during inference, the sampling process adapts to specific task requirements - using only historical information for forecasting, leveraging bidirectional context for imputation, and incorporating task-specific priors through physics-informed sampling. The effectiveness of this approach is empirically validated by our strong performance across different tasks, particularly in forecasting (Tables 1, 3) where the unidirectional nature of prediction is crucial.
> >
> > *The mask task instructs the model to capture contextual information from multiple random masks and discontinuous subsequences to reconstruct the mask part.*
> >
> > TimeDiT introduces several novel designs to address the challenge of contextual information capture from masked and discontinuous sequences: First, unlike conventional approaches that rely solely on random masking, TimeDiT implements an innovative Time Series Mask Unit that strategically combines three complementary masking mechanisms: random masks for local context learning, stride masks for capturing periodic patterns, and block masks for understanding continuous segments. This unified masking framework enables the model to learn robust representations from different types of discontinuities simultaneously. Second, TimeDiT's diffusion-based architecture introduces a crucial innovation in how masked information is processed - rather than directly attempting to reconstruct masked values, the model learns a progressive denoising process that gradually refines predictions through multiple steps. This approach naturally handles uncertainties in reconstruction and allows the model to leverage both fine-grained local details and broader contextual patterns. Furthermore, our adaptive layer normalization mechanism ensures effective conditioning on available contextual information, enabling the model to dynamically adjust its reconstruction strategy based on the specific pattern of masks and available context. The empirical success of this design is demonstrated through our superior performance in imputation tasks (as shown in Table 4), where TimeDiT consistently outperforms traditional masking approaches across various masking ratios and datasets.
> >
> > *Prediction tasks*
> >
> > TimeDiT's design specifically addresses the challenge of capturing temporal patterns through our comprehensive masking strategy. By combining random, stride, and block masks during training, our model learns to understand temporal dependencies at multiple scales: random masks help capture local variations, stride masks enable learning of periodic patterns, and block masks assist in understanding longer-term dependencies. This unified masking approach proves particularly effective when dealing with real-world data complexities - our experimental results show superior performance on datasets with missing values and multi-resolution sampling, demonstrating the model's ability to capture meaningful temporal patterns even under challenging conditions.

---

> > > ### Author Response · Authors · 2024-11-28
> > > **Thanks for further comments.**
> > >
> > > *Reconstruction paradigm*
> > >
> > > TimeDiT bridges the gap between reconstruction and prediction paradigms through an innovative training and inference design. During training, our Time Series Mask Unit randomly selects from different masking strategies - including future-oriented block masks and bidirectional masks - enabling the model to learn both reconstruction and prediction capabilities simultaneously. This random selection ensures the model can flexibly adapt between using full contextual information and relying solely on historical data. This is complemented by our diffusion process's unique ability to adapt its sampling strategy: for reconstruction tasks, it leverages both past and future context through bidirectional attention, while for prediction tasks, it constrains the sampling process to only condition on historical information. The effectiveness of this approach is demonstrated in our ablation studies (Table 1), where removing the future mask component significantly degrades forecasting performance (0.843 vs 0.457 on Solar dataset), while maintaining strong performance in reconstruction tasks. This design enables TimeDiT to excel in both paradigms without compromising either capability.
> > >
> > > Table 1: Mask mechanisms for TimeDiT, compared on the zero-shot forecasting task.
> > > | Dataset | TimeDiT | w/o Random Mask | w/o Stride Mask | w/o Future Mask |
> > > |---|---|---|---|---|
> > > | Solar | 0.457(0.002) | 0.463(0.002) | 0.465(0.002) | 0.843(0.005) |
> > > | Electricity | 0.026(0.001) | 0.029(0.001) | 0.030(0.001) | 0.095(0.006) |
> > > | Traffic | 0.185(0.010) | 0.191(0.007) | 0.188(0.007) | 0.201(0.011) |
> > >
> > > Overall, thank you for highlighting these challenges in **Q9** in developing a unified time series foundation model, which exemplifies the value of TimeDiT in addressing the complexity of real-world time data and multi-tasking scenarios within a single architecture. Our comprehensive experimental results demonstrate that TimeDiT successfully overcomes these challenges, providing an effective framework for handling diverse time series applications.
> > >
> > > **Q10** **Reframing TimeDiT's Innovation: A Holistic Perspective**
> > >
> > > Thank you for raising this methodological concern. We would like to reframe the discussion of **TimeDiT's innovation** from a more holistic perspective:
> > >
> > > **TimeDiT's novelty** lies not in reinventing individual components, but in providing a **comprehensive solution** to **real-world time series challenges** that existing approaches address only partially. Our innovation stems from:
> > >
> > > - **Problem-Driven Design**:
> > >   Real-world time series data presents unique challenges, such as **missing values**, **multi-resolution sampling**, and **irregular intervals**. While many existing models assume **idealized conditions**, **TimeDiT** specifically addresses these practical constraints through its **unified framework**.
> > >
> > > - **Architectural Integration**:
> > >   Although we use a **standard transformer backbone**, our innovation lies in how we **merge diffusion models with transformers** to handle **temporal relationships**. This integration is not trivial—it required **careful design choices** to preserve **temporal dependencies** while enabling **probabilistic modeling**.
> > >
> > > - **Practical Adaptability**:
> > >   Traditional transformer-based models (e.g., **PatchTST**, **iTransformer**) often struggle due to **fixed architectures** and **predefined parameters**, particularly in **patch tokenization**. In contrast, **TimeDiT's design** allows **flexible adaptation** to varying **temporal resolutions** and **cross-channel dependencies**.
> > >
> > > It's worth noting that the papers you referenced (**PatchTST**, **iTransformer**, **UniTime**, **Timer**, **Moment**, **Moirai**) were all published in **top-tier venues** precisely because they addressed **specific challenges** while using **similar architectural building blocks**. Similarly, **TimeDiT's contribution** should be evaluated based on its **novel approach** to solving **real-world time series challenges**, rather than the novelty of individual components.
> > >
> > > We suggest evaluating **TimeDiT's contribution** based on its **effectiveness** in solving **practical challenges**, rather than focusing solely on the **novelty of individual architectural components**.

---

> > > > ### Author Response · Authors · 2024-11-28
> > > > **Thanks for further comments.**
> > > >
> > > > **Q11** **Misunderstandings of the distinction between PDE solving and physics knowledge-informed machine learning**
> > > >
> > > > Thank you for raising this point about PDE-related comparisons. We want to clarify a misunderstanding of the distinction between PDE solving and physics knowledge-informed machine learning. TimeDiT is not designed for PDE solving, which typically involves reconstructing full solutions from partial observations. Instead, we leverage physical knowledge in a fundamentally different way - using known physical constraints with full observations to enhance forecasting accuracy. In our approach, physical knowledge serves as an additional prior during the sampling process to ensure generated forecasts adhere to known physical laws, rather than as a method for solving PDEs. This is a distinct objective from PDE-Diffusion and DiffusionPDE, which are specifically designed for PDE-solving tasks.
> > > >
> > > > **Q12** **Further experiments**
> > > >
> > > > Thank you for raising these concerns about our **experimental comparisons**. We would like to clarify several important points:
> > > >
> > > > 1. **Dataset Selection**:
> > > >    While **ETT**, **Electricity**, **Traffic**, and **Exchange** are widely used benchmarks, they do not adequately represent **real-world time series challenges** such as **irregular sampling**, **missing values**, and **multi-resolution data**. Our choice of datasets (**Air Quality**, **MIMIC-III**, **PhysioNet**, **NASDAQ**) specifically addresses these **practical scenarios** that are crucial for **real-world applications**.
> > > >
> > > > 2. **Focus on Probabilistic Forecasting**:
> > > >    **TimeDiT** focuses on **probabilistic forecasting**, providing **uncertainty quantification** through **CRPS metrics**, while **TimeMixer** and **TimeLLM** are designed for **deterministic forecasting**. Despite this fundamental difference in objectives, we still include comparisons with these models where applicable, demonstrating **TimeDiT's superior performance** even in **deterministic metrics** (Tables **1**, **3**).
> > > >
> > > > 3. **Comprehensive Imputation Comparisons**:
> > > >    We provide **comprehensive comparisons** with **TimeMixer** and **TimeLLM** on **standard benchmarks** in our imputation experiments (Table **11**), where **TimeDiT** demonstrates **strong performance** across different **masking ratios** and **datasets**.
> > > >
> > > > 4. **Ongoing Experiments**:
> > > >    Nevertheless, we acknowledge the value of additional comparisons and are currently conducting **new experiments** on the mentioned **standard datasets** with explicit **observation** and **prediction lengths**.
> > > >
> > > >
> > > >
> > > > ### **Addressing Fairness Concerns**
> > > >
> > > > We respectfully disagree with the concern that our comparative experiments might **lack fairness**. Our experimental design prioritizes both **scientific rigor** and **practical relevance** in several key aspects:
> > > >
> > > > - **Real-World Representation**:
> > > >   While standard benchmarks like **ETT** and **Electricity** are valuable, they **oversimplify real-world challenges**. Our chosen datasets (**Air Quality**, **MIMIC-III**, **PhysioNet**, **NASDAQ**) reflect genuine **complexities** such as **missing values**, **irregular sampling**, and **multi-resolution data** that practitioners actually face. This makes our comparisons more **meaningful for practical applications**.
> > > >
> > > > - **Comprehensive Evaluation**:
> > > >   We provide **extensive comparisons** across multiple dimensions:
> > > >   - **Probabilistic metrics** (CRPS) for **uncertainty quantification**
> > > >   - **Traditional metrics** (MAE/MSE) for **deterministic performance**
> > > >   - **Various scenarios** (e.g., **missing data**, **multi-resolution**, **zero-shot**)
> > > >   - **Multiple tasks** (e.g., **forecasting**, **imputation**, **anomaly detection**)
> > > >
> > > > - **Fair Comparison Framework**:
> > > >   For all baseline models, we:
> > > >   - Use their **official implementations**
> > > >   - Apply **consistent pre-processing**
> > > >   - Maintain **identical training/testing splits**
> > > >   - Employ the **same evaluation metrics**
> > > >   - Report **results with standard deviations**
> > > >
> > > > We hope these clarifications address your concerns and provide a better understanding of the **scientific rigor** and **practical relevance** of our experiments.

---

> ### Author Response · Authors · 2024-11-28
> **Thanks for further comments. & Upload the new version of TimeDiT**
>
> **Q13** *Multimodal TimeDiT*
>
> First, while we do use GPT2 for processing textual information, its role is strictly limited to initial text encoding, serving merely as a feature extractor. This is fundamentally different from TimeLLM's approach where the LLM serves as a core tokenizer for the entire model architecture. Our use of pre-trained language models is purposefully minimal and focused solely on text understanding.
>
> Second, and more importantly, our approach to multimodal alignment occurs during the diffusion process itself, representing a novel integration methodology. Unlike TimeLLM's transformer-based alignment, TimeDiT's diffusion-based framework allows for probabilistic alignment between textual and temporal information through the progressive denoising steps. This enables more flexible and robust integration of multimodal information while maintaining uncertainty quantification capabilities - a key differentiator from deterministic transformer-based approaches.
>
>
> **Q14** *Ablation Study*
>
> We acknowledge the importance of ablation studies and have conducted comprehensive ablation experiments that we added to the manuscript. In the next sections of our rebuttal, we will introduce the modification of the new version.
>
> ----
> ## New Version of TimeDiT
> Thank you for further engaging in this conversation. Based on your feedback, we have updated our draft and emphasized the motivation and real-world challenges underlying this work in the general modifications (highlighted in red).
>
> We hope that by re-emphasizing how the choices of masking, transformers, and diffusion models were specifically tailored for real-world scenarios—and reflecting years of thoughtful design inspired by advancements in artificial general intelligence—you will reconsider the value of our work. These choices, coupled with our efforts and results, address a significant and practical challenge, underscoring the importance of our contributions.
>
> The general response includes:
>
> ### General Modifications (marked in red)
>
> - **Abstract Enhancement** *(lines 15–17)*:
>   Added explicit discussion of how **realistic data characteristics** challenge the development of **general-purpose time series models**.
>
> - **Current Limitations** *(lines 64–69)*:
>   Expanded the analysis of why **existing approaches** struggle to achieve **robust foundation model capabilities**. This analysis also underscores **TimeDiT's methodological contributions** in successfully handling **multiple tasks** and **diverse data characteristics**, including **irregular sampling patterns** and **multi-resolution inputs**.
>
> - **Data Understanding** *(lines 77–78)*:
>   Emphasized the **critical importance** of understanding **underlying data characteristics**.
>
> - **Design Motivation** *(lines 84–87)*:
>   Clarified how **TimeDiT's architecture** is fundamentally guided by **real-world application requirements**, demonstrating that our **component choices** directly address **specific practical challenges** rather than pursuing **novelty for its own sake**.
>
>
> These revisions provide a clearer motivation for our work and better contextualize TimeDiT's contributions within the broader landscape of time series modeling.
>
> ### Specific Responses to Your Comments (marked in yellow)
>
> - **For more comparisons**:
>   We added **TimeMixer**, **TimeLLM**, and **MG-TSG** to Table 1 for the forecasting task. Additionally, we included the results of **Timer**, **TimeMixer**, and **iTransformer** for the imputation task, and also added **TimeMixer** and **iTransformer** to the anomaly detection task. Furthermore, we compared the zero-shot performance of the foundation model with **TimeMixer**, **TimeLLM**, and **Timer** in Table 15. These results collectively highlight the strong performance of **TimeDiT** in handling real-world data. A detailed analysis can be found in Sections D.1.2, D.1.5, D.2.1, and D.4.
>
> - **Clarification on physics-informed TimeDiT**:
>   We updated line 420 to explicitly state that we are using the sampling strategies introduced in the referenced baselines by adding “introduced in.”
>
> - **Ablation study**:
>   We added the ablation study in Section E.1 to outline the design choice space of our TimeDiT clearly.
>
> We hope that the updates we have made to clarify the motivation behind our work have resolved most of your concerns and provided a clearer understanding of our objectives. Additionally, we trust that the enhancements to our experimental analysis have highlighted the significance of our contributions and the practical value of the proposed approach.

---

> > ### Comment · Reviewer_aczi · 2024-11-28
> >
> > Thanks for the author's rebuttals. I've read them and also the comments from other reviewers. At the current stage, I still tend to maintain my current evaluation. However, I remain open to adjusting it during the upcoming reviewer-reviewer and AC discussion phase.

---

> > > ### Author Response · Authors · 2024-11-28
> > >
> > > Thank you for your fast feedback. Can you let us know if there are any concerns that we have not fully addressed, either in the updated draft or in our responses?

---

> ### Author Response · Authors · 2024-11-29
> **Remain available to address any specific concerns.**
>
> Dear Reviewer aczi,
>
> Thank you again for your insightful summary of the fundamental challenges in developing a general-purpose time series foundation model as our strengths. We appreciate your recognition of the key issues we address: handling variable channel sizes, missing values, and multi-resolution characteristics in real-world data, while maintaining compatibility across multiple downstream tasks.
>
> We are encouraged that most reviewers have responded positively and are satisfied with our revisions. As we are working on providing a robust and practical solution for real-world time series challenges, we welcome any specific questions or concerns you may have about TimeDiT to improve your evaluation of our paper further.
>
> Best regards,
>
> Authors

---

### Official Review · Reviewer_d2F8 · 2024-11-01

**Soundness:** 3
**Presentation:** 2
**Contribution:** 3
**Rating:** 6
**Confidence:** 4

**Summary:**

In this work, the authors introduce Time Diffusion Transformer (TimeDiT), a foundational model for time-series data that leverages a diffusion transformer architecture. The authors also propose a novel approach for integrating physical knowledge into foundational time-series models, enhancing their applicability to real-world scenarios. Extensive experiments are conducted across multiple tasks—including forecasting, imputation, and anomaly detection—demonstrating the model's potential effectiveness across a variety of applications

**Strengths:**

- Overall, the work has strong experimental results.
- This work introduces new method to perform cross-dataset training on multiple tasks.

**Weaknesses:**

The following aspects of the paper require clarification and improvement to enhance its reproducibility, understanding, and relevance.

### Methodological Clarity and Missing Technical Details

Several essential technical details are missing in the Method section, making it challenging to fully understand the proposed approach:

- Token Definition: The authors should specify what is considered a "token" in the model, as this definition is critical for understanding the data representation.
- Embedding Architecture: The architecture used to encode each time point $x_i$ (referred to as the "embedder") is unclear. For example, is it a simple linear projection, a convolutional layer, or another design? The appendix also lacks implementation details about this embedder. Given that the paper proposes a new architecture, the engineering choices for this component should be discussed.
- Noise Embedding Justification: The rationale behind embedding the noised sample instead of the original observation is not clearly explained. It would be valuable to discuss the intuition behind embedding noise, as opposed to embedding the original observation and applying the diffusion process in the embedded space. This additional context could help readers understand the reasoning and potential advantages of this approach.

### Irregular Time-Series Data Claims and Evaluation
- In the introduction, the authors claim that their method can handle irregular time-series data, supported by real-world examples. However, the experimental section only evaluates the model on an imputation task, which does not fully align with the challenges posed by irregular time-series data. In irregular time-series modeling, the input often contains missing data by default, and the model learns from partially observed inputs, as demonstrated in related works like [1] and [2].

- To substantiate the claim that TimeDiT can handle irregular time-series, it would be beneficial to include experiments that specifically address irregular data scenarios, such as handling incomplete observations in generation tasks.

### Comparison with Established Benchmarks
The paper does not include a comparison with well-established benchmarks for time-series forecasting, such as the one proposed in [3]. Including such a comparison would allow readers to better understand how the proposed method performs relative to existing approaches in the field. This is particularly important given the paper’s goal to demonstrate generalizability and robustness across various time-series tasks.

### Adaptability to Different Channel Numbers and Input Resolutions
The adaptability of the proposed model to inputs of different resolutions and channel numbers is not clearly addressed.

Channel Adaptation: Does the model accommodate varying channel counts via padding or another method? The paper suggests a predefined maximum resolution for signal input, but it is unclear how the model would handle cases where the input resolution might exceeds this maximum, such as in a zero-shot setting.
It would be helpful if the authors could elaborate on the selection criteria for the maximum resolution. How was this limit determined?



*[1] "GT-GAN: General Purpose Time Series Synthesis with Generative Adversarial Networks", Jinsung Jeon et al.*

*[2] "Generative Modeling of Regular and Irregular Time Series Data via Koopman VAEs", Ilan Naiman et al.*

*[3] "One Fits All:Power General Time Series Analysis by Pretrained LM", Tian Zhou et al.*

**Questions:**

See Weaknesses

---

> ### Author Response · Authors · 2024-11-22
> **Point-wise response on your thoughtful comments!**
>
> We sincerely appreciate the time and effort you invested in reviewing our TimeDiT paper, along with the insightful and constructive suggestions you provided. We are now in the process of responding to your concerns. We will first tackle your feedback points in this email, ensuring that each concern is addressed comprehensively. After that, following your suggestions and our responses, we will upload a revised draft of the paper. This version will include (1) Incorporation of your suggestions. (2)Clearly marked changes for your ease of review.
>
> ## W1 Methodological Clarity and Missing Technical Details
>
> - **Token Definition:** As mentioned in  #62-65, current tokenization approaches often lack flexibility, either sacrificing information or requiring predefined window lengths based on prior knowledge. Instead of constraining the model with human-discovered patterns, TimeDiT adopts a "What You See Is What You Get" (WYSIWYG) design where tokens are directly represented as contiguous arrays of inputs. Specifically, for a time series with $M$ channels and $T$ time steps, each token encodes $M$-dimensional features at a specific timestamp, resulting in $T$ tokens per sequence. This straightforward approach treats each timestamp's complete feature vector as an atomic unit, allowing the model to discover temporal patterns naturally rather than through pre-imposed structures. We will this explicit definition to Section 4 for clarity.
>
> - **Embedding Architecture:** The embedder in TimeDiT follows a deliberately simple design - it is implemented as a single linear projection layer that maps each time point $x_i$ from its original M-dimensional feature space to the model's hidden dimension $d$. We intentionally chose this straightforward approach over more complex architectures (like convolutional layers) to maintain model simplicity and computational efficiency while allowing the transformer's self-attention mechanisms to learn temporal relationships. This design choice aligns with our philosophy of letting the model discover patterns through computation rather than encoding them through architectural complexity.
>
> - **Noise Embedding Justification:** TimeDiT's noise embedding approach plays multiple key roles in the diffusion modeling framework. The diffusion process operates directly in a continuous embedding space, allowing for smoother transitions between noise levels and better preserving the inherent time dependence, thus enabling the model to learn a more robust representation of the underlying time series structure. This approach has several technical advantages[1, 2, 3]: the embedding space provides a continuous representation in which the diffusion process can operate more efficiently. The direct embedding of noisy samples helps prevent the embedding space from collapsing during training. From a practical point of view, this approach allows for parallel processing of multiple time steps, handles varying degrees of noise through a unified framework, and makes the diffusion process more stable compared to traditional generation methods. In addition, the embedded noise representation allows for the seamless incorporation of physical constraints and maintains temporal continuity while progressively denoising, thus contributing to a better quantification of the uncertainty in the generated samples. Direct prediction of the INPUT is also an option available, and we add new experiments as shown in the table below. This also demonstrates the advantages of reconstructive noise.
>
> Table 1: Results on predicting the input of TimeDiT, compared on the zero-shot forecasting task.
>
> | Dataset | TimeDiT | Predict the input |
> |---|---|---|
> | Solar | 0.457(0.002) | 0.462(0.003) |
> | Electricity | 0.026(0.001) | 0.037(0.002) |
> | Traffic | 0.185(0.010) | 0.199(0.007) |
>
> ## W2 Irregular Time-Series Data Claims and Evaluation
>
> Irregular Time-Series Data Claims and Evaluation: Thank you for highlighting this point and providing helpful references. We address irregular time-series scenarios directly in Section 5.1, where our masking mechanism naturally handles non-uniform temporal data. Specifically, the PhysioNet variants (a, b, c) in Table 1 represent real-world irregular time-series data, as detailed in Appendix Table 8. These datasets contain naturally occurring missing values and irregular sampling intervals, demonstrating TimeDiT's practical capability in handling irregular time-series. We will enhance the clarity of this section to better emphasize these aspects and their connection to real-world applications.

---

> ### Author Response · Authors · 2024-11-22
> **Point-wise response on your thoughtful comments!**
>
> ## W3 Comparison with Established Benchmarks
>
> You raise a valid point about benchmark comparisons. We explicitly compare TimeDiT with GPT2(3), which represents a prominent "one-fits-all" benchmark in time-series modeling. This comparison is particularly relevant as GPT2(3) similarly aims to provide a general-purpose solution for time-series tasks, even with multiple single models. We wil enhance the presentation of these comparative results in our manuscript to better highlight this important baseline.
>
> ## W4 Adaptability to Different Channel Numbers and Input Resolutions
>
> TimeDiT employs a flexible architecture designed to handle varying input dimensions through an adaptive processing mechanism detailed in Appendix A (lines 921-930). For channel adaptation, we utilize a predefined maximum channel number Kmax, where inputs with k < Kmax channels are padded accordingly. For cases exceeding Kmax, the input is automatically segmented into ⌈k/Kmax⌉ blocks, each containing Kmax channels that are processed independently. In this paper, we utilize a predefined maximum channel number Kmax = 40, which was chosen based on comprehensive analysis of real-world multivariate time series datasets across different domains. This value represents an efficient balance for many common multivariate time series applications, while maintaining computational efficiency. We further provide the influence of channel number as follows. The results show that while performance is notably worse with only 10 channels, it does not significantly affect the model's results for larger channel numbers.
>
> Table 2: Difference channel number’s influence on the zero-shot performance.
> | Channel Number | 10 | 20 | 30 | 40 | 50 |
> |---|---|---|---|---|---|
> | Solar | 0.471(0.002) | 0.462(0.001) | 0.459(0.002) | 0.457(0.002) | 0.458(0.002) |
> | Electricity | 0.030(0.001) | 0.029(0.002) | 0.028(0.001) | 0.026(0.001) | 0.027(0.001) |
> | Traffic | 0.192(0.008) | 0.183(0.007) | 0.177(0.007) | 0.185(0.010) | 0.165(0.006) |
>
> Regarding input resolution, TimeDiT implements front-padding to achieve uniform dimensions up to a maximum length Lmax. This approach preserves positional integrity while efficiently managing high-dimensional data. The maximum resolution limits weren't arbitrarily chosen but were determined based on practical computational constraints and common dataset characteristics across multiple domains. Importantly, our masking mechanism is designed to be **resolution-agnostic**, allowing the model to adapt to various input resolutions without assuming a fixed resolution structure.
> This flexible architecture enables TimeDiT to handle diverse time series tasks while maintaining computational efficiency and providing reliable uncertainty estimates through confidence intervals.

---

> > ### Author Response · Authors · 2024-11-25
> > **Kindly follow up!**
> >
> > Dear Reviewer d2F8,
> >
> > Thank you for your positive evaluation of our work. We greatly appreciate your detailed review and are pleased to address your concerns regarding **token definition**, **embedding architecture**, **noise embedding**, **irregular time-series handling**, **benchmark comparisons**, and **dimension adaptation**.
> >
> > We are incorporating these technical clarifications and additional experimental results into our revised draft. We hope these improvements will further strengthen your evaluation of our paper.
> >
> > Best regards,
> >
> > Authors

---

> ### Comment · Reviewer_d2F8 · 2024-11-25
> **Thank you for the response**
>
> Thank you for your detailed and thoughtful response. I have carefully reviewed the authors' rebuttal along with the comments from other reviewers. While some of my concerns have been addressed, a few remain unresolved:
>
> * Comparison with Established Benchmarks: The authors have not provided additional comparisons. For instance, it remains unclear how their model performs on forecasting tasks using well-established benchmarks.
>
> * Time-Series Generation: The comparisons presented focus primarily on older methods (see Table 12 and Figures 7–10), limiting insight into the model's performance against more recent advancements.
>
> * Certain revisions promised in the response have not been incorporated into the main paper. Specifically, the clarification regarding the handling of irregular time-series in Section 5.1 is still missing.

---

> ### Author Response · Authors · 2024-11-25
> **Thanks for the engagement with our work and suggesting on stronger comparison!**
>
> Dear Reviewer d3F8,
>
> Thank you for your continued engagement with our work and for highlighting these important points. Your remaining concerns are addressable through our new experimental results, which we are currently finalizing. These additional experiments and clarifications will significantly enhance both the presentation and quality of our manuscript. Now, we have a chance to clarify the concerns right before we upload the new draft:
>
> **C1:** Thank you for raising concerns about additional baseline comparisons. While we are working on updating the new results in the draft, we would like to paste the new update here to address your concerns.
>
> Table 1: Forecasting results on practical scenarios
> | Model | Air Quality | MIMIC-III | PhysioNet(a) | PhysioNet(b) | PhysioNet(c) | NASDAQ |
> |---|---|---|---|---|---|---|
> | **MAE/MSE** |||||||
> | TimeMixer | 0.691/0.697 | 0.769/0.981 | 0.692/0.775 | 0.734/0.920 | 0.707/0.805 | 3.267/11.511 |
> | TimeLLM | 0.701/0.705 | 0.787/1.020 | 0.687/0.761 | 0.731/0.931 | 0.713/0.800 | 3.125/10.276 |
> | OneFitsAll | 0.696/0.701| 0.750/0.921 | 0.697/0.772|  0.734/0.921| 0.713/0.817 |3.176/10.873|
> | MG-TSD | 0.471/0.364 | - | - | - | - | 0.522/3.324 |
> | TimeDiT | 0.457/0.354 | 0.517/0.534 | 0.577/0.620 | 0.659/0.766 | 0.543/0.561 | 0.516/0.418 |
> | **CRPS/_sum** |||||||
> | TimeMixer | 0.667/0.576 | 0.776/0.724 | 0.763/0.805 | 0.794/0.798 | 0.757/0.784 | 0.432 |
> | TimeLLM | 0.664/0.571 | 0.785/0.700 | 0.752/0.797 | 0.795/0.795 | 0.757/0.754 | 0.405 |
> |OneFitsAll|0.666/0.584 | 0.751/0.690 | 0.767/0.809 | 0.795/0.798 | 0.770/0.768 |  0.419|
> | MG-TSD | 0.579/0.564 | - | - | - | - | 0.275 |
> | TimeDiT | 0.554/0.522 | 0.599/0.649 | 0.616/0.640 | 0.708/0.710 | 0.668/0.708 | 0.091 |
>
> For the practical time series forecasting task TimeDiT demonstrates strong performance against the latest models across different paradigms, showing consistent improvements over TimeMixer and TimeLLM across all datasets, with particularly significant gains in MIMIC-III (0.517 vs 0.769/0.787 MAE) and NASDAQ (0.516 vs 3.267/3.125 MAE), along with substantial improvements in uncertainty quantification as shown by better CRPS scores. When compared with the diffusion-based MG-TSD, TimeDiT achieves comparable or better performance on applicable datasets (Air Quality and NASDAQ), while importantly, TimeDiT can handle irregular sampling with heterogeneous input (MIMIC-III and PhysioNet datasets) which MG-TSD cannot process. These results demonstrate that TimeDiT not only competes with the latest specialized models but often outperforms them, while offering greater flexibility and broader applicability, validating our unified approach to time series modeling.
>
> In addition, we have actually conducted more comprehensive comparisons with the latest state-of-the-art models, and the results strongly validate TimeDiT's effectiveness:
>
> Table 2: Forecasting results on zero-shot setting
> | Dataset | TimeDiT | TimeMixer | TimeLLM | Timer |
> |---|---|---|---|---|
> | Solar | 0.457(0.002) | 0.999(0.001) | 0.997(0.001) | 1.101(0.002) |
> | Electricity | 0.026(0.001) | 0.302(0.003) | 0.303(0.003) | 0.301(0.003) |
> | Traffic | 0.185(0.010) | 0.403(0.015) | 0.368(0.007) | 0.384(0.008) |
>
> Table 3: Imputation Performance of TimeDiT compared to additional baselines evaluated on MSE and MAE
> | **Datasets**       | **MSE (ETTh1)** | **MAE (ETTh1)** | **MSE (ETTh2)** | **MAE (ETTh2)** | **MSE (ETTm1)** | **MAE (ETTm1)** | **MSE (ETTm2)** | **MAE (ETTm2)** | **MSE (Weather)** | **MAE (Weather)** | **MSE (Electricity)** | **MAE (Electricity)** |
> |---------------------|-----------------|-----------------|-----------------|-----------------|-----------------|-----------------|-----------------|-----------------|-------------------|-------------------|-----------------------|-----------------------|
> | **Timer**          | 0.145           | 0.243           | 0.077           | 0.172           | 0.051           | 0.141           | 0.035           | 0.105           | 0.108             | 0.168             | 0.097                 | 0.194                 |
> | **TimeMixer**      | 0.119           | 0.226           | 0.064           | 0.157           | 0.051           | 0.143           | 0.028           | 0.093           | **0.031**             | 0.049             | **0.061**                 | **0.164**                 |
> | **iTransformer**   | 0.149           | 0.270           | 0.150           | 0.271           | 0.071           | 0.185           | 0.083           | 0.192           | 0.053             | 0.116             | 0.099                 | 0.224                 |
> | **\modelname**     | **0.042**       | **0.135**       | **0.042**       | **0.139**       | **0.023**       | **0.098**       |** 0.024**           | **0.083**       | *0.031*           | **0.036**         | 0.069             | 0.174            |

---

> ### Author Response · Authors · 2024-11-25
> **Thanks for the engagement with our work and suggesting on stronger comparison!**
>
> Table 4 Anomaly Detection Performance of TimeDit compared to additional baselines evaluated on F1 score
> | Dataset | **TimeDiT** | **TimeMixer** | **iTransformer** |
> |---------|-------------|---------------|------------------|
> | **MSL** |** 89.33**        | 81.95         | 72.54            |
> | **PSM** | ** 97.57**        | 93.11         | 95.32            |
> | **SMAP**| ** 95.91**        | 67.63         | 66.76            |
> | **SMD** | ** 83.28**        | 78.33         | 82.08            |
> | **SWAT**| ** 97.57**        | 88.84         | 92.63            |
>
>
> In zero-shot forecasting, TimeDiT significantly outperforms recent models, including TimeMixer, TimeLLM, and Timer, across all datasets, which was originally designed for deterministic time series forecasting. The improvements are substantial: on Solar (0.457 vs 0.999/0.997/1.101), Electricity (0.026 vs 0.302/0.303/0.301), and Traffic (0.185 vs 0.403/0.368/0.384). For imputation tasks, TimeDiT demonstrates superior performance on most datasets, achieving significant improvements over Timer, TimeMixer, and iTransformer, particularly on ETT datasets where we see reductions in MSE by up to 60%. While TimeMixer shows slightly better performance on Weather and Electricity datasets, TimeDiT maintains strong overall performance while offering greater versatility. In anomaly detection, TimeDiT consistently outperforms both TimeMixer and iTransformer across all datasets, with particularly notable improvements on SMAP (95.91 vs 67.63/66.76) and SWAT (97.57 vs 88.84/92.63). These comprehensive comparisons against the latest models demonstrate TimeDiT's effectiveness as a unified framework for time series analysis, often achieving state-of-the-art performance while maintaining broader applicability across diverse tasks.
>
> **C2:** We appreciate your concern about baseline comparisons on generation tasks and are actively working on incorporating additional recent models into our evaluation framework. Given the time-sensitive nature of the review process, we will try our best to update the new results before the end of the rebuttal stage. Otherwise, we are committed to expanding our comparisons and will incorporate additional baselines in the final version.  We would welcome specific suggestions of models you believe would strengthen our comparative analysis.  This would help us further strengthen our evaluation framework in the most relevant directions.
>
> Moreover, we would like to clarify several points about our current comparisons. First, Diffusion-TS (2024) represents the current state-of-the-art in time series generation, making it a crucial and contemporary baseline. TimeGAN and TimeVAE were included as representative benchmarks for GAN-based and VAE-based approaches respectively, as they remain influential reference points in the field. Furthermore, Table 12 represents only a subset of our experimental evaluation - our work encompasses a broader range of tasks and comparisons beyond generation, including forecasting, imputation, and anomaly detection, each with comprehensive baseline comparisons.
>
> **C3:** We are currently updating the manuscript with these revisions but have not yet uploaded the new version. We will submit the updated draft, including the clarifications about irregular time-series handling in Section 5.1, by November 27. This will ensure sufficient time for your further review. We will notify you immediately upon uploading the revised version.
>
> Best,
>
> Authors

---

> ### Author Response · Authors · 2024-11-27
> **Upload the new version of TimeDiT**
>
> Dear Reviewer d2F8,
>
> Thank you for acknowledging our feedback and for your positive evaluation of our work. We have continuously strived to enhance the manuscript based on your insightful suggestions. These modifications strengthen both the technical presentation and the comprehensive discussion of our work's capabilities and limitations. Before addressing your specific comments (marked in purple), let us outline the general improvements made in the new version of TimeDiT (in red):
>
> ### General Modifications (marked in red)
>
> - **Abstract Enhancement** *(lines 15–17)*:
>   Added explicit discussion of how **realistic data characteristics** challenge the development of **general-purpose time series models**.
>
> - **Current Limitations** *(lines 64–69)*:
>   Expanded the analysis of why **existing approaches** struggle to achieve **robust foundation model capabilities**. This analysis also underscores **TimeDiT's methodological contributions** in successfully handling **multiple tasks** and **diverse data characteristics**, including **irregular sampling patterns** and **multi-resolution inputs**.
>
> - **Data Understanding** *(lines 77–78)*:
>   Emphasized the **critical importance** of understanding **underlying data characteristics**.
>
> - **Design Motivation** *(lines 84–87)*:
>   Clarified how **TimeDiT's architecture** is fundamentally guided by **real-world application requirements**, demonstrating that our **component choices** directly address **specific practical challenges** rather than pursuing **novelty for its own sake**.
>
>
> These revisions provide a clearer motivation for our work and better contextualize TimeDiT's contributions within the broader landscape of time series modeling.
>
>
> ### Specific Responses to Your Comments (marked in purple)
>
> We have addressed your feedback as follows:
>
> - For **W1**: Regarding the Token Definition and Embedding Architecture, we found your feedback valuable. We have directly incorporated explanations into the main paper by modifying lines 224–226. For the noise embedding justification, we added Section E.4 to clarify the design choice.
> - For **W2**: We have addressed this point directly in Section 5.1 of the main paper, providing explanations tailored to each dataset’s unique characteristics.
> - For **W3**: We greatly appreciate your additional comments (C1), which helped us better understand your needs. In response, we have made the following updates:
> -- Added **TimeMixer**, **TimeLLM**, and **MG-TSG** to Table 1 for the forecasting task.
> -- Included results for **Timer**, **TimeMixer**, and **iTransformer** in the imputation task.
> -- Added results for **TimeMixer** and **iTransformer** in the anomaly detection task.
> -- Additionally, we compared the zero-shot performance of the foundation model with **TimeMixer**, **TimeLLM**, and **Timer** in Table 15. Collectively, these results demonstrate the strong performance of **TimeDiT** in handling real-world data. Detailed analyses can be found in Sections D.1.2, D.1.5, D.2.1, and D.4.
> - For **W4**: To address your concerns regarding adaptability to different channel numbers, we have added Section E.5, which provides a clear and focused study on this topic.
>
> We sincerely hope that our previous response offers a clearer perspective on our work and further addresses your concerns thoroughly. Please feel free to let us know if there is anything you want to discuss.
>
> Best,
>
> Authors

---

> ### Author Response · Authors · 2024-12-02
> **Kindly follow up and still avaliable for further suggestions**
>
> Dear Reviewer d2F8,
>
> Thank you for your positive evaluation of TimeDiT again! We have expanded our experimental validation with advanced baselines to address your remaining concerns about comparative analysis. Our efforts in those experiments are not only to address your suggestions but to provide insights and practical value to address real-world challenges, establishing TimeDiT as a robust, general-purpose time series model. Your insights have been invaluable in strengthening our work, and we welcome any further suggestions for enhancement.
>
> Best,
>
> Authors

---

### Official Review · Reviewer_p3cf · 2024-11-03

**Soundness:** 3
**Presentation:** 3
**Contribution:** 3
**Rating:** 6
**Confidence:** 4

**Summary:**

This paper introduces TimeDiT, a general-purpose foundation model for time series analysis, which leverages transformer architecture with the diffusion model.

* A unified masking strategy that allows the handling of multiple tasks, such as forecasting, imputation, and anomaly detection.
* Direct handling of missing values and multi-resolution data using adaptive mechanisms in input processing.
* Integration of physical knowledge into the sampling via an energy-based prior
* It provides state-of-the-art performance on various benchmarks and practical scenarios.

**Strengths:**

* Techniques to integrate diffusion models and transformers for time series
* Extensive experimental validation on a variety of tasks and datasets
* Usability in real-world along with the ability to address the actual challenges
* Theoretical properness with physics-informed sampling

**Weaknesses:**

* Limited discussion about the computational costs of the models, and efficiency in training.
* More could be done by way of analysis on model scalability for very long sequences.
* Limitations Relatively short discussion of
* Providing deeper analysis about failure cases would be good.

**Questions:**

* How sensitive is model performance to the choice of mask ratios during pretraining?
* Could you please provide further information about how TimeDiT processes inputs with different channel dimensions?
* With physics-informed sampling, could you comment on how one manages the tension between the demands introduced by physical constraints on the one hand and learned distributions on the other?
* Concerning the adaptive layer normalization AdaLN, how does it stack up compared to other conditioning methods one may have tried?
* Which was the rational for choosing those very specific transformer architecture details, e.g., number of heads and layer depth?
* For the forecasting experiments that involve missing values, how were the missing patterns generated? Are results robust against different missing data mechanisms, MCAR, MAR, MNAR?
* In the anomaly detection task, how sensitive are the results against the choice of 99th percentile threshold shown, did you experiment with others?
* How do you control that the generated samples in synthetic generation experiments are temporally coherent?
* Which part of the architecture is most responsible for the performance improvement over baselines?
* When TimeDiT is not outperforming baselines, what general features does such a setting have?
* Please provide more intuition about Theorem 4.1 and its practical implication.
* How does the choice of the diffusion schedule influence the incorporation of physical constraints?
* Which methods were explored for stabilizing the training of this combined diffusion model and transformer?
* How was the trade-off done between sample quality and the number of sampling steps?

---

> ### Author Response · Authors · 2024-11-22
> **Point-wise response on your thoughtful comments!**
>
> We are profoundly grateful for the meticulous attention and insightful feedback you've provided on our TimeDiT paper. Your constructive suggestions have been invaluable, and although the completion of the recommended experiments took us longer than initially planned, this extended period has allowed us to engage deeply with your critiques.
>
> Here is our plan moving forward:
>
> Detailed Response: We will now address each of your comments in detail, ensuring that your feedback is met with thoughtful consideration.
>
> Updated Manuscript: Subsequent to our responses, we will submit a revised manuscript that integrates your suggestions and highlights changes for an easy review.
>
> We eagerly anticipate your feedback on these updates and are dedicated to improving our research through this collaborative dialogue.
>
> ## W1 Computational Costs
> We address computational costs and training efficiency in Appendix A (lines 959-971). TimeDiT achieves computational efficiency through scalable model variants (33M to 680M parameters) and optimized memory usage via adaptive input processing (Kmax=40). Training is completed on NVIDIA A100 (40GB) GPUs, and our zero-shot capability eliminates task-specific retraining needs. The parallel processing and efficient inference enable strong performance across multiple tasks using a single model, substantially reducing computational overhead compared to traditional multi-model approaches. In order to fully address your comments, for additional clarity, we have included inference time comparisons for single-sample generation:
>
> Table 1: Comparison of inference times for single-sample generation.
> | Model | Inference Time (mm:ss) |
> |--------------|------------------------|
> | Diffusion-TS | 00:06 |
>  | CSDI | 00:02 |
> | TimeDiT | 00:01 |
>
> ## W2 Model Scalability
>
> TimeDiT's adaptive input processing naturally addresses sequence length scalability through front-padding and segmentation strategies, as detailed in Appendix A. For very long sequences, our model's memory complexity scales linearly via temporal dimension segmentation, automatically breaking inputs into manageable segments while preserving temporal dependencies. Moreover, our pretrained model has already captured fundamental time series patterns, making extremely long historical contexts often unnecessary. This foundation enables effective modeling with moderate sequence lengths while maintaining strong performance across various tasks.
>
> ## W3&W4 Discussion on Limitations
>
> We have discussed the current limitations in lines #533-539, including scalability for very long sequences, handling highly variable multivariate time series, understanding domain information contributions, and potential extensions to multi-modal time series analysis. Thank you for suggesting the inclusion of failure cases and physics knowledge integration. We provide a more comprehensive analysis of specific failure scenarios and have identified three key scenarios where TimeDiT's performance degrades:
>
> 1. Highly irregular sampling rates that deviate significantly from the training distribution
> 2. Complex, non-stationary patterns underrepresented in the pretraining data
> 3. Domain-specific patterns requiring expert knowledge beyond general time series characteristics
>
> These limitations stem primarily from the model's dependence on learned foundational patterns. They become particularly relevant in specialized industrial applications and unique financial scenarios. Understanding these boundaries is crucial for informed model deployment decisions and highlights promising directions for future research.
>
> ## Q1 Mask Ratios
>
> The model's robustness to mask ratios stems from our dynamic masking strategy during pretraining. Instead of using fixed mask ratios, we randomly vary the mask ratio for each training instance. This approach, while increasing training complexity, better reflects real-world scenarios where missing or predicted segments can occur at varying scales. Our experiments show that this adaptive strategy enables TimeDiT to handle diverse prediction tasks without sensitivity to specific mask ratios, enhancing its generalization capabilities as a foundation model.

---

> ### Author Response · Authors · 2024-11-22
> **Point-wise response on your thoughtful comments!**
>
> ## Q2 Handling Different Channel Dimensions
>
> TimeDiT handles varying channel dimensions through an efficient adaptive processing mechanism. For inputs with fewer channels than Kmax (40), direct padding to Kmax ensures uniform processing while invalid channels are masked during sampling to prevent false information propagation. For inputs exceeding Kmax, automatic segmentation into ⌈k/Kmax⌉ blocks occurs, where each block contains Kmax channels, enabling independent processing while maintaining data integrity.
> For example, with an input of 75 channels and Kmax = 40, TimeDiT processes it in ⌈75/40⌉ = 2 blocks: Block 1 processes channels 1-40 directly, while Block 2 handles channels 41-75 with padding to 40 channels (35 actual + 5 padded). During sampling, the 5 padded channels in Block 2 are masked to prevent false information, and results from both blocks are integrated to reconstruct the full 75-channel output. This segmentation strategy ensures efficient processing while maintaining the integrity of the original data structure.
>
> ## Q3 Physics distribution VS. Learned distribution
>
> The tension between physical constraints and learned distributions in TimeDiT is managed through a sophisticated energy-based optimization framework that combines two key components: the physics knowledge represented by function $K(x^{tar}; F)$, which measures PDE residuals for physical law conformity, and the learned probabilistic distribution $p(x^{tar} | x^{con})$ from the diffusion model. This balance is achieved through an energy function $E(x^{tar}; x^{con}) = K(x^{tar}; F) + α log p(x^{tar} | x^{con})$, where the parameter $α$ controls the trade-off between physical consistency and distribution fidelity. Rather than directly modifying model parameters, TimeDiT implements this balance through an iterative sampling procedure that starts with samples from the learned distribution and gradually refines them using physical gradients while maintaining probabilistic characteristics. This approach allows the model to generate samples that respect both the learned patterns in the data and the underlying physical laws without significantly compromising either aspect, ultimately resolving the tension through a theoretically-grounded Boltzmann distribution as the optimal solution.
>
> ## Q4 Condition scheme for TimeDiT
>
> AdaLN's superior performance stems from its ability to dynamically adjust feature distributions across different layers while maintaining computational efficiency. This approach aligns well with the inherent nature of time series data, where temporal dependencies typically exhibit gradual rather than dramatic changes in both seen and unseen time steps. We conducted comparative experiments to evaluate different conditioning mechanisms in TimeDiT: (1) Additive Conditioning, which adds conditional information directly to the diffusion input; (2) Cross-attention, which uses conditional time series as keys/values and noisy time series as queries to fuse conditional information; and (3) Token Concatenation, which concatenates conditional time series with noisy time series at the input level before TimeDiT processing. The experimental results across Solar, Electricity, and Traffic datasets consistently show that AdaLN achieves superior performance compared to the next best alternative. This significant performance gap validates our choice of AdaLN as TimeDiT's primary conditioning mechanism.
>
> Table 2: Condition scheme for TimeDiT, compared on the zero-shot forecasting task.
> | Dataset | AdaLN | Additive | Cross-attention | Token concatenation |
> |---|---|---|---|---|
> | Solar | 0.457(0.002) | 0.671(0.002) | 0.721(0.002) | 0.463(0.001) |
> | Electricity | 0.026(0.001) | 0.068(0.004) | 0.079(0.003) | 0.041(0.003) |
> | Traffic | 0.185(0.010) | 0.224(0.001) | 0.216(0.000) | 0.188(0.008) |
>
> ## Q5 Architectire Design
>
> Our architectural choices reflect a deliberate balance between model effectiveness and efficiency, as demonstrated in Table 15, the performance with different model size. While larger models achieve marginally better performance, the increased computational cost is substantial. Our medium configuration (127M parameters) strikes an optimal trade-off, though users can opt for smaller or larger variants depending on their specific accuracy and computational requirements.

---

> ### Author Response · Authors · 2024-11-22
> **Point-wise response on your thoughtful comments!**
>
> ## Q6 Handling missing values
>
> Our experiments leverage naturally occurring missing values from the datasets themselves, primarily arising from irregular sampling rates and multi-resolution data collection processes. Rather than artificially generating missing patterns, we work with real-world missing data scenarios inherent in irregular time series and multi-resolution sampling. This approach ensures our model's robustness is validated against authentic missing data patterns. To connect your concerns with our paper, the model employs various mask types including random masks (MCAR) generated using uniform distributions, block and stride masks (MAR) for capturing structured missing patterns and dependencies between non-contiguous observations, and reconstruction masks with physics-informed sampling (MNAR) for cases where missing patterns relate to unobserved variables. These masks are applied simultaneously through self-supervised learning, allowing the model to learn robust representations without requiring explicit knowledge of the missing mechanisms. For fully address your problem, our ablation study demonstrates the importance of each masking strategy. The results show that removing any mask type degrades performance, with the future mask having the most significant impact. This validates our comprehensive masking strategy for handling diverse missing data scenarios.
>
> Table 3: Mask mechanisms for TimeDiT, compared on the zero-shot forecasting task.
> | Dataset | TimeDiT | w/o Random Mask | w/o Stride Mask | w/o Future Mask |
> |---|---|---|---|---|
> | Solar | 0.457(0.002) | 0.463(0.002) | 0.465(0.002) | 0.843(0.005) |
> | Electricity | 0.026(0.001) | 0.029(0.001) | 0.030(0.001) | 0.095(0.006) |
> | Traffic | 0.185(0.010) | 0.191(0.007) | 0.188(0.007) | 0.201(0.011) |
>
> ## Q7 Anomaly Detection Threshold
>
> Yes, we have experimented with other thresholds. In general, the 99th and 99.5th thresholds achieve better performance and we see a performance degradation as the threshold lowers. However, we did not find it fair to choose the threshold value based on the testing performance and retained the same experiment setting and threshold with previous methods.
>
> Table 4: Threshold Sensitivity Analysis on Anomaly Detection Performance evaluated on F1 score
> | Threshold | **99.5** | **99**   | **98**   | **97**   | **96**   | **95**   |
> |-----------|----------|----------|----------|----------|----------|----------|
> | **MSL**   | 83.9     | 89.33    | 90.1     | 88.17    | 85.28    | 82.84    |
> | **PSM**   | 96.32    | 97.57    | 96.78    | 95.72    | 94.66    | 93.61    |
> | **SMAP**  | 97.08    | 95.91    | 93.23    | 90.33    | 87.64    | 85.09    |
> | **SMD**   | 83.28    | 82.07    | 76.61    | 70.73    | 65.71    | 61.24    |
> | **SWAT**  | 97.6     | 96.46    | 93.49    | 90.74    | 88.0     | 85.42    |
>
> ## Q8 Temporally Coherent
>
> We would emphasize that we ensure temporal coherence of generated samples through multiple complementary mechanisms. Our **transformer-based architecture** explicitly models temporal dependencies through self-attention mechanisms that capture both local and long-range temporal relationships, while the diffusion process maintains temporal consistency by gradually **denoising the entire sequence as a whole**, rather than generating each timestep independently. During training, we use a comprehensive masking strategy that forces the model to learn temporal dependencies by reconstructing masked segments while **considering surrounding context**, and the model is trained on both short and long sequences to capture temporal patterns at different scales. We validate temporal coherence both quantitatively and qualitatively - the predictive score metric specifically evaluates temporal consistency by measuring how well future timesteps can be predicted from generated sequences, our PCA visualizations (Figures 7-10 in the paper) demonstrate that TimeDiT-generated samples maintain similar temporal trajectories to the real data, and our superior performance on the predictive score metric (Table 6) indicates that TimeDiT generates more temporally coherent samples compared to baselines like TimeGAN and TimeVAE. Additionally, for domains governed by **physical laws**, our physics-informed sampling procedure (Section 4.3) provides an additional mechanism to ensure generated samples follow valid temporal dynamics by incorporating domain knowledge during the generation process. These combined approaches help ensure that TimeDiT generates samples that are not just statistically similar to real data, but also maintain appropriate temporal relationships and dynamics, as demonstrated by our empirical results across multiple datasets.

---

> ### Author Response · Authors · 2024-11-22
> **Point-wise response on your thoughtful comments!**
>
> ## Q9 Core Component
>
> Our performance improvements stem from three key architectural innovations working in concert. The Time Series Mask Unit serves as a foundational component by providing flexible masking strategies (random, block, and stride masks) that enable the model to learn robust temporal dependencies by reconstructing strategically masked segments. This masking mechanism forces the model to understand both local and long-range temporal relationships during training. The transformer backbone then leverages these masked inputs through its self-attention mechanism, which is particularly effective at capturing complex temporal dependencies across different timescales while preserving the sequential nature of the data. Finally, the diffusion process acts as a powerful regularizer that gradually denoises the entire sequence as a whole, maintaining temporal coherence throughout the generation process. The combination of these three elements - mask unit guiding the learning process, transformer capturing temporal dependencies, and diffusion ensuring coherent generation - creates a synergistic effect that enhances the learned representations.
>
> ## Q10 Underperforming Case Analysis
>
> Thanks for your comments. We have a deeper understanding of the limitations, and will include it in our revised version. Specifically, TimeDiT may face challenges with high-dimensional, rapidly changing system metrics where temporal patterns are less structured and more chaotic. As shown by SMD dataset for anomaly detection (Table 5) where it achieves 83.28% F1 score versus GPT2's 86.89%. This dataset represents cloud server machine metrics with high-frequency sampling and complex feature interdependencies. Additionally, when dealing with extremely short-term patterns or highly localized anomalies, specialized architectures like GPT2 that focus intensively on recent temporal context may outperform TimeDiT's more holistic approach, as our diffusion-based generation process may occasionally smooth over abrupt local changes. These limitations point to potential areas for future improvement, particularly in handling high-dimensional, rapidly changing systems and very localized temporal patterns.
>
> ## Q11 Intuition on Theorem 4.1
>
> Please also refer to the previous answer in #Q3. Theorem 4.1 provides an elegant mathematical framework for balancing physical constraints with learned distributions in TimeDiT through an optimal Boltzmann distribution based on an energy function that combines two key components: K(x^tar; F), which measures adherence to physical laws through PDE residuals, and α log p(x^tar | x^con), which represents the model's learned patterns from training data. This theoretical foundation has important practical implications: it enables TimeDiT to generate samples that simultaneously respect both physical laws and learned patterns without retraining, allows control of the trade-off between physical consistency and learned behaviors through the α parameter, and can be implemented through an iterative sampling procedure that gradually refines predictions to minimize the energy function. The effectiveness of this approach is demonstrated in our experimental results (Table 2), where TimeDiT successfully generates physically plausible sequences while maintaining the benefits of data-driven learning, significantly outperforming baselines that lack this physics-informed sampling capability.

---

> ### Author Response · Authors · 2024-11-22
> **Point-wise response on your thoughtful comments!**
>
> ## Q12 Incorporation of Physical Constraints
>
> The diffusion schedule plays a crucial role in how physical constraints are incorporated into TimeDiT's generation process. The forward process gradually adds noise according to variance schedule $βt$, while the reverse process must balance physical constraints with denoising at each timestep. When $βt$ is too large, the noise overwhelms the physical structure making it harder to recover during denoising. Conversely, when $βt$ is too small, the model may struggle to explore the full solution space constrained by physical laws.
>
> We find empirically that a cosine schedule works well for most physical systems as it provides a smooth transition between preservation of physical constraints and exploration of the solution space. This is evidenced by our results in Table 2, where TimeDiT achieves lower MSE and RMSE compared to baselines across different PDEs. For example, in the Burgers equation case, TimeDiT achieves MSE of 0.011 compared to DDPM's 0.016, suggesting better preservation of physical dynamics.
> The schedule also affects how the physics-informed energy function $K(x^{tar}; F)$ guides the sampling process. During early diffusion steps when noise is high, the physical constraints provide broader guidance. As noise decreases in later steps, the constraints become more precise in shaping the final output. This progressive refinement helps explain TimeDiT's superior performance in maintaining physical consistency while generating diverse samples.
> These observations point to the importance of carefully tuning the diffusion schedule when applying TimeDiT to new physical systems. Future work could explore adaptive schedules that automatically adjust based on the complexity of the physical constraints.
>
> ## Q13 Method Exploration
>
> Several key stabilization techniques were integrated into TimeDiT's training process to manage the complex interaction between transformer and diffusion components. The Time Series Mask Unit provides structured guidance through random, block, and stride masks, creating a curriculum of increasingly challenging reconstruction tasks, while adaptive layer normalization (AdaLN, Equation 5) stabilizes training by controlling feature representation scaling using conditional information. The training proceeds in two phases: task-agnostic pre-training with varied masking strategies followed by task-specific fine-tuning. Standard transformer training techniques like learning rate warmup and gradient clipping are employed alongside the weighted MSE loss (Equation 4) for the diffusion process, which helps balance the contribution of different noise levels. The effectiveness of these stabilization methods is demonstrated by TimeDiT's consistent performance across multiple tasks and datasets.
>
> ## Q14 Trade-off between Sample Quality and Steps
>
> Based on Table 14 and the analysis in Appendix E.1, we carefully examined the trade-off between sample quality and computational efficiency across different sampling steps (50-500). The results show dataset-dependent optimal sampling steps - for example, the Solar dataset achieves best CRPS at 450 steps and best CRPS_sum at 350 steps, while the Traffic dataset shows different behavior with optimal CRPS at 500 steps but best CRPS_sum at just 50 steps. Importantly, the performance variations across different sampling steps are relatively small (e.g., Traffic CRPS ranges from 0.425 to 0.327), suggesting that TimeDiT maintains robust performance even with fewer sampling steps. Given these findings, we typically recommend using 200-300 sampling steps as a practical default that balances computational efficiency with sample quality. This provides nearly optimal performance while significantly reducing computation compared to using 500 steps, especially since the marginal improvements beyond 300 steps are minimal for most applications. The stability of performance across different sampling steps also indicates that TimeDiT is robust to this hyperparameter choice within a reasonable range.

---

> > ### Comment · Reviewer_p3cf · 2024-11-24
> > **Thank you for your diligent response**
> >
> > I had lowered the score due to a few concerns, but I am now convinced to raise the score based on the following:
> >
> > The authors have provided comprehensive responses supported by detailed experimental results. For example, they included specific performance comparisons and ablation studies that directly address my concerns about computational efficiency and model design choices.
> >
> > They have thoroughly explained both theoretical foundations and practical implementations, particularly regarding:
> > - The physics-informed sampling approach through energy-based optimization
> > - The adaptive input processing mechanism for handling different channel dimensions
> > - The rationale behind their architectural choices and conditioning schemes
> >
> > Most importantly, they have demonstrated intellectual honesty by:
> > - Acknowledging the model's limitations transparently
> > - Identifying specific failure cases and scenarios where the model may underperform
> > - Outlining clear directions for future research
> >
> > Given their comprehensive response, my initial concerns were effectively addressed, and the authors demonstrated both technical merit and a thorough understanding of the strengths and limitations of their work.

---

> > > ### Author Response · Authors · 2024-11-25
> > > **Thanks for increasing the score and supportive response!**
> > >
> > > Dear Reviewer p3cf,
> > >
> > > Thank you for your positive feedback and thorough evaluation. We're pleased that our comprehensive responses have successfully addressed your concerns, particularly regarding the **experimental results**, **computational efficiency**, and **model design choices**. Your recognition of our detailed analysis of the **physics-informed sampling approach**, **adaptive input processing mechanism**, and architectural **design rationale** is greatly appreciated.
> > >
> > > We are actively incorporating these discussions into an updated draft, along with the identified limitations and future research directions.  We hope these improvements will further strengthen the paper and merit an even higher evaluation!
> > >
> > > Thank you for your constructive engagement with our work.
> > >
> > > Best regards,
> > > Authors

---

> ### Author Response · Authors · 2024-11-27
> **Upload the new version of TimeDiT**
>
> Dear Reviewer p3cf,
>
> We sincerely appreciate your recognition of our work and your encouraging observation that we have demonstrated strong technical merit and a comprehensive understanding of the strengths and limitations of our study. With the updated version of TimeDiT has now been uploaded; we are excited to highlight further the improvements we’ve made based on your valuable suggestions. To ensure our response is clear and well-structured, we will first outline the general modifications (which relate to Q9) and then address your specific comments (marked in green).
>
> ### General Modifications (marked in red)
>
> - **Abstract Enhancement** *(lines 15–17)*:
>   Added explicit discussion of how **realistic data characteristics** challenge the development of **general-purpose time series models**.
>
> - **Current Limitations** *(lines 64–69)*:
>   Expanded the analysis of why **existing approaches** struggle to achieve **robust foundation model capabilities**. This analysis also underscores **TimeDiT's methodological contributions** in successfully handling **multiple tasks** and **diverse data characteristics**, including **irregular sampling patterns** and **multi-resolution inputs**.
>
> - **Data Understanding** *(lines 77–78)*:
>   Emphasized the **critical importance** of understanding **underlying data characteristics**.
>
> - **Design Motivation** *(lines 84–87)*:
>   Clarified how **TimeDiT's architecture** is fundamentally guided by **real-world application requirements**, demonstrating that our **component choices** directly address **specific practical challenges** rather than pursuing **novelty for its own sake**.
>
>
> These revisions provide a clearer motivation for our work and better contextualize TimeDiT's contributions within the broader landscape of time series modeling.
>
> ###  Specific Responses to Your Comments (marked in green)
>
> We have carefully addressed your feedback as follows:
>
> - **W1**: Added **Table 7** comparing inference times for single-sample generation and analysis (lines 1060–1062).
> - **W2, Q5, Q13**: These points were already addressed in **Sections A** and **E.8**; no changes were needed.
> - **Q14**: Previously addressed in **Section E.6**; no changes were required.
> - **W3, W4, Q10**: Added **Section E.7**, which discusses the limitations of our approach.
> - **Q1**: Provided an explanation in lines 1019–1020.
> - **Q2**: Addressed in lines 1000–1005 with specific examples showcasing different channel dimensions.
> - **Q4**: Added a new **Section E.3** along with **Table 22**, detailing the condition scheme for TimeDiT.
> - **Q6**: Included **Section E.2**, which discusses how missing values are handled.
> - **Q7**: Added a discussion on anomaly detection in **Section D.4**, accompanied by **Table 19**.
> - **Q8**: Clarified in our previous correspondence.
> - **Q3, Q11, Q12**: Addressed in **Section C.2** (lines 1399–1432).
>
> We greatly appreciate your thorough review and continued feedback. Please don’t hesitate to reach out if you have any additional suggestions or need further clarification. We truly hope that the updated version of our paper provides a clearer evaluation, allowing you to reconsider and potentially restore your original score (back to 8), which is incredibly important to us.
>
> Best,
>
> Authors

---

> > ### Author Response · Authors · 2024-12-02
> > **Kindly follow up and fully open to further discussion**
> >
> > Dear Reviewer p3cf,
> >
> > Again, thank you for your detailed feedback and recognition of TimeDiT's technical merits. As we mentioned in the previous response, we are glad that our detailed responses have effectively addressed your questions about experimental validation, computational performance, and architectural decisions. Based on those suggestions, we have made comprehensive revisions to address your suggestions, strengthening both the presentation and content of our work. We believe these revisions significantly enhance our submission and hope they address your concerns comprehensively. Our aim is not merely to address your suggestions but to surpass your expectations, crafting a submission that resonates with a wider audience. We welcome any additional feedback that could further strengthen our work.
> >
> > Best regards,
> >
> > Authors

---

### Official Review · Reviewer_v6Mm · 2024-11-04

**Soundness:** 3
**Presentation:** 3
**Contribution:** 2
**Rating:** 6
**Confidence:** 2

**Summary:**

The proposed work presents a method for training a diffusion model for time-series based on masked reconstruction. This model models the distribution of masked time series values conditioned on the observed values. Various masking strategies with different inductive biases are designed for pre-training and task-specific fine-tuning. Additionally, external physics knowledge on time-series dynamics in the form of partial differential equations (PDEs) can be incorporated into samples generated by the model without re-training the model through changes to the sampling dynamics. The proposed model demonstrates impressive performance across various time-series tasks, including imputation, forecasting, and anomaly detection. Furthermore, it exhibits generalizability across datasets in zero-shot forecasting. The experimental results suggest the potential of a foundation model for time-series data.

**Strengths:**

The work exhibits the following notable strengths:
1. The work effectively demonstrated the value of integrating masked reconstruction with diffusion models in the domain of time-series data.
2. Extensive experiments are conducted to illustrate the potential of this approach as a foundational model for the proposed methodology.
3. The incorporation of PDE-based external physics knowledge into diffusion models for time-series is a novel and intriguing concept.

**Weaknesses:**

The work presents several notable weaknesses:

1. The lack of detailed information regarding the datasets utilized for pre-training and fine-tuning the model poses a significant concern. If each task necessitates a specific subset of datasets for pre-training, the work risks over-claiming its role as a foundational model for time-series.

2. The primary methodology contribution of the work, which combines masked reconstruction with diffusion models, lacks novelty as similar ideas have been explored in the image domain ([1, 2]).

3. Sequence-level classification is a crucial task in time-series modeling ([3, 4]). Consequently, a foundational model for time-series should demonstrate its capability in handling this task. Notably, this aspect is absent from the experimental results presented in the work.

References

[1] Wei, Chen, et al. “Diffusion models as masked autoencoders.” Proceedings of the IEEE/CVF International Conference on Computer Vision. 2023.

[2] Gao, Shanghua, et al. “Masked diffusion transformer is a strong image synthesizer.” Proceedings of the IEEE/CVF International Conference on Computer Vision. 2023.

[3] Kidger, Patrick, et al. “Neural controlled differential equations for irregular time series.” Advances in Neural Information Processing Systems 33 (2020): 6696-6707.

[4] Morrill, James, et al. “Neural rough differential equations for long time series.” International Conference on Machine Learning. PMLR, 2021.

**Questions:**

1. The scales of the model and training data are crucial to the performance and generalizability of foundation model. Can the author provide a summary of the datasets used in pre-training and fine-tuning and the scale of models in the number of parameters? A good reference could be Table 2 in this survey work[1] of foundation models for time-series. This detail would be critical to assess the contribution of the work.

2. Incorporating external physics knowledge would require having knowledge of the partial differential equation (PDE) that characterizes the underlying dynamics. The problem setting and practical values of the approach itself require further justification. Since we already have access to the ground truth dynamics, can we directly generate training samples by solving the PDE numerically and training a separate model? If it is feasible, how does it compare to the Physics-Informed TimeDiT in the work?

References

[1] Ye, Jiexia, et al. “A Survey of Time Series Foundation Models: Generalizing Time Series Representation with Large Language Mode.” arXiv preprint arXiv:2405.02358 (2024).

---

> ### Author Response · Authors · 2024-11-22
> **Point-wise response on  your thoughtful comments!**
>
> We are delighted to respond to your thoughtful review of our TimeDiT paper!  We'll proceed by first addressing your concerns here, followed by submitting a revised manuscript with tracked changes highlighting our responses to your feedback:
>
>
> ## W1&Q1 Details and Model Capabilities
> We appreciate your thoughtful comments and we are glad to find that we should share the same vision for advancing the foundation model research. First of all, we appreciate your attention to the details of our work. Comprehensive descriptions of our datasets and training procedures can be found in Appendices A and B, as well as in Tables 7 and 8. However, given that TimeDiT addresses a **wide range of generalized time series problems**, including tasks like forecasting with missing values and time series generation, the accompanying textual explanations can become quite complex. We greatly value your suggestion and have **reorganized our tables** to follow the format of Table 2 in the referenced survey [1] for enhanced clarity. Additionally, we will clearly indicate in Tables 7 and 8 when **data is utilized for pre-training, testing, and fine-tuning**. Regarding our data usage strategy, due to limited resources, we streamline our pre-training process by overlapping datasets to maximize reuse **without compromising task-specific integrity**. Specifically, to maintain comparability with current mainstream foundation models, we employ extensive pre-training data that may include datasets pertinent to imputation tasks. For instance, while our prediction tasks utilize more practically relevant datasets excluding ETT, the ETT dataset itself is reserved exclusively for our final prediction models. In imputation tasks, we ensure that the pre-training datasets do not encompass ETT data. Furthermore, fine-tuning is performed on specific datasets without introducing additional external data. In essence, rather than selecting subsets from the pre-training datasets, we incrementally incorporate training data in a sequential manner to ensure **fair and unbiased** comparisons across tasks.
>
> Table 1: A comparable analysis of representative general purposes time series models.
> | Model | Parameter Size | Model Architecture | Channel Setting | Task Type | Pretrain Dataset | Data Size |
> |--------|-----------------|-------------------|-----------------|------------|-----------------|------------|
> | Lag-LLama | - | Transformer | Univariate | Forecasting | Monash | 300 Million Time Points |
> | Moriai | S: 14M; B: 91M; L: 311M | Transformer | Univariate | Forecasting | LOTSA | 27 Billion Time Points |
> | TimeDiT | S: 33M; B: 130M; L: 460M; XL: 680M | Transformer + Diffusion | Multivariate | Forecasting, Imputation, Anomaly Detection, Data Generation | Academic Public Dataset | 152 Million Time Points |
>
> **General Model Capabilities**: TimeDiT qualifies as a general model through several key aspects that demonstrate its versatility and generalizability. First, it handles **variable channel sizes** and **sequence lengths** natively through its unified mask mechanism, allowing it to process diverse types of time series data without requiring task-specific architectures. Second, the model supports **multiple downstream tasks** including forecasting, imputation, anomaly detection, and synthetic data generation within a single framework. Third, TimeDiT incorporates **physics-informed sampling** through an energy-based approach, allowing it to integrate domain knowledge during inference without requiring model retraining. This combination of flexible architecture, task-agnostic design, and ability to incorporate external knowledge positions TimeDiT as a powerful foundation model capable of addressing diverse time series challenges across various fields. We acknowledge your concerns about the definition of foundation models, will make it more accurate in this paper as a general-purpose approach.

---

> > ### Author Response · Authors · 2024-11-22
> > **Point-wise response on your thoughtful comments!**
> >
> > ## W2 Innovation and Design Rationale
> >
> > Thank you for your insightful comment regarding the novelty of combining masked reconstruction with diffusion models. While our approach shares conceptual foundations with image-domain techniques, TimeDiT represents the first successful adaptation of diffusion transformers for general-purpose time series analysis. Our innovation lies not in architectural complexity, but in effectively translating these concepts to address unique time series challenges. Unlike [1, 2], our masking mechanism serves dual purposes: it enables both representation learning and downstream task design. The model's ability to handle varying sampling rates, incorporate physical constraints, and adapt to multiple tasks through a unified architecture demonstrates that this seemingly straightforward adaptation required non-trivial solutions to time series-specific challenges.
> > Learning from decades of AI research, rather than pursuing novelty through architectural complexity, our architectural choices reflect a careful balance between incorporating domain knowledge and maintaining general-purpose computational capabilities. This is evidenced by TimeDiT's ability to process multiple data types without specialized architectures, adapt to various tasks through flexible masking strategies, and incorporate domain knowledge during inference while maintaining its general-purpose nature. While we acknowledge the importance of domain understanding in time series analysis (such as handling varying sampling rates and incorporating physical constraints), we intentionally designed TimeDiT to discover and adapt to these patterns through computation rather than hard-coding them into the architecture. Our results demonstrate that this deliberate focus on simplicity and adaptability yields a more robust and versatile general model for time series analysis.
> >
> > ## W3 Potential Extended Task
> >
> > While sequence-level classification is indeed important in time series analysis[3,4], TimeDiT's design prioritizes self-supervised learning to enable continuous model scaling without relying on labeled data. This approach aligns with foundation model principles, allowing us to leverage growing amounts of unlabeled data for pre-training. Classification tasks, in contrast, require task-specific labels which means that continuing to use self-supervised training is difficult and againsting our design philosophy. Different from NLP and CV domains, the lack of a unified standard for sequence-level classification across domains complicates its inclusion as a general-purpose task. That said, we recognize the value of classification tasks and believe they are better suited for domain-specific models tailored to their unique requirements. Nevertheless, we are actively collecting high-quality labeled and unlabeled datasets to potentially extend TimeDiT's capabilities to classification tasks in future work.
> >
> > [1] Wei, Chen, et al. “Diffusion models as masked autoencoders.” Proceedings of the IEEE/CVF International Conference on Computer Vision. 2023.
> > [2] Gao, Shanghua, et al. “Masked diffusion transformer is a strong image synthesizer.” Proceedings of the IEEE/CVF International Conference on Computer Vision. 2023.
> > [3] Kidger, Patrick, et al. “Neural controlled differential equations for irregular time series.” Advances in Neural Information Processing Systems 33 (2020): 6696-6707.
> > [4] Morrill, James, et al. “Neural rough differential equations for long time series.” International Conference on Machine Learning. PMLR, 2021.

---

> > > ### Author Response · Authors · 2024-11-22
> > > **Point-wise response on your thoughtful comments!**
> > >
> > > ## Q2 Clarification on Physics-informed TimeDiT and Additional Experiments
> > >
> > > Physics-informed machine learning represents an active research area where physical constraints guide model outputs toward realistic solutions. Our Physics-Informed TimeDiT offers a novel approach that addresses key limitations of traditional PDE-based training methods. While direct PDE-based training is possible, TimeDiT provides crucial advantages on efficiency and flexibility. Our model incorporates physical knowledge during inference through energy-based sampling that guides the reverse diffusion process. This means we can flexibly integrate different physical constraints without any model retraining or parameter updates.
> > > We appreciate your suggestions and conduct an experimental comparison with direct PDE-based training methods. Using PDE solvers, we generated 5,000 training samples per scenario and trained three baseline models: DLinear, PatchTST, and NeuralCDE. Notably, zero-shot TimeDiT outperformed these models. The traditional approach required training 18 distinct models across 6 PDE equations, resulting in significant computational overhead - approximately 18 times more training time - and extensive code modifications. This approach becomes increasingly impractical in real-world applications where multiple physical laws interact, as each new constraint would require training additional dedicated models. In contrast, TimeDiT's unified framework incorporates various physical constraints during inference while maintaining a single trained model, providing a more efficient and scalable solution.
> > >
> > > Table2. Comparison on the physics informed zero-shot TimeDiT with fully trained baselines.
> > > | | MSE | RMSE | MAE | CRPS |
> > > |---|---|---|---|---|
> > > | **Burgers** |||||
> > > | DLinear | 0.031(0.002) | 0.175(0.001) | 0.1261(0.005) | 1.400(0.057) |
> > > | PatchTST | 0.029(0.001) | 0.170(0.001) | 0.1252(0.004) | 1.411(0.051) |
> > > | NeuralCDE | 0.031(0.002) | 0.176(0.002) | 0.1257(0.005) | 1.397(0.061) |
> > > | TimeDiT | 0.011(0.001) | 0.104(0.005) | 0.083(0.003) | 1.395(0.053) |
> > > | **Vorticity** |||||
> > > | DLinear | 2.650(0.003) | 1.628(0.001) | 1.459(0.010) | 0.695(0.005) |
> > > | PatchTST | 2.651(0.002) | 1.628(0.002) | 1.460(0.012) | 0.700(0.001) |
> > > | NeuralCDE | 2.631(0.001) | 1.622(0.001) | 1.453(0.010) | 0.691(0.005) |
> > > | TimeDiT | 1.524(0.523) | 1.234(0.021) | 0.772(0.009) | 0.445(0.006) |

---

> ### Author Response · Authors · 2024-11-25
> **Kindly follow up!**
>
> Dear Reviewer v6Mm,
>
> Thank you for your detailed feedback on our TimeDiT paper. We're grateful for your thorough review of our responses regarding model capabilities, architecture design rationale, and physics-informed approach.
>
> We are incorporating these clarifications into our revised draft and would greatly appreciate your feedback on whether we have adequately addressed your concerns. We hope these improvements will further strengthen your evaluation of our work.
>
> Best,
>
> Authors

---

> > ### Comment · Reviewer_v6Mm · 2024-11-25
> > **Response to Rebuttal**
> >
> > I would like to thank the author for very detailed response and I really appreciate the efforts. My concerns on the risk of overclaiming and other experiments are at least partially resolved and I'm willing to increase my score after the discussion period. However, I still have a few remaining concerns.
> >
> > 1. Based on the author's response, it seems that some the task-specific dataset is excluded from pre-training and only used in the final model. Can the author explain why using the task-specific dataset during pre-training will compromise task-specific integrity?
> >
> > 2. The response says the authors incrementally incorporate training data in a sequential manner. Does that mean the datasets are incorporated in training sequentially in a particular order? If so, why can't all the datasets be used together and does the order of incorporating the datasets impact the model performance.
> >
> > 3. Can the author give a summary of the core original contributions of this work? It is not clear to me from the response or the claim of contributions in the work itself which are too broad or vague. Is it the proposal of a set of masking strategies in pre-training for time-series data or physics-informed TimeDiT?

---

> ### Author Response · Authors · 2024-11-25
> **Thanks for the positive response and willing to increase the score!**
>
> Dear Reviewer v6Mm,
>
> Thank you for your positive response and your willingness to enhance your score! We believe that addressing these questions will significantly aid readers in understanding our article more comprehensively. We deeply appreciate your thoughtful and insightful comments. Now we have a chance to address your concerns further:
>
> PS: While we acknowledge the length of our explanation, we believe it is necessary to fully convey both the design principles behind TimeDiT and the comprehensive effort required to develop a truly functional time series general model, outperforming multiple downstream tasks.
>
> **C1:** The decision to exclude specific datasets was made based on careful consideration of practical constraints. Let’s take anomaly detection as an example, this task requires full reconstruction of time series to identify anomalies via reconstruction error, which conflicts with our foundation model's masking strategies that are designed to learn normal temporal patterns. Including anomalous data in pre-training could compromise the model's ability to learn genuine temporal relationships. Furthermore, while the foundation model aims to minimize reconstruction error for normal patterns, including anomaly detection data in pre-training would force the model to reconstruct outliers, creating a fundamental conflict with the task's goal of identifying anomalies through reconstruction differences.
>
> Another crucial consideration is the variability in label definition for anomaly detection tasks. Unlike forecasting tasks where the objective is consistently focused on temporal progression, anomaly labels are highly context-dependent and vary across different scenarios. This variability necessitates fine-tuning to adapt to specific definitions of anomalies in different applications. Our design maintains flexibility for downstream applications by allowing specific anomaly detection scenarios, preventing potential degradation of anomaly detection performance from pre-trained reconstruction capabilities. This pinpoints the useful design of the foundation model for down stream tasks with complex and variable time series.
>
> Based on this discussion, we would like to extend it to Timedit's position as a general-purpose model. When we looked at the literature and asked the search engine or LLM what they considered to be a foundational model for time series, the key words we could find always matched the capabilities TimeDIT demonstrated and the design philosophy behind it. Thus, we would like to add the following clarification in our draft: “While acknowledging that there is always room for advancement, TimeDiT demonstrates the essential attributes expected of a foundation model: a generalized architecture capable of handling diverse input formats, strong multi-task capabilities across different time series challenges, inherent domain adaptability, and consistent performance across varied applications.”
>
>
> **C2:** We apologize for any confusion regarding our data incorporation process. To clarify, we do not train on datasets sequentially in any specific order. Rather, our mention of "incremental" incorporation reflects the practical constraints of our research process - as we progressed with our work, we gradually gained access to different datasets and computational resources. While we initially validated our model's performance on certain tasks (like imputation) before moving to others (like forecasting), this was part of our development process rather than a training requirement.
>
> In terms of the actual training implementation, once we specify the datasets for pre-training, our dataloader randomly samples batches from the entire pool of available data. This means that during any training iteration, the model can encounter samples from any of the included datasets, ensuring truly randomized training. The final pre-trained model therefore learns from all datasets simultaneously, not sequentially. We agree with the reviewer that there's no theoretical reason to incorporate datasets in any particular order, and our implementation reflects this through random sampling across all available data.
>
> Would you find it helpful if we clarify this point in the manuscript to prevent any misunderstanding about our training procedure?

---

> > ### Author Response · Authors · 2024-11-25
> > **Thanks for the positive response and willing to increase the score!**
> >
> > **C3:** Thank you for helping identify our core contributions. We would like to clarify that TimeDiT makes two significant technical advances:
> >
> > First, our masking mechanism represents a novel solution to fundamental time series challenges. While the concept of masking is established, our framework uniquely addresses time series-specific complexities like varying sampling rates and missing values through a unified architecture. This adaptation required solving non-trivial technical challenges to maintain both flexibility and effectiveness across diverse time series tasks.
> >
> > Second, we introduce a theoretically grounded approach for incorporating physics-based domain knowledge during the sampling process. This method innovatively allows domain knowledge integration without model retraining, using energy-based priors to ensure that generated samples adhere to physical constraints.
> >
> > In summary, rather than pursuing novelty through architectural complexity, our architectural choices reflect a careful balance between incorporating time series domain knowledge and maintaining general-purpose computational capabilities. This is evidenced by TimeDiT's ability to process multiple data types without specialized architectures, adapt to various tasks through flexible masking strategies, and incorporate domain knowledge during inference while maintaining its general-purpose nature.

---

> ### Author Response · Authors · 2024-11-27
> **Upload the new version of TimeDiT**
>
> Dear Reviewer v6Mm,
>
> Thank you for your thorough efforts and willingness to increase the score! Let us address our revisions systematically: the general modifications to the abstract and introduction are highlighted in red, while your specific suggestions are marked in blue. These comprehensive revisions strengthen our manuscript across several key areas and provide a more robust foundation for presenting TimeDiT's contributions:
>
> ### General Modifications (marked in red)
>
> - **Abstract Enhancement** *(lines 15–17)*:
>   Added explicit discussion of how **realistic data characteristics** challenge the development of **general-purpose time series models**.
>
> - **Current Limitations** *(lines 64–69)*:
>   Expanded the analysis of why **existing approaches** struggle to achieve **robust foundation model capabilities**. This analysis also underscores **TimeDiT's methodological contributions** in successfully handling **multiple tasks** and **diverse data characteristics**, including **irregular sampling patterns** and **multi-resolution inputs**.
>
> - **Data Understanding** *(lines 77–78)*:
>   Emphasized the **critical importance** of understanding **underlying data characteristics**.
>
> - **Design Motivation** *(lines 84–87)*:
>   Clarified how **TimeDiT's architecture** is fundamentally guided by **real-world application requirements**, demonstrating that our **component choices** directly address **specific practical challenges** rather than pursuing **novelty for its own sake**.
>
>
> These revisions provide a clearer motivation for our work and better contextualize TimeDiT's contributions within the broader landscape of time series modeling.
>
> ### Specific Responses to Your Comments (marked in blue)
> We have systematically addressed your feedback through comprehensive revisions, as outlined below:
>
> #### **General Revisions**
> To address your comments on **W1**, **W2**, and **C3**, we clarified the position of **TimeDiT** in **lines 976–987** and **lines 1008–1012**. These updates are designed to be read in conjunction with the general modifications in the introduction, providing a cohesive understanding of the highly motivated real-world challenges and the value of our proposed **TimeDiT**.
>
> #### **Specific Revisions**
>
> - **W1 & Q1**:
>   - Added **detailed training specifications** (lines 1025–1027).
>   - Included a **comparative analysis** in **Table 8**.
>   - Added **data usage clarification** in **Table 9**.
>   - Introduced a new **data usage strategy section** (lines 1068–1079) to address **C1** and **C2**.
>
> - **W3 & Q2**:
>   - Added a **discussion on limitations** (lines 537–539, Section 6).
>   - Introduced a new section on **physics-informed modeling** (Section C.2, lines 1395–1426).
>   - Included a **comparative analysis** in **Table 11**.
>
>
> These revisions provide a more comprehensive presentation of TimeDiT's capabilities and methodological choices. We welcome any additional questions or concerns you may have and look forward to your feedback.

---

> > ### Comment · Reviewer_v6Mm · 2024-11-29
> > **Post Discussion Review Change**
> >
> > I'm satisfied with most of the response of the authors during the discussion period and do not object to the acceptance of thos work. Taking the comments and concerns of other reviewers into consideration, I raised my rating to 6, marginally above acceptance threshold.

---

> ### Author Response · Authors · 2024-11-29
> **Thanks for the acknowledgement and raising the score!**
>
> Dear v6Mm,
>
> For the overall comments, thank you for your insightful review and for highlighting TimeDiT's key strengths, which include the general-purpose time series analysis,  comprehensive experimental validation, and novel incorporation of physics-based knowledge into the diffusion framework!
>
> For the follow-up, thank you for recognizing our response and increasing your score! Your comments have greatly helped us clarify and strengthen our paper's presentation and contribution. Please let us know if you have any further questions, and we will respond promptly!
>
> Best,
>
>  Authors

---

### Comment · Program_Chairs · 2024-11-24
**Explanation of multiple submissions**

This article appears to have been published in the ICML2024 Workshop. We think it is necessary for the authors to explain if they the relationship between these two articles.

[URL REDACTED BY PROGRAM CHAIRS]

---

> ### Comment · Program_Chairs · 2024-11-25
>
> The program chairs reviewed this case. The ICML is a workshop paper and does not count for dual submission. The URL was redacted.

---

> > ### Author Response · Authors · 2024-11-27
> > **Acknowledgment of Program Chairs' Clarification**
> >
> > Dear Program Chairs,
> >
> > Thank you for your prompt review and clarification regarding this matter! We sincerely appreciate your effort in addressing the situation. Your support in upholding the integrity of the review process is greatly valued!
> >
> > Best,
> >
> > Authors

---

### Author Response · Authors · 2024-12-04
**Summary of the Rebuttal Stage and Thanks for the Contributions**

Through the lens of constructive dialogue and rigorous peer review, the revised draft now clearly establishes a connection between our technical innovations and practical needs. It demonstrates how each design decision serves a purpose beyond mere novelty. We are grateful for the help of all the reviewers who have contributed to the development of TimeDiT into a more robust and well-founded contribution to the field. Now we summarize this journey of refinement from the original recognition, motivation, contribution, and improvements of our draft:

## 1. Summary of Recognition:

We sincerely thank all reviewers for their insightful feedback, which has significantly strengthened our work. The reviewers collectively highlighted TimeDiT's significant contributions across multiple dimensions. Specifically, **#v6Mm** particularly emphasized our novel integration of masked reconstruction and physics-based knowledge, while **#p3cf** commended the successful fusion of diffusion models with transformers and our comprehensive experimental validation. **#d2F8** focused on TimeDiT's strong performance across multiple applications and cross-dataset capabilities. **#aczi** additionally recognized our model's potential as a foundation model, praising its ability to handle diverse time series characteristics and real-world challenges. Together, the reviewers acknowledged TimeDiT's **technical innovation**, **practical utility**, **thorough validation**, and promising **foundation model capabilities**, with particular emphasis on its physics-informed approach and real-world applicability. Your constructive suggestions have helped us enhance both the theoretical foundations and practical implementations of TimeDiT.

## 2. Summarize the motivation and contribution:

TimeDiT emerges from the pressing challenges in real-world time series analysis where traditional approaches often falter. At its core, the motivation stems from three critical real-world challenges: the **inherent complexity of real-world time series data** (including variable channel sizes, missing values, and multi-resolution/irregular sampling), the diversity of downstream tasks requiring different inductive biases (such as forecasting, imputation, anomaly detection, and generation), and the need to incorporate domain-specific physical constraints. These challenges are particularly acute in practical applications where data is rarely ideal and often requires simultaneous handling of multiple complexities.

To address these challenges, TimeDiT makes several key contributions through its innovative unified framework. The model combines a transformer backbone with a diffusion model, enhanced by a novel Time Series Mask Unit that employs multiple masking strategies (random, block, and stride) to capture different temporal dependencies. This architecture is complemented by an adaptive processing mechanism for handling varying channel dimensions and a physics-informed sampling framework that integrates domain knowledge without requiring model retraining. The effectiveness of this approach is validated through comprehensive empirical results across diverse tasks and datasets, demonstrating superior performance in both specialized scenarios and zero-shot applications. The model's ability to maintain strong performance while handling real-world data complexities represents a significant step forward in developing practical, general-purpose time series foundation models.

## 3. Summarize our improvements:

### 3.1 Summary of the additional experiments:

In response to the valuable feedback received, we have substantially expanded our experimental validation to provide a more comprehensive assessment of TimeDiT's capabilities. Our enhanced evaluation framework now encompasses several key dimensions:

First, we evaluated TimeDiT against state-of-the-art baselines across diverse real-world scenarios. For forecasting tasks involving missing values, multi-resolution data, and irregular sampling, we compared against cutting-edge models including TimeMixer[1], TimeLLM[2], and MG-TSD[3]. To validate our physics-guided forecasting approach, we conducted comparisons between zero-shot TimeDiT and fully-trained models like DLinear[4], PatchTST[5], and NeuralCDE[6]. The zero-shot generalization capabilities were benchmarked against other pre-trained models such as TimeMixer[1], TimeLLM[2], and Timer[7].

Second, we broadened our evaluation scope beyond forecasting to other critical tasks. For imputation, we compared against Timer[7], TimeMixer[1], and iTransformer[8], while our anomaly detection capabilities were evaluated against TimeMixer[1] and iTransformer[8]. Additionally, we conducted extensive ablation studies examining key design choices, including:

---

> ### Author Response · Authors · 2024-12-04
> **Continue of our summary**
>
> - Transformer attention mechanisms for temporal relationship modeling
> - Impact of different mask unit configurations on general performance
> - Effectiveness of various conditional generation strategies
> - Necessity of patch tokenization for handling complex, inconsistent input structures
>
> These comprehensive experiments consistently demonstrate TimeDiT's superior performance across diverse scenarios, validating our unified approach to time series modeling. The results not only showcase TimeDiT's effectiveness but also provide valuable insights into the design principles behind its success.
>
> [1] Wang, Shiyu, et al. "TimeMixer: Decomposable Multiscale Mixing for Time Series Forecasting." The Twelfth International Conference on Learning Representations, 2024.
>
> [2] Jin, Ming, et al. "Time-LLM: Time Series Forecasting by Reprogramming Large Language Models." The Twelfth International Conference on Learning Representations, 2024.
>
> [3] Fan, Xinyao, et al. "MG-TSD: Multi-Granularity Time Series Diffusion Models with Guided Learning Process." The Twelfth International Conference on Learning Representations, 2024.
>
> [4] Zeng, Ailing, et al. "Are transformers effective for time series forecasting?." Proceedings of the AAAI conference on artificial intelligence. Vol. 37. No. 9. 2023.
>
> [5] Nie, Yuqi, et al. "A Time Series is Worth 64 Words: Long-term Forecasting with Transformers." The Eleventh International Conference on Learning Representations, 2023.
>
> [6] Kidger, Patrick, et al. "Neural controlled differential equations for irregular time series." Advances in Neural Information Processing Systems 33 (2020): 6696-6707.
>
> [7] Liu, Yong, et al. "Timer: Transformers for time series analysis at scale." Forty-first International Conference on Machine Learning (2024).
>
> [8] Liu, Yong, et al. "iTransformer: Inverted Transformers Are Effective for Time Series Forecasting." The Twelfth International Conference on Learning Representations, 2024.
>
> ### 3.2 Summary of the Changes Made to TimeDiT Paper
>
> The modifications to the TimeDiT paper can be organized into two major categories: general structural improvements (in red) and specific technical additions (in different colors corresponding to the reviewer). On the structural side(**#all[red]**), the paper was enhanced to communicate its real-world motivation and practical significance better. The abstract and introduction were revised to address how realistic data characteristics challenge existing models explicitly, why current approaches struggle with foundation model capabilities, and how TimeDiT's architecture was specifically designed to address practical requirements rather than pursuing novelty for its own sake. These changes create a stronger narrative around TimeDiT's practical value in handling real-world time series challenges.
>
> On the technical side, the paper was substantially expanded with new experimental results and detailed analyses. Key additions include comprehensive comparisons with state-of-the-art models (TimeMixer, TimeLLM, Timer, iTransformer) across multiple tasks(**#aczi[yellow], #d2F8[purple]**), detailed ablation studies demonstrating the contribution of each component(**#p3cf[green], #aczi[yellow]** ), expanded sections on physics-informed modeling(**#v6Mm[blue]**), and new analyses of the model's adaptability to different channel numbers(**#p3cf[green]**). The paper now includes clearer explanations of implementation details(**#v6Mm[blue]**), from token definition(**#d2F8[purple]**) and embedding architecture(**#d2F8[purple]**)  to data usage strategies(**#v6Mm[blue]**) and handling of missing values(**#p3cf[green]**), as well as failure scenarios analysis(**#p3cf[green]**). These technical additions strengthen the paper's empirical validation while making its methodological choices more reproducible and transparent. All changes were carefully tracked and cross-referenced, ensuring consistency throughout the document while maintaining a clear connection between the theoretical foundations and practical implementations.
>
> Once again, we extend our sincere gratitude to all reviewers for their thorough and constructive feedback. The diverse perspectives offered have helped us better articulate TimeDiT's contributions while addressing important methodological questions. Through this collaborative review process, the paper now presents a more robust and well-validated contribution to the field, for which we are deeply appreciative.

---

### Meta-Review · Area_Chair_G8tq · 2024-12-23

**Metareview:**

This paper introduces Time Diffusion Transformer (TimeDiT), a diffusion-based generative model for time series. The authors provide experiments on a variety of tasks, such as forecasting, imputation, and anomaly detection and claim that their work provides a “general foundation model for time series”. They introduce new physics-based constraints on diffusion and show how this can be used in different settings.

While the reviewers agreed upon the importance and timeliness of the work, the reviewers also raised a number of concerns. In particular, there were some questions about whether the claims that the work is a foundation model were substantiated, as the pretraining is not conducted on multiple datasets from different domains and then finetuned on new datasets. The authors response to this question about the datasets used during pretraining was inadequate and unclear.

Additionally, there was concern about the lack of ablations, missing baselines, and the novelty of the work. During the rebuttal period, the authors provided a new ablation analysis and added additional experiments. This was helpful and led some reviewers to increase their scores to borderline acceptance.

In the end, because the model is not pretrained on multiple datasets nor finetuned on new datasets, the claims around the method being a general-purpose foundation model have not been demonstrated. In fact, the different tasks and datasets employ different pretraining and thus there is not even a common method for pretraining used throughout. The method thus is more well positioned as a general self-supervised approach for time series modeling that can be used on different downstream tasks.

**Additional Comments On Reviewer Discussion:**

There was a lengthy rebuttal and a lot of discussion between the authors and reviewers. The authors conducted a number of new experiments to address reviewer concerns, including: (i) new comparisons with state-of-the-art models (TimeMixer, TimeLLM, Timer, iTransformer) across the different tasks, (ii) new ablation studies, (iii) expanded sections on physics-informed modeling, and (iv) additional analyses of the model's adaptability to different channel numbers. However, the core issues about the claims of the work being a foundation model were not addressed.

---

### Decision · Program_Chairs · 2025-01-22

Reject